# Scaling Small Agents Through Strategy Auctions

**Lisa Alazraki** [†1] **William F. Shen** [†2] **Yoram Bachrach** [3] **Akhil Mathur** [3]

## Abstract

Small language models are viewed as a promising, cost-effective approach to agentic AI, yet how their performance scales with task complexity remains unclear. While smaller agents match larger ones on simple tasks, it is unknown when large models become necessary and how to better leverage small agents. In this work, we show that small agents fail to scale with task complexity on deep search and coding tasks, and introduce *Strategy Auctions for Workload Efficiency* (SALE), a framework inspired by freelancer marketplaces. In SALE, agents bid with strategic plans scored by a cost–value mechanism and refined via shared auction memory, enabling per-task routing and continual self-improvement without training a router. On average, SALE reduces reliance on the largest agent by 52%, lowers overall cost by 35%, and consistently improves pass@1 with only a negligible token overhead. In contrast, established routers either underperform the largest agent or fail to reduce cost. These results suggest that small agents can be effectively "scaled up" through coordinated allocation and test-time self-improvement. More broadly, they motivate a systems-level view of agentic AI in which gains come less from ever-larger individual models and more from market-inspired coordination mechanisms that organize heterogeneous agents into efficient, adaptive ecosystems.

## 1. Introduction

Recent work on tool-augmented AI agents has led to growing optimism that small language models may suffice for many real-world applications. By offloading computation and knowledge to external tools, small agents are argued to

---
[†]Work done at Meta. [1]Imperial College London [2]University of Cambridge [3]Meta Superintelligence Labs. Correspondence to: Lisa Alazraki <lisa.alazraki20@imperial.ac.uk>, Akhil Mathur <akhilm@meta.com>.

*Proceedings of the $43^{rd}$ International Conference on Machine Learning*, Seoul, South Korea. PMLR 306, 2026. Copyright 2026 by the author(s).

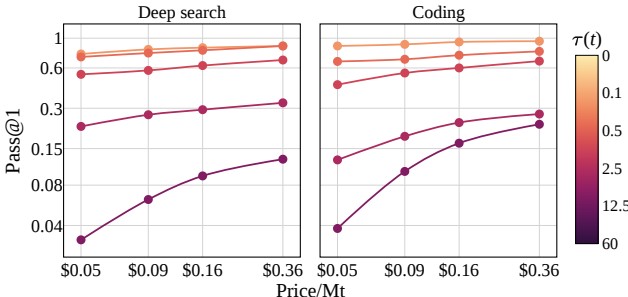

*Figure 1.* Pass@1 as a function of task complexity $\tau(t)$ (increasing top to bottom) and agent price per million tokens. $\tau(t)$ is measured in minutes (see §4 for details).

need less parametric capacity while still supporting complex, multi-step behavior (Houliston et al., 2025). Combined with advances that narrow the performance gap between small and large models (Hooker, 2025), this has led to claims that small, inexpensive agents can replace large ones as the foundation of agentic AI (Belcak et al., 2025).

Yet much of this optimism is framed in terms of model size and capabilities, with comparatively little attention to how these interact with task complexity. In practice, agentic workloads span a wide spectrum: from short, well-specified tasks to open-ended, long-horizon problems that require extended reasoning, integrating different types of information, and maintaining coherence over many steps (Wang et al., 2025d). It is not obvious that small agents performing well on simple tasks will also succeed as demands on reasoning, planning, and context management grow.

This raises two central questions: *how does task complexity mediate the relative effectiveness of small and large agents?* And given heterogeneous models with different capabilities and costs, *how should we route tasks across agents to balance accuracy and cost, maximizing the workload handled by small, cheap agents without degrading performance on complex tasks?* Existing approaches provide only a partial answer. Non-predictive strategies that generate full outputs are tractable for single-shot QA but infeasible for agents with trajectories spanning millions of tokens. Predictive routers, in turn, require training separate routing models that are costly, do not generalize well, and degrade as task difficulty increases (Dhrif, 2025).

To study how task complexity shapes small and large agents'

relative usefulness, we evaluate deep search and coding tasks across multiple horizons. These domains typify agentic workflows: deep search requires extended reasoning and information integration (Zhang et al., 2025), while coding demands multi-step planning and precise execution (Wang et al., 2025a). Following Kwa et al. (2025), we operationalize **task complexity** via human solution time: the average time expert annotators need to complete each task, from few seconds to one hour. Using off-the-shelf Qwen3 models (4B-32B parameters) and the Agent Research Environment (ARE) (Froger et al., 2025), we find that on the simplest tasks the smallest agent attains $\sim$90% of the largest agent's pass@1, but on the most complex tasks this drops to only $\sim$21% (ref. Figure 1 and Section 4). Small agents' performance thus fails to scale with task complexity, suggesting that small agents alone are unlikely to be sufficient for many high-value applications and that model size should be treated as a *per-task* decision rather than a global choice.

In response, we develop a routing mechanism that is compute-efficient, applicable to off-the-shelf agents, and preserves performance on complex tasks. Inspired by freelance marketplaces and virtual agent economies (Tomasev et al., 2025), we introduce **S**trategy **A**uctions for Work**l**oad **E**fficiency (SALE), a test-time auction framework that leverages a well-established correlation between plan quality and execution quality (Sun et al., 2024). Candidate agents propose strategic solution plans that are scored by predicted value and cost via peer assessment and heuristic predictors, and the winner is selected based on this trade-off. Crucially, plan refinement using past auction memory can overturn initial rankings before execution; a self-improvement process akin to freelancers upskilling to secure work. Thus SALE functions both as a router and as a mechanism that increases the share of work handled by smaller agents, effectively "scaling up" small models via market-like coordination.

We find that SALE exceeds the largest agent's pass@1 (+3.5% on deep search, +2.7% on coding) while offloading much of its workload (−65% and −40%) and reducing total spend (by 42% and 25%). These gains come with negligible additional inference tokens. In contrast, established routers that rely on task descriptions either underperform the largest agent or fail to reduce spend, often both, underscoring their poor fit for agentic workflows where complex trajectories decouple task inputs from downstream success. We also observe that, as the auction memory grows, the smallest agents are selected increasingly often, suggesting that they progressively capitalize on accumulated experience.

In summary, our contributions are:

1. We empirically study how task complexity affects the performance gap between small and large agents, finding that small agents nearly match large ones on simple tasks but diverge sharply as complexity increases. To the best of our knowledge, this is the first such investigation on realistic (non-synthetic) workloads.

2. We develop HST-BENCH[1], a benchmark pairing agentic tasks with human solution times as a complexity proxy.

3. We introduce SALE, a marketplace-inspired framework where heterogeneous AI agents bid with solution plans, are selected based on predicted value and cost, and use auction feedback to refine bids, coupling per-task routing with test-time self-improvement.

4. We show that SALE achieves a better performance–cost Pareto frontier than any individual agent or existing router, demonstrating that strategy-based routing with continual self-improvement outperforms baselines.

5. More broadly, we illustrate how auction-based coordination can structure competition and collaboration among heterogeneous agents at test time, contributing to discussions on how market dynamics may shape future ecosystems of interacting AI agents.

## 2. Related Work

**Agent performance under task complexity.** Scaling AI agents to handle long and difficult tasks is a central focus of recent work (Chan et al., 2025; Chen et al., 2025a; Froger et al., 2025; Wang et al., 2025d). Kwa et al. (2025) tie capability to task duration, defining a 50%-success time horizon in terms of human solution time and studying how it scales on research and software-engineering tasks. Sinha et al. (2026) examine performance degradation on synthetic multi-step tasks, arguing that many long-horizon failures reflect compounding execution errors. We build on both perspectives by analyzing scaling phenomena on realistic workloads and shifting from isolated model behavior to system-level performance in a marketplace allocating tasks across heterogeneous agents.

**Multi-agent routing.** Routing harnesses the diversity of heterogeneous AI systems via two main approaches: non-predictive routing, which selects outputs after running multiple models, and predictive routing, which chooses a model in advance (Hu et al., 2024). Non-predictive methods (Chen et al., 2024) are prohibitively expensive for agents with extended trajectories (Tsiourvas et al., 2025). Predictive approaches (Hu et al., 2024; Stripelis et al., 2024; Somerstep et al., 2025) mitigate this cost by learning separate routing models, but these are themselves costly to fit, model-specific, and have been shown to degrade with task difficulty (Dhrif, 2025). Moreover, existing routers are static and do not improve with experience. In contrast, SALE implements lightweight, strategy-based routing where agents bid with short plans rather than full solutions, leveraging evidence

---

[1] https://github.com/facebookresearch/hst-bench

that plan quality correlates with task success (Sun et al., 2024; Kang et al., 2025; Xiong et al., 2025b). Auction feedback and shared memory refine future bids, coupling routing with continual adaptation and turning agent selection from a purely passive assignment into a mechanism that actively improves small agents' effective capabilities under compute constraints.

**Memory-driven adaptation.** Memory systems help agents improve by reusing past behavior, typically by extracting reusable routines from successful trajectories (Cao et al., 2026; Wang et al., 2025e) or maintaining structured records of past interactions (Salama et al., 2025; Wang & Chen, 2025; Xu et al., 2025). SALE differs in that it stores bidding strategies and auction outcomes rather than execution traces, making memory an explicit mechanism for reallocating work and upgrading the effective capabilities of smaller agents.

**Agent systems as virtual economies.** Prior work argues that as autonomous agents become economically significant, they should be coordinated through explicit market mechanisms, e.g., auction-based interaction (Dütting et al., 2024; Shah et al., 2025; Jiang et al., 2025; Yang et al., 2025b), virtual sandbox economies (Tomasev et al., 2025), and settings where agents transact on behalf of users (Rothschild et al., 2026). Building on this perspective, SALE treats agents as freelancers in a job marketplace, where each job is awarded on the basis of both price and quality through a centralised scoring rule, a setting formally studied as an auction (Hong et al., 2016; Che, 1993). In SALE, bids over strategic plans allocate both work and learning opportunities.

# 3. Experimental Setup

We run all experiments within the Agent Research Environment (ARE) (Froger et al., 2025), which provides a standardized platform for benchmarking agents across domains.

## 3.1. Data

We evaluate on two domains representing agentic workflows requiring extended reasoning and multi-step planning: deep search and coding. Deep search tasks are sampled from SimpleQA (Wei et al., 2024), PopQA (Mallen et al., 2023), HotpotQA (Yang et al., 2018), GAIA (Mialon et al., 2024), and an expert-validated portion of Humanity's Last Exam (Phan et al., 2025; White, 2025). Coding tasks are drawn from MBPP (Austin et al., 2021) and LeetCode (Xia et al., 2025), supplemented with custom multiple-choice questions for lower-complexity cases. These benchmarks span a wide range of task horizons and have been widely adopted for evaluating agentic AI (Coignion et al., 2024; Labruna et al., 2024; Liu et al., 2024b; Amini et al., 2026; Gan et al., 2025; Huang, 2025; Xie et al., 2025).

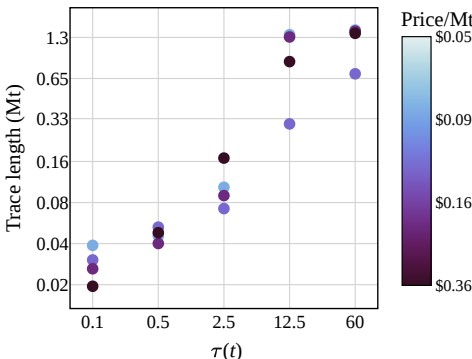

*Figure 2.* Avg. trace length by $\tau(t)$ for agents of different prices.

Following Kwa et al. (2025), who validate human solution time as a complexity metric that naturally integrates reasoning, planning, and execution, we define the *task complexity* of a task $t$ as $\tau(t)$, the average time (in minutes) required by expert annotators to solve $t$. Times are annotated by three experts with reliable agreement (Krippendorff's $\alpha = 0.86$; details in Appendix A.2). We group tasks into five non-overlapping bins by $\tau(t)$: up to 6 seconds, 30 seconds, 2.5 minutes, 12.5 minutes, and 60 minutes. As we show in Section 4, pass@1 degrades monotonically with $\tau(t)$ across all four agent sizes, indicating that HST tracks LLM-perceived difficulty closely on our task distribution. Bins follow a geometric progression ($5\times$ between adjacent bins), yielding equal log-scale spacing appropriate for times spanning nearly three orders of magnitude. The resulting benchmark, HST-BENCH, contains 753 tasks (see Appendix A).

## 3.2. Models

We use the Qwen3 family (Yang et al., 2025a), chosen for its open-weight availability and broad size range. Qwen3 provides checkpoints at 4B, 8B, 14B, and 32B parameters, which prior work has treated as a matched set for studying scaling behavior (Sinha et al., 2026). We define effective prices per million tokens $\pi(a_d)$ based on published API rates (see Appendix B) and observed 4:1 input-to-output ratio: $\pi(a_{4B}) = \$0.05$, $\pi(a_{8B}) = \$0.09$, $\pi(a_{14B}) = \$0.16$, and $\pi(a_{32B}) = \$0.36$. Since size and $\pi$ are monotonically aligned, we use *smaller/cheaper* and *larger/more expensive* interchangeably, though figures plot price explicitly. We run all models with greedy decoding.

# 4. Agent Performance vs. Task Complexity

We evaluate Qwen3 agents of different sizes on deep search and coding tasks, conditioning performance on task complexity $\tau(t)$, with pass@1 scored by an LLM judge against ground truth (see Appendix C.2). Across both domains, agents perform similarly on the simplest tasks (Figure 1).

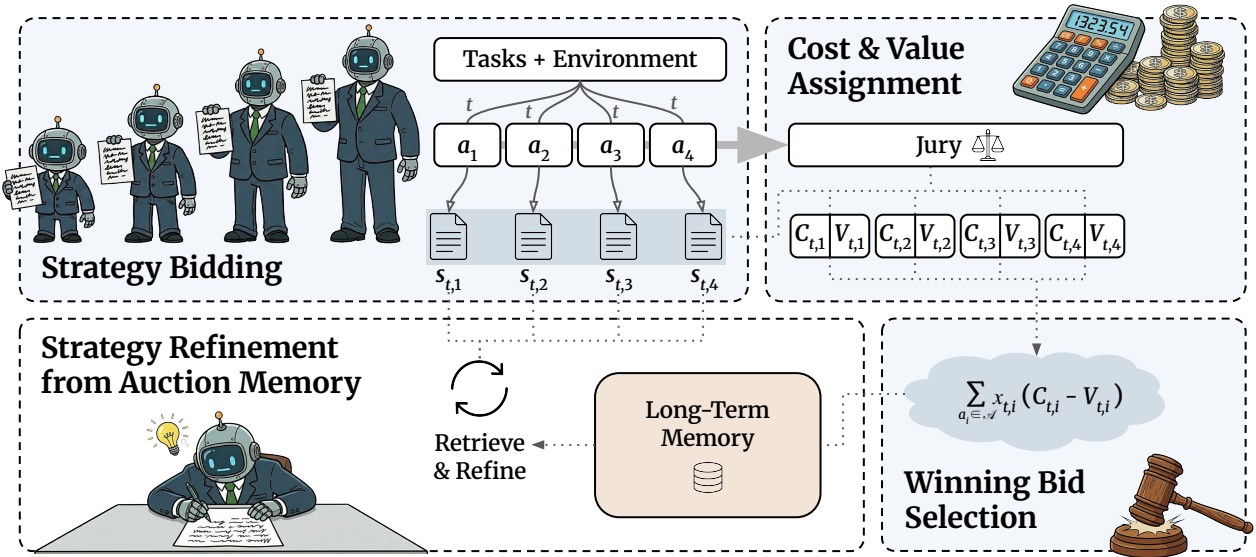

*Figure 3.* The SALE pipeline. Each agent $a_i$ bids with a strategy $s_{t,i}$, evaluated by predicted cost–value trade-off. Agents cheaper than the provisional winner may refine their strategies using past auction outcomes, after which a final winner is selected and its strategy is run.

For deep search, the cheapest agent achieves about 87% of the most expensive agent's pass@1 on tasks with $\tau(t) \leq 0.1$; for coding, about 92%. In this regime, scaling curves are nearly flat. As complexity increases, curves steepen, and by the most complex tasks ($\tau(t) \leq 60$), separation is sharp: the cheapest agent attains only 25% (deep search) and 17% (coding) of the most expensive agent's pass@1.

One might hope that, although larger agents are more expensive per token, they implicitly "pay for themselves" by solving tasks with shorter trajectories. However, as shown in Figure 2, this holds only for low-complexity tasks. As $\tau(t)$ increases, average token usage grows across all models, and larger agents do not consistently achieve shorter traces. Where shorter traces do occur, the savings are too small to offset the higher per-token cost. Hence, increased parametric capacity does not yield more token-efficient solutions on complex workloads, and higher per-token costs are not offset by reduced test-time compute.

In sum, cheaper agents are effective for low-$\tau(t)$ tasks but falter as demands intensify; expensive agents appear indispensable for complex problems yet waste resources on simpler tasks. The challenge, therefore, is to build systems that can dynamically allocate tasks to the right agent, achieving a better balance between resource efficiency and capability.

## 5. Strategy Auctions

Agentic pipelines commonly include a planning phase where agents outline their approach before acting. These strategic plans encode task-relevant information (decomposition strategies, tool selection, anticipated challenges) yet are rarely leveraged beyond the agent that produced them.

SALE (Figure 3) exploits this by casting strategic plans as bids in an auction. Given a task $t$ and heterogeneous agents $\mathcal{A} = \{a_i\}_{i=1}^{|\mathcal{A}|}$, each agent $a_i$ generates a strategy $s_{t,i}$ conditioned on $t$. We interpret $s_{t,i}$ as the "bid" of $a_i$ for task $t$, which is used to compute both cost and value of $a_i$, thus enabling agent selection based on strategic intent rather than task description alone (see Appendix C for the prompts used to obtain $s_{t,i}$).

**Cost and value assignment.** Let $C_{t,i}$ and $V_{t,i}$ denote the cost and value of deploying agent $a_i$ on task $t$. We estimate *cost* as $C_{t,i} = w_c \cdot \pi(a_i) \cdot |s_{t,i}|$, where $\pi(a_i)$ is price per million tokens, $|s_{t,i}|$ is strategy length, and $w_c$ is a tuned weight. Strategy length serves as a cost signal motivated by two prior works. First, Goebel & Zips (2025) shows that plan (or strategy) length is correlated with final trace length, hence serving as a proxy for total inference cost. Second, execution reliability degrades with plan length: prior work finds that success rates decline as plans grow longer (Xiong et al., 2025a). Because failed executions nonetheless consume compute, longer plans entail higher expected cost, both through greater token usage and increased risk of wasted computation. We also show thorough ablations in Appendix I to validate this design choice.

We estimate *value* as $V_{t,i} = w_h \cdot H(s_{t,i}) + \sum_{a_j \in \mathcal{A}} w_j \cdot \gamma_j(s_{t,i})$, where the weights $w_h$ and $w_j$ are tunable. Here, $H(s_{t,i})$ is the normalized entropy of $s_{t,i}$, computed as the mean per-token entropy from the generation log-probabilities, and each $\gamma_j(s_{t,i})$ is an integer judgment score in $\{0, 1, ..., 5\}$ produced by agent $a_j$ in response to a fixed scoring prompt (Appendix C.4). The jury comprises all agents in $\mathcal{A}$, including $a_i$. Value thus combines two signals: intrinsic quality, captured by entropy, and extrinsic quality, captured by self-

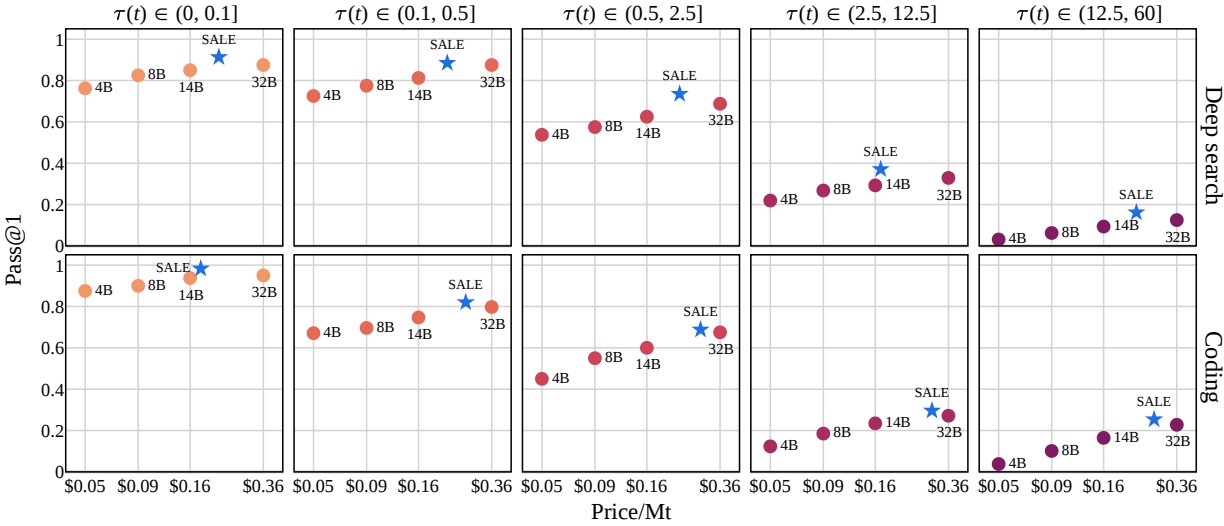

*Figure 4.* Performance–cost trade-offs across task complexity bins. At a given price/Mt, SALE consistently attains higher pass@1 than the linear scaling trend observed for individual Qwen3 agents, systematically exceeding the expected performance–cost curve.

and-peer assessment. The choice of entropy as a proxy for strategy value is motivated by extensive prior literature linking higher-entropy intermediate reasoning to greater informational content and reduced redundancy (Chen et al., 2026; Cheng et al., 2026; Li et al., 2026b; Wang et al., 2025b), and by work suggesting that prioritizing higher-entropy trajectories can be beneficial for planning (Liu et al., 2024a) (validated by our ablations in Appendix I). The use of a mixed self-and-peer jury is motivated by work on LLM juries (Badshah & Sajjad, 2025; Verga et al., 2024) and evidence that combining self-evaluation with peer assessment yields more reliable judgments (Mousavi et al., 2023), consistent with recent findings that 0–5 Likert scoring without chain-of-thought yields strong alignment with human judgments on contemporary models including Qwen3 (Li et al., 2026a; Wang et al., 2025c) (human validation in Appendix J; ablations in Appendix I; prompts in Appendix C).

**Winning bid selection.** SALE selects the agent achieving the optimal cost–value trade-off. We pose a min–max optimization over the assignment variables $x$ and the scoring weights $w = (w_c, w_h, \{w_j\}_{a_j \in \mathcal{A}})$. Let $Q$ denote the maximum cost-minus-value over all tasks. The objective is

$$\min_{w,x,Q} Q \quad \text{s.t.} \quad z_t \leq Q \ \forall t, \quad \sum_{a_i \in \mathcal{A}} x_{t,i} = 1 \ \forall t, \quad w \in \mathbb{R}^{2+|\mathcal{A}|}$$

where $z_t$ is the cost-minus-value for task $t$ and big-$M$ constraints are imposed (see Appendix D). At inference, for each task $t$ with binary assignment variables $x_{t,i} \in \{0,1\}$ indicating whether $a_i$ is selected for $t$, we compute $z_t = \sum_{a_i \in \mathcal{A}} x_{t,i} (C_{t,i} - V_{t,i})$, assigning each task to the agent with lowest cost-minus-value.

**Strategy refinement from auction memory.** After each auction, we store all proposed strategies in a memory bank

$\mathcal{M}$. This enables self-improvement: agents that are not selected can learn from $\mathcal{M}$, refine their strategies, and submit improved bids. Importantly, refinement is *opportunistic*: we first collect baseline strategies; if a cheap agent wins the auction, no refinement is needed; otherwise, only agents cheaper than the provisional winner produce refined, memory-informed bids.

Concretely, for each past task $t'$, we store a record $\mathcal{M}(t') = (t', \{s_{t',i}\}_{a_i \in \mathcal{A}}, y_{t'})$ where $\{s_{t',i}\}_{a_i \in \mathcal{A}}$ are the strategies proposed by all agents for $t'$ and $y_{t'}$ encodes the auction outcome, indicating which strategy won and which ones failed. Given a new task $t$, refinement proceeds as follows (full algorithm in Appendix E):

1. *Initial bids.* Each agent submits $s_{t,i}$ and a provisional winner $\hat{i}(t) = \arg\min_i (C_{t,i} - V_{t,i})$ is selected.

2. *Memory retrieval.* For agents cheaper than the provisional winner, we retrieve $k$ similar past tasks via cosine similarity and extract losing/winning strategy pairs for contrastive learning.

3. *Contrastive prompting.* Pairs are formatted to encourage learning from past outcomes (Appendix C.4).

4. *Reassignment.* Eligible agents produce refined bids $s_{t,i}^r$. The final assignment $i^*(t)$ is updated to the refined bid only if it achieves a lower cost-minus-value than the provisional winner; otherwise, $\hat{i}(t)$ is retained.

5. *Execution.* After selecting $i^*(t)$, we execute agent $a_{i^*(t)}$ conditioned on $t$ and its winning strategy.

It is worth noting that both jury scoring and refinement incur only a small additional inference cost, whereas executing the final agentic trace typically consumes tens of thousands to millions of tokens (see Figure 2), depending on task

*Table 1.* Pass@1 and price per million tokens ($/Mt) on deep search and coding across complexity bins, for baselines and SALE, including an ablated variant (SALE w/o memory). SALE results are averaged over five random orderings; subscripts show standard deviations.

| Task type | $\tau(t)$ | Best single agent | | WTP | | CARROT | | TO-Router | | FrugalGPT | | SALE w/o memory | | SALE | |
|---|---|---|---|---|---|---|---|---|---|---|---|---|---|---|---|
| | | Pass@1($\uparrow$) | $/Mt($\downarrow$) | Pass@1($\uparrow$) | $/Mt($\downarrow$) | Pass@1($\uparrow$) | $/Mt($\downarrow$) | Pass@1($\uparrow$) | $/Mt($\downarrow$) | Pass@1($\uparrow$) | $/Mt($\downarrow$) | Pass@1($\uparrow$) | $/Mt($\downarrow$) | Pass@1($\uparrow$) | $/Mt($\downarrow$) |
| *Deep search* | $\leq 0.1$ | 87.5 | 0.36 | 83.8 | 0.32 | 85.0 | 0.27 | 86.3 | 0.28 | 86.3 | 0.47 | **91.3** | 0.24 | **91.3**$_{0.0}$ | **0.22**$_{0.01}$ |
| | $\leq 0.5$ | 87.5 | 0.36 | 86.3 | 0.33 | 86.3 | 0.28 | 86.3 | 0.32 | 81.3 | 0.48 | 87.5 | 0.24 | **88.5**$_{0.5}$ | **0.22**$_{0.01}$ |
| | $\leq 2.5$ | 68.8 | 0.36 | 67.5 | 0.31 | 66.3 | 0.29 | 67.5 | 0.34 | 66.3 | 0.53 | 72.5 | 0.25 | **73.5**$_{1.2}$ | **0.23**$_{0.01}$ |
| | $\leq 12.5$ | 32.9 | 0.36 | 34.2 | 0.32 | 29.3 | 0.29 | 32.9 | 0.36 | 30.5 | 0.50 | 35.4 | 0.19 | **37.1**$_{1.8}$ | **0.17**$_{0.01}$ |
| | $\leq 60$ | 12.5 | 0.36 | 9.4 | 0.31 | 9.4 | 0.32 | 12.5 | 0.36 | 12.5 | 0.60 | 15.6 | 0.26 | **16.3**$_{1.3}$ | **0.23**$_{0.02}$ |
| | All | 63.8 | 0.36 | 62.4 | 0.32 | 61.3 | 0.28 | 63.0 | 0.33 | 61.0 | 0.51 | 66.4 | 0.24 | **67.3**$_{0.5}$ | **0.21**$_{0.00}$ |
| *Coding* | $\leq 0.1$ | 95.0 | 0.36 | 93.8 | **0.16** | 95.0 | 0.36 | 95.0 | 0.36 | 97.5 | 0.39 | 97.5 | 0.22 | **98.3**$_{1.0}$ | 0.18$_{0.00}$ |
| | $\leq 0.5$ | 79.7 | 0.36 | 76.0 | **0.15** | **82.3** | 0.25 | 79.7 | 0.36 | 69.6 | 0.61 | **82.3** | 0.28 | 82.0$_{0.5}$ | 0.27$_{0.01}$ |
| | $\leq 2.5$ | 67.5 | 0.36 | 60.0 | **0.15** | 60.0 | 0.26 | 67.5 | 0.36 | 56.3 | 0.61 | 68.8 | 0.31 | **69.0**$_{0.5}$ | 0.29$_{0.00}$ |
| | $\leq 12.5$ | 27.2 | 0.36 | 14.8 | **0.05** | 27.2 | 0.36 | 27.2 | 0.36 | 18.5 | 0.61 | 27.2 | 0.32 | **30.4**$_{2.2}$ | 0.30$_{0.02}$ |
| | $\leq 60$ | 22.8 | 0.36 | 6.3 | **0.05** | 21.5 | 0.35 | 22.8 | 0.36 | 10.1 | 0.61 | 24.1 | 0.31 | **26.1**$_{2.4}$ | 0.29$_{0.01}$ |
| | All | 58.4 | 0.36 | 50.1 | **0.11** | 57.1 | 0.31 | 58.4 | 0.36 | 50.4 | 0.57 | 59.9 | 0.29 | **61.1**$_{0.6}$ | 0.27$_{0.00}$ |

complexity. In both token and wall-clock terms, the auction phase accounts for under 1% of the total inference cost on average (Appendix E.5). *Thus, the overhead introduced by the auction mechanism is negligible relative to the overall test-time compute.*

# 6. Results

We run SALE on the full HST-BENCH test set with tasks from all complexity levels interleaved randomly, partitioning into bins only for analysis. All runs use greedy decoding. Single-model baselines are from a single run; since SALE's auction memory evolves online, we report metrics averaged over five random permutations following established practice (Fan et al., 2026; Wu et al., 2024).

Figure 4 summarizes performance–cost trade-offs across complexity bins; detailed results appear in Table 1. For deep search, SALE exceeds the best single agent's pass@1 by 3.8 points on the lowest-complexity tasks while reducing cost by 39%. On medium-complexity tasks, it improves

pass@1 by 1–4.7 points while reducing cost by 36–53%. On the most complex tasks, it still outperforms by 3.8 points at 36% lower cost. For coding, SALE similarly beats the best single agent on simple tasks (50% cost reduction, +3.3 pass@1), achieves 1.5–3.2 pass@1 improvement on medium-complexity tasks at 17–25% cost reduction, and improves by 3.3 points on the most complex tasks at 19% lower cost. Across all domains and complexity levels, the auction ensemble dominates the single-agent Pareto frontier: no fixed model attains higher pass@1 at equal or lower price per million tokens, indicating that strategy-based routing with self-improvement yields strictly better performance–cost trade-offs than any single model size. All improvements are statistically significant (Appendix F).

**Comparison with existing routers.** SALE is deliberately lightweight: it leverages agents' strategic-planning abilities through a low-dimensional scoring function rather than learning a separate routing model. We compare against four baselines. WTP (Hu et al., 2024) uses nearest-neighbor retrieval to predict performance–cost trade-offs. CAR-

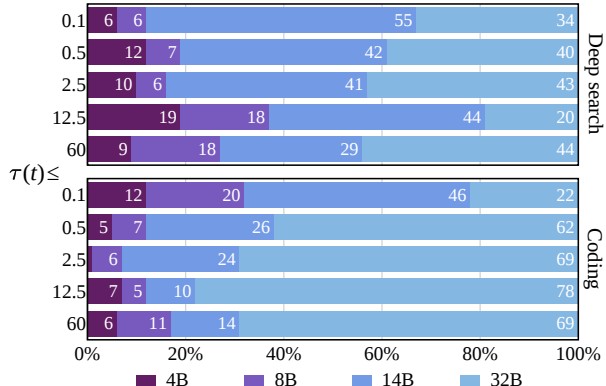

*Figure 5.* SALE workload allocation across agents for deep search and coding by complexity bin. Labels show task share per agent.

*Table 2.* Shapley values (%) of each agent's marginal contribution with and without memory-based refinement, across task types and complexity bins. Values are normalized to sum to 100.

| Task type | $\tau(t)$ | SALE w/o memory | | | | SALE | | | |
|---|---|---|---|---|---|---|---|---|---|
| | | 4B | 8B | 14B | 32B | 4B | 8B | 14B | 32B |
| *Deep search* | $\leq 0.1$ | 22.0 | 23.9 | 24.1 | 30.0 | 25.6 | 24.2 | 26.3 | 23.9 |
| | $\leq 0.5$ | 21.7 | 23.6 | 24.5 | 30.2 | 24.1 | 25.4 | 25.2 | 24.8 |
| | $\leq 2.5$ | 19.8 | 21.6 | 24.7 | 33.9 | 23.4 | 26.7 | 25.7 | 24.2 |
| | $\leq 12.5$ | 10.9 | 23.6 | 29.3 | 36.2 | 20.0 | 22.6 | 27.3 | 30.0 |
| | $\leq 60$ | 0.0 | 13.9 | 38.9 | 47.2 | 7.1 | 24.2 | 35.6 | 33.1 |
| *Coding* | $\leq 0.1$ | 23.2 | 23.8 | 24.3 | 28.7 | 26.2 | 24.5 | 23.6 | 25.8 |
| | $\leq 0.5$ | 21.4 | 23.2 | 24.0 | 31.4 | 19.1 | 24.8 | 26.4 | 29.7 |
| | $\leq 2.5$ | 16.2 | 20.5 | 28.0 | 35.3 | 25.0 | 25.0 | 25.0 | 25.0 |
| | $\leq 12.5$ | 3.8 | 17.4 | 22.0 | 56.8 | 13.2 | 27.3 | 35.8 | 23.7 |
| | $\leq 60$ | 0.0 | 9.6 | 24.6 | 65.8 | 9.3 | 26.5 | 33.8 | 30.4 |

ROT (Somerstep et al., 2025) fine-tunes an encoder to jointly estimate cost and accuracy. TO-Router (Stripelis et al., 2024) trains a classifier to predict the best-performing model from task descriptions. These are *predictive* routers selecting agents before execution. FrugalGPT (Chen et al., 2024) is a *non-predictive* cascade executing trajectories sequentially until a response is accepted. All baselines are trained on the same split as SALE's weights (Appendix G). As shown in Table 1, WTP yields modest cost reductions on deep search (11%) but underperforms at most complexity levels; on coding, it sacrifices pass@1 as complexity rises (dropping to 6.3% vs. 22.8% for the best agent). CAR-ROT reduces cost by 22% (deep search) and 14% (coding) with small accuracy drops but still underperforms SALE. TO-Router defaults to the strongest agent, so both metrics remain near baseline. FrugalGPT matches the best agent on simple tasks but declines sharply as complexity grows while costs increase, exposing how non-predictive routing struggles when long trajectories mediate task-solution mappings. In contrast, SALE maintains or improves pass@1 while reducing cost by 36–53% (deep search) and 17–50% (coding), advancing the Pareto frontier more consistently than any alternative (Figure 7; oracle comparison in Appendix H).

**Ablation.** To isolate the contribution of self-refinement, we remove the memory-based stage and evaluate a variant of SALE that performs only strategy-based routing. Table 1 shows that even without self-refinement, SALE matches or improves average pass@1 while reducing the cost in all bins. Comparing to full SALE shows that memory consistently improves the trade-off, pushing the Pareto frontier further, and remains effective even under aggressive pruning of memory size and retention window (Appendix I.3). Further ablations (Appendix I) confirm all cost–value terms contribute meaningfully and that jury diversity provides regularization no single judge can replicate.

# 7. Analysis

## 7.1. Agent Allocation

Figure 5 shows how SALE allocates workload across agents. For deep search, the 32B agent's share ranges from 20% to 44% across complexity bins, with the 14B handling 29–55% and the 4B/8B agents together accounting for 12–37%. Even in the highest-complexity bin, smaller agents process nearly 30% of tasks. For coding, the 32B agent handles a substantial proportion except in the easiest bin (22%), while the 4B/8B agents together account for 7%–32% across bins, demonstrating substantial offloading from the largest model.

## 7.2. Agent Contributions via Shapley Values

Given SALE's cooperative nature, we quantify each agent's contribution using Shapley values (Lundberg & Lee, 2017),

following established uses in the analysis of auction outcomes (Van Essen & Wooders, 2021; Lindsay, 2018). We define a cooperative game $(\mathcal{A}, \nu)$ where the value of coalition $\mathcal{A}' \subseteq \mathcal{A}$ is the performance achieved by SALE restricted to $\mathcal{A}'$:

$$\phi_i = \sum_{\mathcal{A}' \subseteq \mathcal{A} \setminus \{i\}} \frac{|\mathcal{A}'|! \, (|\mathcal{A}| - |\mathcal{A}'| - 1)!}{|\mathcal{A}|!} \left[ \nu(\mathcal{A}' \cup \{i\}) - \nu(\mathcal{A}') \right].$$

Table 2 reports values with and without memory. Without memory, the largest agent has highest Shapley values across all bins, even when it is not the most-selected agent (Figure 5), indicating SALE benefits from its jury contributions while saving costs by selecting smaller agents for execution. Introducing memory consistently lowers the 32B agent's Shapley value while smaller agents gain contribution, especially on complex coding tasks.

Since Shapley computation removes agents from all roles (candidate pool, jury, memory), an agent's Shapley value captures its total contribution including indirect effects. This explains, for example, why the 4B model can attain a relatively high Shapley value on coding tasks despite being selected infrequently for final inference. Hence, the distributions in Figure 5 and Table 2 need not correlate.

## 7.3. Smallest Agent Selection Over Time

Beyond static routing, SALE enables smaller agents to "scale up" through auction feedback, effectively expanding their competitiveness over time. We test this by tracking how often the smallest (4B) agent is selected as the final executor over time. Figure 6 plots the cumulative selection rate of the 4B agent. For deep search, this share grows from 3.7% early on to 11.1% by the final tasks (a threefold increase). For coding, it increases from 1.4% to 5.3% (a nearly four-fold gain), with most of the rise concentrated in the first ~150 tasks. As the memory bank accumulates, increasingly relevant feedback scales up smaller agents' practical contribution, distinguishing SALE from conventional routers that treat selection as stationary. Similar plots for all agents appear in Appendix K.

## 7.4. Qualitative Analysis of Refined Strategies

Consistent with work on reusable behaviors in LLM reasoning (Didolkar et al., 2025), we observe that memory-guided refinement tends to ground strategies in auction feedback by reusing recurrent structural elements from past winning bids on similar tasks. For search, refined strategies more frequently mention tools explicitly, impose tighter search-space constraints, and add cross-reference checks. For coding, refinement specifies function names precisely, maintains alignment with objectives, and performs unit tests, including for edge cases. Both domains show clearer layout and step organization. Table 10 reports pattern frequencies; Appendix E.4 shows representative examples.

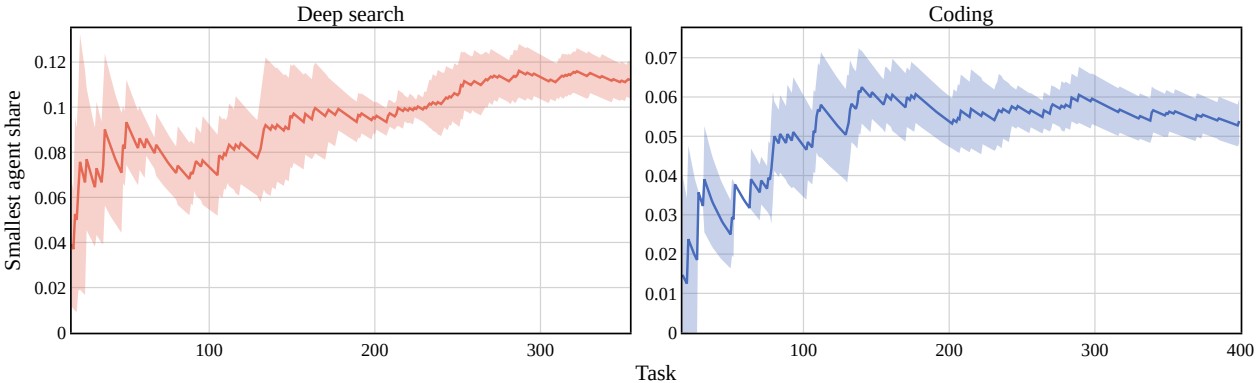

*Figure 6.* Cumulative share of tasks routed to the smallest agent over time. Solid lines show the mean across 5 runs with randomized task orderings; shading denotes ±1 standard deviation. Upward trends reflect increased delegation as auction history accumulates; short-term fluctuations reflect the random interleaving of tasks of varying complexity.

### 7.5. Complementary Failure Modes

For SALE to outperform any single agent, smaller agents must succeed where the largest agent fails. In Appendix L, we investigate such complementary failures. The largest agent often fails through overconfident behavior: bypassing tools for parametric recall, over-engineering simple problems, or skipping verification. Smaller agents more often adhere to simpler strategies and perform explicit checks. Crucially, these tendencies are visible in the initial strategic plans before execution, so the auction can predict the divergence in failure modes already at bidding time. While the largest agent remains superior on aggregate (Section 4), this consistent complementarity in failure modes allows SALE to exploit early strategy differences for workload division.

### 8. Conclusion

We investigated how task complexity affects the relative performance of small and large language-model agents, and how to allocate work across them efficiently. On deep search and coding tasks spanning multiple horizons, smaller agents perform comparably to larger ones on simple instances but fall sharply behind on more complex ones. This suggests that small agents alone are insufficient for complex workloads, whereas always defaulting to the largest model ignores substantial efficiency opportunities.

To address this, we proposed SALE, a strategy-auction framework where heterogeneous agents bid with strategic plans scored by a cost–value objective, and refine them using a shared auction memory. SALE runs at test time on off-the-shelf models without training a separate router, thus adding negligible overhead. Existing routers that rely on task descriptions tend to break down on complex agentic workflows or fail to reduce cost. SALE, instead, uses strategic plans as routing signals and maintains gains even on the most complex tasks. Empirically, it improves pass@1 over

the strongest single agent while reducing cost and shifting a substantial share of the workload away from the largest model. It also adaptively improves smaller agents over time.

Our findings indicate that scaling individual models is only one axis of progress, and *how* we coordinate agents can be equally as powerful. Rather than treating capability as a property of a single, ever-larger model, SALE reframes it as a quality emerging from a system that allocates work, prices compute, and lets agents learn from each other. This points toward a view of agentic AI where advances come not only from stronger models, but from coordination mechanisms that bind them into adaptive systems.

### Limitations

We study deep search and coding, which are standard benchmarks in the agentic AI literature. These domains test complementary capabilities: search requires retrieval and multi-step exploration, while coding emphasizes generation and logical reasoning. Together, they cover diverse agentic patterns, with distinct tool-use profiles, and offer objective, automatable evaluation metrics. That said, they do not span all applications of agentic systems; future work can apply SALE to additional task families (e.g., data analysis or long-form report writing) to test how broadly our findings generalize.

On the modeling side, we work with Qwen3 models from 4B to 32B parameters. We focus on a single model family for our complexity-scaling analysis because cross-family comparisons would confound scale effects with differences in architecture, tokenizer, and training recipe, thus making it impossible to isolate how model size alone mediates performance as task complexity grows. Qwen3 is the only contemporary open-weight model suite offering a dense, consistently-trained ladder (4B → 8B → 14B → 32B) suitable for this methodology. In contrast, other open-weight

families offer narrower size ranges, larger gaps between sizes, or mix architectures across scales, and closed-source models do not disclose parameter counts. Importantly, SALE is model-agnostic: the auction mechanism and cost–value objectives do not depend on model-specific properties, so our findings about when to route to larger versus smaller models should transfer qualitatively to other families. Indeed, we provide preliminary empirical support for this in Appendix M, where the auction mechanism produces complexity-sensitive allocations on the Gemma 3 family analogous to those observed for Qwen3. As for size, the 4B-32B range already yields a clear task–complexity-dependent performance gap and a roughly $8\times$ cost spread, but it still sits below the largest frontier models. That said, much of the empirical literature on scaling behavior draws inferences from trends observed across multiple smaller, systematically spaced model sizes, and our controlled size ladder is designed to support that style of analysis by isolating scale while holding other factors as constant as possible. Evaluating SALE with substantially larger models (e.g., 70B+) would be a useful extension to assess how the cost–value trade-offs behave when agents are even more capable and more expensive. While the controlled ladder suits our scaling analysis, SALE is designed to coordinate heterogeneous pools; we provide preliminary evidence of this in Appendix M, where Llama 3.1 and Ministral 3 agents are added to a Qwen3 pool as bidders and receive sensible allocations without retraining the MILP weights.

Our auction memory bank grows linearly with the number of tasks. In our experiments this remained tractable, and our ablations indicate that refinement gains remain robust under aggressive memory size and retention window pruning. We did not, however, study retrieval at much larger scales, across model-version upgrades, or under task-distribution shift where the memory may contain strategies from earlier distributions. For such settings, approximate nearest-neighbor indexing is recommended to scale retrieval to larger memories, while utility-based retention can prune stale or low-value records. We note that both integrate with our retrieval step without modifying the auction mechanism.

Finally, our agent cost computation focuses on language-model tokens and does not explicitly price tool calls. This is appropriate for our setup where token costs dominate, however, the modular cost function can be extended for deployments where tool costs are non-negligible. For instance, the cost term can be augmented with tool-related features extracted directly from the strategy, or per-token price can be replaced with a blended cost-per-step metric incorporating empirical tool latency. Neither extension requires changes to the auction mechanism; we provide a preliminary demonstration of this in Appendix D.5, where adding a tool-latency penalty to the cost term progressively shifts workload away from agents that generate the most tool calls.

## Impact Statement

This work contributes to our understanding of how task complexity shapes the effectiveness of language-model agents, and proposes a coordination mechanism for allocating work across heterogeneous models. We discuss several dimensions of potential impact below.

**On the marketplace metaphor.** We employ auction-based coordination and freelancer-marketplace terminology as conceptual tools for organizing AI agents, not as prescriptions for labor markets. This analogy is strictly methodological: it motivates a mechanism design perspective on multi-agent systems where bids, competition, and learning from feedback govern allocation among software components. We emphasize that the "agents" in our system are language models executing computational tasks within sandboxed research environments, and the "marketplace" is a metaphor for principled resource allocation; distinct from, and not intended to inform, policies regarding human employment.

**Efficiency and environmental considerations.** By reducing reliance on the largest agent by approximately 52% and lowering overall inference cost by 35%, SALE promotes a more efficient use of computational resources. Given the substantial energy footprint of large-scale language-model inference, mechanisms that route simpler tasks to smaller models without degrading performance can contribute to more sustainable AI deployment.

**Democratizing access to capable AI.** Our findings suggest that carefully coordinated ensembles of smaller, less expensive models can match or exceed the performance of larger models on heterogeneous workloads. If such coordination mechanisms become practical, they may lower the cost barrier to deploying capable agentic systems, potentially broadening access beyond well-resourced institutions. We should note, however, that the benefits of efficiency gains are not automatically equitably distributed, and deployment contexts will shape who ultimately benefits.

**Dual-use considerations.** As with most advances in AI capability and efficiency, our contributions are dual-use. More efficient agentic systems could be applied to beneficial domains (e.g., scientific research, accessibility tools) or to applications with negative societal consequences. In particular, mechanisms that lower the cost of agentic capability, whether through cheaper base models or, as here, through more efficient coordination, can amplify the scale at which both beneficial and harmful applications become feasible. We do not foresee unique risks introduced by SALE beyond those already present in the underlying language models and agentic frameworks; our contribution is to coordination, not to novel capabilities. Nonetheless, we encourage practitioners to consider deployment contexts carefully.

# Acknowledgments

We would like to thank Enrique Alfonseca, Misha Bilenko, Cheng Zhang, Yue Zhang, Igor Tufanov, Virginie Do, Emilien Garreau, Amine Benhalloum, Mathieu Rita, Romain Froger, Lovish Madaan, Anirudh Goyal, Iva Simon-Bubalo, Cindy Lee, Derek George Chan, Jordan Ward, and Joshua Lim for their valuable technical guidance and support in the development of this work. We are also grateful to Parag Jain, Amar Budhiraja, Graeme Nail, Thomas Scialom, Grégoire Mialon, Marin Vlastelica, Jenny Zhang, Md Rifat Arefin, Ulyana Piterbarg, Shashwat Goel, Philipp Mondorf, and Dulhan Jayalath for insightful discussions that helped shape and refine this research.

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

# A. Dataset

We describe here how we construct HST-BENCH, its composition across source datasets and complexity bins, and the details of the human solution–time annotation protocol.

## A.1. Data Composition

HST-BENCH is built from existing open-source benchmarks spanning deep search and coding. Concretely, we draw from SimpleQA (Wei et al., 2024), PopQA (Mallen et al., 2023), HotpotQA (Yang et al., 2018), GAIA (Mialon et al., 2024), Humanity's Last Exam (HLE) (Phan et al., 2025), MBPP (Austin et al., 2021), and LeetCode (Xia et al., 2025). In addition, we construct a small corpus of multiple-choice coding questions, which we refer to as Coding-MCQ (see example questions in Appendix A.4), to better populate the lowest-complexity bin for coding tasks. We randomly sample instances from the official test splits of each benchmark. In order to ensure label quality, we validate all samples, discarding and replacing those for which it is not possible to derive the provided ground-truth answer from the question. For HLE, we restrict to chemistry and biology questions that have been validated by domain experts (White, 2025). For GAIA, we sample from the validation split, which includes human solution times collected under comparable experimental conditions (timed, independent problem-solving by proficient users) and verified by the original authors; we directly reuse these annotations. After sampling, we annotate and aggregate human solution times for each instance and assign it to one of the five non-overlapping complexity bins defined in Section 3.1, based on its average human solution time. Table 3 reports, for each complexity bin, how many HST-BENCH instances originate from each source dataset. This reveals a shift from short-form factual QA and CODING-MCQ in the lower-complexity bins toward tasks demanding extended agentic workflows: multi-source information retrieval, cross-referencing, and synthesis for reasoning benchmarks (e.g., HotpotQA, GAIA, HLE), and iterative implementation with intermediate testing and debugging for coding problems (e.g., LeetCode 'Hard') in the higher-complexity bins.

*Table 3.* Composition of HST-BENCH by complexity bin (grouped by average human solution time $\tau(t)$, in minutes). We report the percentage of instances contributed from each source dataset to each bin, rounded to the nearest integer.

| Domain | Complexity bin | # tasks | Source | |
|---|---|---|---|---|
| | | | **Dataset** | **Percentage** |
| *Deep search* | $0 < \tau(t) \le 0.1$ | 80 | SimpleQA | 38% |
| | | | PopQA | 50% |
| | | | HotpotQA | 13% |
| | $0.1 < \tau(t) \le 0.5$ | 80 | SimpleQA | 8% |
| | | | PopQA | 5% |
| | | | HotpotQA | 88% |
| | $0.5 < \tau(t) \le 2.5$ | 80 | HotpotQA | 98% |
| | | | HLE | 3% |
| | $2.5 < \tau(t) \le 12.5$ | 82 | HotpotQA | 2% |
| | | | GAIA | 98% |
| | $12.5 < \tau(t) \le 60$ | 32 | HLE | 100% |
| *Coding* | $0 < \tau(t) \le 0.1$ | 80 | Coding-MCQ | 100% |
| | $0.1 < \tau(t) \le 0.5$ | 79 | MBPP | 100% |
| | $0.5 < \tau(t) \le 2.5$ | 80 | MBPP | 99% |
| | | | LeetCode (Medium) | 1% |
| | $2.5 < \tau(t) \le 12.5$ | 81 | MBPP | 2% |
| | | | LeetCode (Medium) | 98% |
| | $12.5 < \tau(t) \le 60$ | 79 | LeetCode (Hard) | 100% |

The distribution of source datasets across complexity bins reflects the design intent of existing benchmarks, many of which target specific difficulty ranges. This naturally results in a greater proportion of certain datasets within specific bins.

In addition to the test split, we construct separate development sets for both domains. For deep search, the development set contains 68 instances sampled from SimpleQA; for coding, it comprises 88 instances drawn from 40 Coding-MCQ questions and 48 LeetCode 'Easy' problems. These development sets are disjoint from the test data and reflect the need for instances on which models exhibit a balanced mix of successes and failures to enable effective validation and tuning.

## A.2. Data Annotation

To obtain human solution times for HST-BENCH, we recruited a pool of paid annotators who are graduates in computer science or closely related fields, with demonstrated expertise in programming and familiarity with the types of deep search and coding problems we study. This helps ensure that the reported solution times reflect the behavior of reasonably proficient users rather than novices. For each task, we collect solution-time annotations from at least three distinct annotators, who work independently and use only tools permitted by the task guidelines (e.g., a web browser and search engine for search tasks, or a local editor/IDE for coding), while refraining from language models or other assistants that could directly solve the task. Annotators are given written task-specific guidelines (Section A.3), read each task once in full, then start a stopwatch, solve the task as quickly as possible while maintaining accuracy, and finally submit both their measured solution time and final answer (or code). For LeetCode 'Hard' tasks, due to annotation cost constraints, we do not collect new human timings and instead rely on published human time estimates reported by Siroš et al. (2024). To verify consistency, we independently annotate a random subset of 8 tasks ($\sim$10%) and confirm that all measured times fall within the published ranges.

All collected annotations undergo a subsequent quality-control pass. First, submitted solutions are checked for correctness. Once a minimum of three correct solution times has been collected for a given task, we lightly filter for outliers to reduce the influence of anomalous timings (e.g., due to interruptions or misunderstandings), and collect further annotations if necessary. Concretely, solution times associated with incorrect answers or that deviate by more than two standard deviations from the task-wise mean are removed from the dataset. Once quality control and any necessary data re-collection have concluded, the times for each task are averaged together. We find good inter-annotator agreement across HST-BENCH (CCI $= 0.83$, 95% CI $[0.81, 0.85]$; Krippendorff's $\alpha = 0.86$, 95% CI $[0.84, 0.87]$, $p < 0.001$), indicating that human solution times are reliably reproducible.

## A.3. Annotators' Guidelines

Below we reproduce the instructions provided to annotators for both deep search and coding tasks. These guidelines specify the allowed tools, what constitutes a correct solution in each domain, and how annotators should measure and report their solution times, ensuring consistency across annotators and task types.

**Deep search guidelines.** *Read these instructions very carefully. Only after you have understood them well, navigate to the tasks in the next tab.*

*Goal*

*The goal of this annotation exercise is to label how much time it will take a human (not an LLM) to solve a given question.*

*You will be provided with questions and you will need to solve each with web searches, using Google or Bing.*

*You will need to use a stopwatch to measure your task completion time.*

*Task completion time must be reported in the format <HH hours MM minutes SS seconds>, for example, 25 seconds would be written as <00 hours 00 minutes 25 seconds> . DO NOT REPORT MILLISECONDS, EVEN IF YOUR STOPWATCH SHOWS THEM.*

*BE FAST: We are trying to measure a human's \*BEST\* completion time, so please complete the task (correctly) as quickly as you can. While for most questions you will likely need web search, it is fine not to use it if you already know the answer. For some tasks, it*

*is likely that you will need multiple, in-depth web searches.*

*It is assumed that the search engine is already open in a tab. To avoid wasting time unnecessarily, please arrange the windows on your screen so that you can see both the question and the search engine side by side at the same time.*

*Solve the task by following these steps:*

*- Step 1: Read the question first, slowly and carefully.*
*- Step 2: Start the stopwatch.*
*- Step 3: Text can be copy-pasted to the search engine directly from the question. Indeed, for most questions this is advisable as it can save time. As soon as the answer to the question is found, stop the stopwatch (i.e., do not wait to type the answer) and record the completion time.*
*- Step 4: Provide the answer and the task completion time (as per the stopwatch).*

*Note: You are allowed to read the question directly from the AI-generated summary at the top of the search engine page, if this is given. However, you are not allowed to copy-paste the question into an LLM chat interface. Use Google or Bing search.*

***Examples:***

*Question:*
*What is Miley Cyrus' occupation?*

*Completion Time: 00 hours 00 minutes 04 seconds*
*Your Answer: Singer, songwriter, actress*

*—*

*Question:*
*Which came out first, Titanic or Clueless?*

*Completion Time: 00 hours 00 minutes 17 seconds*
*Your Answer: Clueless*

**Coding guidelines.** *Read these instructions very carefully. Only after you have understood them well, navigate to the tasks in the next tab.*

***Goal***

*The goal of this annotation exercise is to label how much time it will take a human (not an LLM) to solve a given coding question.*

*You will be provided with coding questions and you will need to solve each by writing code.*

*You will need to use a stopwatch to measure your task completion time.*

*Task completion time must be reported in the format <HH hours MM minutes SS seconds>, for example, 25 seconds would be written as <00 hours 00 minutes 25 seconds> . DO NOT REPORT MILLISECONDS, EVEN IF YOUR STOPWATCH SHOWS THEM.*

*BE FAST: We are trying to measure a human's \*BEST\* completion time, so please complete the task (correctly) as quickly as you can. You are allowed to use web search to look up syntax, however please do not overuse web search unnecessarily, as it tends to increase the completion time.*

*If the question requires writing code, you MUST use a Python shell which allows running code at the click of a button. For example, use Google Colab or https://pythonhow.com/python-shell . For code-writing questions, you will be provided with one single test to check your code. We will run your code on more tests later to validate its correctness.*

*It is assumed that the Python shell and the search engine are already open in a window. To avoid wasting time unnecessarily, please arrange the windows on your screen so that you can see both the question text, the coding editor and the search engine side by side at the same time.*

*Solve the task by following these steps:*

*- Step 1: Read the question first, slowly and carefully.*
*- Step 1a: If the question requires writing a Python function, copy the function header and, at the bottom, the given test into your Python shell \*BEFORE\* you start the stop watch. The required function name and arguments will be clear from the test.*
*     For example you may have:* `def my_function(my_arg):`



`assert my_function(3)==True`


*     So that when the stopwatch starts you will only need to write the function body.*
*- Step 2: Start the stopwatch.*
*- Step 3a: If the question is multiple choice, stop the stopwatch as soon as the correct answer has been identified (no need to type it anywhere) and record the completion time.*
*- Step 3b: If the answer requires writing code, stop the stopwatch as soon as you have completed and run the code, and record the completion time.*
*- Step 4: Provide the answer and the task completion time (as per the stopwatch).*

*Note: You are allowed to use Google, but not allowed to use AI Assistants.*

***Examples:***

*Question:*
*Which of the following lines of code is the correct way to raise a to the power of b in python? Give only the number corresponding to the answer, and nothing else.*

1: `a^b`
2: `a**b`

*Completion Time: 00 hours 00 minutes 02 seconds*
*Your Answer: 2*

—

*Question:*
*Write a python function to find the first even number in a given list of numbers.*

*Your function should satisfy the following test:*
`assert first_even ([1, 3, 5, 7, 4, 1, 6, 8]) == 4`

*Completion Time: 00 hours 00 minutes 39 seconds*
*Your Answer:*
```
def first_even(nums):
    first_even = next((el for el in nums if el%2==0), -1)
    return first_even
```

## A.4. Coding-MCQ Examples

Below are representative multiple-choice questions from the Coding-MCQ dataset, designed to assess performance on short, low-complexity coding tasks that target core programming concepts.

---

Which of the following lines of code prints the word 'hello'? Give only the number corresponding to the answer, and nothing else.

```
1: print('hello') if 1%2==0 else print('goodbye')

2: print('goodbye') if 1%2==0 else print('helloworld'[:5])
```

---

Which of the following lines of code will not throw an error in Python? Give only the number corresponding to the answer, and nothing else.

```
1: 100 & 100

2: 100.0 & 100.0
```

---

Which of the following files is a configuration file? Give only the number corresponding to the answer, and nothing else.

```
1: run_agent.yaml

2: README.md

3: run_agent.py
```

---

Which of the following lines of code returns an empty list in python? Give only the number corresponding to the answer, and nothing else.

```
1: [elem for elem in [2,3,4,5] if elem // 2 == 0]

2: [elem for elem in [2,3,4,5] if elem % 2 == 0]
```

---

Which of the following lines of code correctly replaces a character in a string in Python? Give only the number corresponding to the answer, and nothing else.

```
1: "a,b,d".replace("d", "c")

2: [char if char in "a,b" else "c" for char in "a,b,d"]
```

---

# B. Estimated Cost of Running Models

Inference prices for Qwen3 models vary substantially across providers and deployment settings, reflecting differences in supported context length, geographical region, and commercial factors such as traffic volume and competition.[2] To obtain a simple, reproducible cost model for our experiments, we adopt an empirically calibrated pricing schedule. Our approach is grounded in recent empirical analyses of inference markets demonstrating that, for dense models, per-token prices scale approximately linearly with the number of parameters (Scher, 2025).

Following this established relationship, we model cost as proportional to the number of parameters and anchor our schedule using publicly advertised prices from established inference providers. Specifically, at the time of writing Groq reports separate prices for input and output tokens for Qwen3 32B,[3] listing

$$\$0.29/\text{Mt for input tokens} \quad + \quad \$0.59/\text{Mt for output tokens},$$

where *Mt* denotes one million tokens. We use these figures as a reference anchor for a high-capacity Qwen3 model and scale costs for other sizes in proportion to their parameter counts.

In our agentic runs we consistently observe an average input-to-output token ratio of about 4:1 across task domains and horizons. Under this assumption, we take the expected cost per million *total* tokens for an agent instantiated with Qwen3 32B to be

$$\pi(a_{32B}) \;=\; \frac{4 \cdot 0.29 + 1 \cdot 0.59}{5} \;\approx\; \$0.36/\text{Mt}.$$

Applying the same linear scaling in parameter count yields the effective prices per million total tokens used in our experiments:

$$\pi(a_{4B}) = \$0.05, \quad \pi(a_{8B}) = \$0.09, \quad \pi(a_{14B}) = \$0.16,$$

where $\pi(a_{4B}) \approx 0.045$ is rounded to $0.05.

**Empirical validation.** To verify that the linear scaling assumption holds in practice for the other Qwen3 sizes in our experiments, we compare our derived prices against independently advertised rates from major inference providers at the time of writing.[4][5][6] Since providers list separate prices for input and output tokens, we compute comparable per-million-total-token rates by applying the same 4:1 input-to-output weighting used in our estimates. Table 4 reports this comparison. The mean absolute deviation between our estimates and observed provider averages is within 6%, confirming that the linear approximation is well-supported for this model family.

*Table 4.* Comparison of estimated prices (derived via linear scaling from Qwen3 32B) against average advertised prices across providers (Nebius, Novita, Alibaba Cloud). Deviations are within 6%, validating the linear cost model.

| Model | Estimated ($/Mt) | Provider Avg. ($/Mt) | Deviation |
|-------|------------------|----------------------|-----------|
| Qwen3-4B | 0.05 | 0.05 | 0% |
| Qwen3-8B | 0.09 | 0.09 | 0% |
| Qwen3-14B | 0.16 | 0.17 | 6% |

**Scope of the cost model.** Our cost model assumes access via third-party inference APIs, where infrastructure overhead (including hardware provisioning, energy consumption, and maintenance) is fully absorbed into the provider's per-token pricing. Under usage-based API billing, the user incurs costs only for tokens consumed, making $/Mt the appropriate metric for our analysis. We note that latency and throughput vary substantially across providers, regions, and time of day, making them difficult to model consistently (though smaller models are typically also faster); we therefore leave them outside the scope of our study. Per-token pricing, by contrast, is publicly advertised and stable, providing a reproducible basis for cost comparison.

---

[2]https://huggingface.co/datasets/reach-vb/inference-provider-pricing
[3]https://groq.com/pricing
[4]https://nebius.com/token-factory/prices
[5]https://novita.ai/pricing
[6]https://www.alibabacloud.com/help/en/model-studio/models

# C. Environment, Prompts and Hyperparameters

All experiments are conducted within the open-source Agent Research Environment (ARE), which provides a standardized, tool-augmented interface for evaluating heterogeneous language-model agents on real-world tasks. Unless otherwise noted, agents, tools, and evaluation protocols follow the default ARE configuration. In the subsections below, we describe the exact model hyperparameters, environment prompts, and other implementation details needed to fully reproduce our setup.

## C.1. Model Hyperparameters

All experiments were run on NVIDIA A100 and H100 GPU clusters with 40–80 GB of HBM per accelerator. We use the same decoding configuration across all agents. The full set of model- and decoding-related hyperparameters used in our experiments is summarized in Table 5.

*Table 5.* Decoding and batching hyperparameters used for all Qwen3 agents.

|  | **Max length** | **Temperature** | **Top-p** | **Top-k** | **Batch size** |
|---|---|---|---|---|---|
| Values | 40,960 | 0.0 | 1.0 | 0 | 10 |

## C.2. Environment Hyperparameters

We run all experiments under the default configuration of ARE. Each episode is terminated as soon as either the time or iteration budget is exhausted, and agents must return a single final solution (i.e., we report pass@1 under the environment's LLM-as-a-judge evaluation). Notably, the LLM-as-a-judge evaluation is straightforward in our setup: search outputs are directly matched against ground truth, while coding outputs, though potentially differing lexically, can be reliably compared for functional equivalence by models like GPT-4o. We refer readers to the ARE default configuration for LLM-as-a-judge prompts and other standard hyperparameters not explicitly mentioned here. We report environment hyperparameters in Table 6.

*Table 6.* Environment-level limits used for all tasks; an episode terminates when either limit is reached.

| Hyperparameter | Value | Description |
|---|---|---|
| `task_timeout_seconds` | 3600 | Maximum wall-clock time per task |
| `max_iterations` | 100 | Maximum agent steps per task |
| `llm_judge` | GPT-4o | Base model for LLM-as-a-judge evaluation |

*Table 7.* Tools and execution environment available to agents. Deep search tasks use all tools; coding tasks use `ask_search_agent`, `final_answer`, and the Python environment.

| Tool | Description |
|---|---|
| `ask_search_agent` | Delegates a natural-language query to a web search agent and returns its response. |
| `inspect_file_as_text` | Reads a file from the workspace as markdown text and returns its contents for subsequent inspection and reasoning |
| `final_answer` | Submits the agent's final solution and terminates the episode. |
| Python environment | Executes Python code for calculations, data manipulation, and lightweight scripting; it is preconfigured with the standard library and commonly used packages sufficient to solve the benchmark tasks. |

For deep search tasks, each episode begins with a *fact extraction* pre-step, followed by a *strategy planning* step. For coding tasks, the agent performs only a strategic planning step without explicit fact extraction. Table 7 summarizes the tools and Python execution environment available to the agent in each domain and provides brief descriptions of their functionality.

For all remaining environment details (e.g., the exact tool interfaces, the format of observations returned to the agent, and error handling), we refer the reader to the original ARE paper (Froger et al., 2025).

### C.3. Retrieval Hyperparameters

We use embedding-based retrieval over the shared auction memory: at each episode, the agent retrieves strategies that both won and lost past auctions for the $k$ most similar tasks to the current one. Following established practice, we set $k = 8$, which prior studies commonly find to be a strong practical trade-off between coverage/diversity and context/latency overhead (Dai et al., 2023; Wang et al., 2024; Rashid & Hakak, 2025). Table 8 summarizes the retrieval hyperparameters used in all experiments.

*Table 8.* Strategy retrieval hyperparameters used across all experiments.

| | **Embedding model** | **Distance metric** | **Top-$k$** |
|---|---|---|---|
| Values | `all-MiniLM-L6-v2` | cosine | 8 |

### C.4. Prompts

We detail here the *judge* prompt for scoring candidate strategies, the *strategy* prompts used to generate $s_{t,i}$ for a task $t$, and the *refinement* prompts to produce $s_{t,i}^r$. The judge prompt uses a 0–5 Likert scale rather than a wider range (e.g., 0–100), based on preliminary experiments where 0–5 yielded higher correlation with human scores, consistent with Li et al. (2026a). For coding tasks, we omit fact extraction before planning as the task specification is self-contained. Strategy refinement uses a template similar to that introduced by Alazraki et al. (2025) to facilitate contrastive learning from prior outcomes. For all remaining prompts used to interact with the environment, tools, and task wrappers, refer to the standard ARE configuration.

---

**Judge Prompt**

```
Provide an integer reward score between 0 and 5 (inclusive) for the quality
of the provided plan steps, using strict evaluation standards.  Ensure the
reward reflects how effectively the plan contributes to progressing toward the
correct solution.

Problem Statement:
***begin problem statement***
{task}
***end problem statement***

Plan:
{plan}

Be harsh in your evaluation.  Only plans that you are extremely confident will
succeed should be assigned the maximum score.

Score:  [Strictly provide an integer reward score between 0 and 5]
```

---

**Strategy Prompt (Deep Search)**

```
You are a world expert at making efficient plans to solve any task using a set
of carefully crafted tools.

Now for the given task, develop a step-by-step high-level plan taking into
account the following inputs and list of facts.
This plan should involve individual tasks based on the available tools, that
if executed correctly will yield the correct answer.
Do not skip steps, do not add any superfluous steps.  Only write the
high-level plan, DO NOT DETAIL INDIVIDUAL TOOL CALLS.
After writing the final step of the plan, write the '<end_plan>' tag and stop
there.
Always search for the exact task at the beginning.  If you are given an
external file, always inspect it first to explore its content.
Do a very concise plan that only focus on the given task.
Do not attempt to answer the question without calling tools, even if you know
the answer.  You must always use at least one tool to find the answer.

Here is your task:

Task:

```
{task}
```

Your plan can leverage any of these tools:
{tool_descriptions}

List of facts that you know:
```
{answer_facts}
```

Now begin!  Write your plan below.
```

## Strategy Prompt (Coding)

```
You are a world expert at making efficient plans to solve any task using a set
of carefully crafted tools.

Now for the given task, develop a step-by-step high-level plan taking into
account the following inputs.
This plan should involve individual tasks based on the available tools, that
if executed correctly will yield the correct answer.
Do not skip steps, do not add any superfluous steps.  Only write the
high-level plan, DO NOT DETAIL INDIVIDUAL TOOL CALLS.
After writing the final step of the plan, write the '<end_plan>' tag and stop
there.
Do a very concise plan that only focus on the given task.
Do not attempt to answer the question without calling tools, even if you know
the answer.  You must always use at least one tool to find the answer.

Here is your task:

Task:

```
{task}
```

Your plan can leverage any of these tools:
{tool_descriptions}

Now begin!  Write your plan below.
```

**Strategy Refinement Prompt (Deep Search)**

You are a world expert at making efficient plans to solve any task using a set
of carefully crafted tools.

Now for the given task, develop a step-by-step high-level plan taking into
account the following inputs and list of facts.
This plan should involve individual tasks based on the available tools, that
if executed correctly will yield the correct answer.
Do not skip steps, do not add any superfluous steps.  Only write the
high-level plan, DO NOT DETAIL INDIVIDUAL TOOL CALLS.
After writing the final step of the plan, write the '<end_plan>' tag and stop
there.
Always search for the exact task at the beginning.  If you are given an
external file, always inspect it first to explore its content.
Do a very concise plan that only focus on the given task.
Do not attempt to answer the question without calling tools, even if you know
the answer.  You must always use at least one tool to find the answer.

Here is your task:

Task:

```
{task}
```

Your plan can leverage any of these tools:
{tool_descriptions}

List of facts that you know:
```
{answer_facts}
```

Below you will find some example tasks followed by two corresponding plans –
one plan that lost in a previous plan competition and one that won.  Use these
examples to understand what makes a plan lose or win.

{retrieved_tasks_and_plans}

Now apply what you have learned and given the task and a corresponding losing
plan, write a winning plan.

{previous_losing_plan}
Winning plan:

**Strategy Refinement Prompt (Coding)**

```
You are a world expert at making efficient plans to solve any task using a set
of carefully crafted tools.

Now for the given task, develop a step-by-step high-level plan taking into
account the following inputs.
This plan should involve individual tasks based on the available tools, that
if executed correctly will yield the correct answer.
Do not skip steps, do not add any superfluous steps.  Only write the
high-level plan, DO NOT DETAIL INDIVIDUAL TOOL CALLS.
After writing the final step of the plan, write the '<end_plan>' tag and stop
there.
Do a very concise plan that only focus on the given task.
Do not attempt to answer the question without calling tools, even if you know
the answer.  You must always use at least one tool to find the answer.

Here is your task:

Task:

```
{task}
```

Your plan can leverage any of these tools:
{tool_descriptions}

Below you will find some example tasks followed by two corresponding plans -
one plan that lost in a previous plan competition and one that won.  Use these
examples to understand what makes a plan lose or win.

{retrieved_tasks_and_plans}

Now apply what you have learned and given the task and a corresponding losing
plan, write a winning plan.

{previous_losing_plan}

Winning plan:
```

# D. Cost–Value Optimization

## D.1. Cost and Value Function Design

**Strategy length in cost function.** In our setting, strategy length correlates with realized trajectory length (Spearman's $\rho = 0.39$, Pearson's $r = 0.36$, both $p < 0.001$), consistent with Goebel & Zips (2025). The close alignment between rank-based and linear correlation measures indicates a stable monotonic relationship rather than an artifact driven by outliers, reinforcing that longer strategies generally yield longer trajectories. In agentic planning, such a stable and consistent proxy signal provides meaningful guidance for cost estimation. Although the correlation is moderate in absolute magnitude, this level of predictive strength is sufficient in practice: prior work on agentic orchestration demonstrates that proxies of relatively moderate strength can meaningfully inform decision-making, even in the presence of systematic over- or underestimation (Amin, 2026). Crucially, including $|s_{t,i}|$ in $C_{t,i}$ improves development set accuracy, and removing it degrades performance in ablations (Appendix I).

**Entropy and jury in value function.** We use normalized entropy $H(s_{t,i})$, computed as the mean per-token entropy from generation log-probabilities, as a lightweight signal of informational content/non-redundancy. In addition to being motivated by prior work linking higher-entropy planning to better outcomes (see Section 5), this term is also validated by development set performance and confirmed by ablations. Finally, we score each strategy with a full jury (including self-judgment): this configuration performs best on validation, learned weights calibrate each judge's influence, and ablations show that removing any judge reduces accuracy (Appendix I).

## D.2. Min-Max Formulation

In Mathematical Program 1, we provide the full mixed-integer linear program (MILP) for learning the scoring weights $w = (w_c, w_h, \{w_j\}_{a_j \in \mathcal{A}})$. Let $\mathcal{T}$ denote the training set of tasks.

---

**Mathematical Program 1** Min-Max Weight Optimization

---

$$\min_{w,x,z,Q} \quad Q$$

$$\text{s.t.} \quad \sum_{a_i \in \mathcal{A}} x_{t,i} = 1 \qquad \forall t \in \mathcal{T}$$

$$x_{t,i} \in \{0,1\} \qquad \forall t, a_i$$

$$z_t \geq C_{t,i} - V_{t,i} - M(1 - x_{t,i}) \qquad \forall t, a_i$$

$$z_t \leq C_{t,i} - V_{t,i} + M(1 - x_{t,i}) \qquad \forall t, a_i$$

$$z_t \leq Q \qquad \forall t \in \mathcal{T}$$

$$w \in \mathbb{R}^{2+|\mathcal{A}|}$$

---

The big-$M$ constraints ensure that $z_t = C_{t,i} - V_{t,i}$ only for the selected agent (i.e., when $x_{t,i} = 1$), while remaining inactive otherwise. We set $M = 10^4$, chosen to exceed the observed range of $C_{t,i} - V_{t,i}$ scores across all task-agent pairs in the training set. Specifically, $M$ exceeds $\max_{t,i} |C_{t,i} - V_{t,i}|$ by at least two orders of magnitude, ensuring numerical stability

without introducing solver issues.

### D.3. Implementation Details

We solve the MILP using PuLP (Mitchell et al., 2011) with the CBC solver. Weights are unconstrained. The optimization is performed separately on the development sets for deep search and coding tasks (see Appendix A for details), with $|\mathcal{A}| = 4$ agents.

The learned weights exhibit consistent patterns across both domains. Entropy receives the highest weight in both cases, suggesting that information density in the generated strategy is a strong indicator of expected performance. This aligns with prior findings on the informativeness of high-entropy reasoning (Li et al., 2026b) and is further supported by our ablation study in Appendix I. The relative importance of individual judge models varies by domain: for deep search tasks, smaller model judgments receive higher weight, whereas for coding tasks, the largest agent (32B) contributes most to the value estimate.

### D.4. Alternative Formulations

**Rationale for the optimization objective.**  The difference $C_{t,i} - V_{t,i}$ can be interpreted as a *negative net utility*: lower cost and higher value both reduce this quantity, so minimizing it naturally favors agents that are both efficient and effective. This formulation mirrors classical economic frameworks where profit is defined as revenue minus cost. Crucially, because the weights $(w_c, w_h, \{w_j\})$ are learned jointly and left unconstrained, the optimization implicitly calibrates the relative scales of cost and value components without requiring manual normalization or threshold selection.

**Normalized features.**  An alternative approach is to normalize $C$ and $V$ before combining them, for instance by Z-score, standardizing all input features prior to learning the weights. While this ensures that cost and value terms are on comparable scales a priori, it discards meaningful structural information. In our setting, cost and value features operate on different scales: entropy in particular is orders of magnitude smaller than cost, while judge scores fall in between. These differences are not arbitrary; instead, they reflect the heterogeneous nature of the signals: token-based costs, normalized entropy, and Likert-scale judgments each carry information at their natural scale. Our learned weights implicitly calibrate these differences, ensuring that smaller-scale features (particularly entropy, which receives the largest weight magnitude) contribute meaningfully to routing decisions.

To validate this intuition empirically, we evaluated a variant in which all input features are Z-score normalized using training-set statistics before weight optimization. This variant achieves lower accuracy while disproportionately selecting cheaper models. Inspection of the learned weights reveals the cause: under normalization, the optimizer discovers a degenerate solution in which several value-related weights become negative (e.g., $w_h < 0$, $w_{32B} < 0$), effectively *penalizing* quality signals rather than rewarding them. This occurs because, once all features are rescaled to comparable magnitudes, the optimizer can minimize $C - V$ simply by selecting cheap models; there is no structural pressure to preserve value contributions. In contrast, our unnormalized formulation forces the learned weights to jointly encode both relative importance and scale calibration, preventing such degenerate solutions.

**Constrained optimization.**  Another alternative is constrained optimization, for example minimizing cost subject to $V > \eta$ for some threshold $\eta$. However, this formulation requires choosing $\eta$ a priori, which is difficult in practice: the appropriate threshold may vary across task distributions, agent pools, and deployment settings. Setting $\eta$ too high excludes cost-efficient agents that would have succeeded on easier tasks, while setting it too low allows poor-quality assignments to slip through on harder ones. Our unconstrained formulation sidesteps this problem altogether. The learned weights end up implicitly encoding the right cost–value trade-off for the training distribution, and because the min–max objective optimizes against worst-case tasks, we do not need to hand-pick thresholds for robustness.

In conclusion, normalization and constrained optimization both introduce additional design choices (normalization statistics, threshold values) that need to be tuned and often do not transfer well across settings. By minimizing $C - V$ with unconstrained, jointly learned weights, we avoid this overhead, while still obtaining strong empirical results. The natural scale differences between cost and value features are informative on their own, and the learned weights can exploit this, as confirmed by ablation experiments showing that adding explicit normalization hurts routing quality.

## D.5. Tool-Latency Cost Extension

To demonstrate that the modular cost function naturally accommodates non-token costs, we simulate a deployment where tool calls contribute to overall latency. We augment the cost term with a tool-call penalty:

$$C_{t,i} = w_c \cdot \pi(a_i) \cdot |s_{t,i}| + \beta \cdot n_{\text{tools}}(s_{t,i}) \cdot \ell,$$

where $n_{\text{tools}}(s_{t,i})$ counts the tool calls extracted from the strategy, $\ell$ is a synthetic average per-call latency, and $\beta$ controls the penalty strength. Note that the value function is unchanged. We sweep $\beta$ from 0 (original cost function) to 5 (strong tool penalty) on a 100-task subset evenly stratified by type and complexity. Results are reported in Table 9.

As $\beta$ grows, the 32B agent's overall share drops from 23% to 11% (a 52% relative reduction), reflecting its higher per-token price combined with the highest tool density in its generated strategies. The 14B agent rises from 18% to 28% overall, offering the best cost–value trade-off under tool latency. The effect is also domain-sensitive: when we consider only deep search tasks, where tool use is inherent, the 32B share drops from 20% to 8%. The MILP converges to optimal at every $\beta$, confirming that the cost function accepts this extension without modifications to the auction mechanism.

*Table 9.* Agent allocation under increasing tool-latency penalty $\beta$, overall and broken down by domain. The 32B agent, which generates the highest mean tool-call count (2.48 vs. 0.64 for the 4B), is progressively deselected as $\beta$ grows, with the sharpest decline observed in the deep-search subset where tool use is inherent.

| $\beta$ | Subset | 4B | 8B | 14B | 32B |
|---|---|---|---|---|---|
| | **All** | **26%** | **33%** | **18%** | **23%** |
| 0.0 | Deep search | 28% | 34% | 18% | 20% |
| | Coding | 24% | 32% | 18% | 26% |
| | **All** | **29%** | **34%** | **21%** | **16%** |
| 1.0 | Deep search | 30% | 38% | 18% | 14% |
| | Coding | 28% | 30% | 24% | 18% |
| | **All** | **32%** | **29%** | **28%** | **11%** |
| 5.0 | Deep search | 34% | 32% | 26% | 8% |
| | Coding | 30% | 26% | 30% | 14% |

# E. Strategy Refinement

## E.1. Algorithm

In Algorithm 1, we provide a complete description of the strategy refinement procedure outlined in Section 5. The algorithm details how initial bids are collected, how the provisional winner is selected, and how cheaper agents refine their strategies using retrieved contrastive examples from the auction memory $\mathcal{M}$.

---

**Algorithm 1** Strategy Refinement from Auction Memory

---

**Require:** Task $t$, agents $\mathcal{A}$, memory $\mathcal{M}$, learned weights $w$, retrieval size $k$
**Ensure:** Final agent assignment $i^*(t)$

  **for** each agent $a_i \in \mathcal{A}$ **do**
    Generate initial strategy $s_{t,i}$
    Compute cost $C_{t,i} \leftarrow w_c \cdot \pi(a_i) \cdot |s_{t,i}|$
    Compute value $V_{t,i} \leftarrow w_h \cdot H(s_{t,i}) + \sum_{a_j \in \mathcal{A}} w_j \cdot \gamma_j(s_{t,i})$
  **end for**

  $\hat{i}(t) \leftarrow \arg\min_i(C_{t,i} - V_{t,i})$
  $i^*(t) \leftarrow \hat{i}(t)$

  **for** each agent $a_i \in \mathcal{A}$ where $\pi(a_i) < \pi(a_{\hat{i}(t)})$ **do**
    $\tilde{k} \leftarrow \min(k, |\mathcal{T}_{\mathcal{M}}|)$
    Retrieve $\mathcal{T}' \leftarrow \text{top-}\tilde{k}_{t' \in \mathcal{T}_{\mathcal{M}}} \, \text{sim}(t, t')$
    $\mathcal{M}_{t,i} \leftarrow \{(s_{t'}^{\text{lose}}, s_{t'}^{\text{win}})_i \mid t' \in \mathcal{T}'\}$
    Format $\mathcal{M}_{t,i}$ as contrastive examples
    Generate refined strategy $s_{t,i}^r$ conditioned on $s_{t,i}$ and $\mathcal{M}_{t,i}$
    Compute $C_{t,i}^r \leftarrow w_c \cdot \pi(a_i) \cdot |s_{t,i}^r|$
    Compute $V_{t,i}^r \leftarrow w_h \cdot H(s_{t,i}^r) + \sum_{a_j \in \mathcal{A}} w_j \cdot \gamma_j(s_{t,i}^r)$
    **if** $C_{t,i}^r - V_{t,i}^r < C_{t,i^*(t)} - V_{t,i^*(t)}$ **then**
      $i^*(t) \leftarrow i$
    **end if**
  **end for**

  **return** $i^*(t)$

---

## E.2. Contrastive Example Selection

The refinement step exposes agents to contrastive pairs of winning and losing strategies from past auctions. Since the auction objective minimizes cost-minus-value, winning strategies tend to originate from cost-efficient agents—typically smaller models achieving competitive value. This aligns naturally with our refinement setting, where only cheaper agents refine: the positive examples shown to a refining agent come from models of comparable capability. Prior work has shown that smaller models can benefit from exposure to high-quality plans (Sun et al., 2024; Kang et al., 2025; Xiong et al., 2025b), yet often struggle to execute strategies designed by much larger models when the required reasoning exceeds their capacity (Chen et al., 2025b; Lee et al., 2025). By using auction winners, which inherently balance quality with executability, as positive examples, our approach provides smaller models with effective yet feasible strategies to learn from.

An alternative design would label contrastive examples by downstream task success rather than auction selection. However, this assumes access to a reliable automated evaluator, an assumption that often fails in practice, where ground-truth feedback

may be sparse, noisy, or available only through human assessment or ground truth labels. The auction selection signal, by contrast, emerges naturally from the mechanism itself and requires no external oracle, making our refinement procedure broadly applicable.

## E.3. Refinement Patterns

Table 10 reports, for both search and coding tasks, how often each refinement pattern appears in selected winning strategies. These statistics show that the observed structural elements recur consistently across tasks.

*Table 10.* Refinement patterns and their prevalence in chosen refined plans for search and coding tasks. Note that patterns are not mutually exclusive; a single refined plan may exhibit multiple patterns simultaneously.

| Domain | Refinement pattern | Prevalence |
|---|---|---|
| *Deep search* | Improved tool and argument specification | 68% |
| | Provides (additional) trusted sources for search | 15% |
| | Intermediate cross-reference/consistency checks | 32% |
| | Clearer/more structured layout | 49% |
| *Coding* | Precise function and helper signatures | 33% |
| | Explicit focus on returning the correct artifact | 30% |
| | Systematic testing and edge-case coverage | 41% |
| | Clearer/more structured layout | 26% |

## E.4. Representative Examples

Below we show side-by-side representative examples of initial and refined strategies for individual tasks, together with the refinement patterns (see Table 10) identified in each. These examples illustrate how refinement turns loosely specified or linear strategies into more constrained, check-pointed procedures that score higher in the auction. In all examples, the unrefined strategy is rejected and the refined one is selected, yielding a successful downstream answer.

### E.4.1. DEEP SEARCH EXAMPLES

---

**Example 1**

**Task**

Do The Importance of Being Icelandic and The Five Obstructions belong to different film genres?

---

**Unrefined plan**

1. Use the `ask_search_agent` to determine the genre of "The Importance of Being Icelandic" by querying a reliable film database or source. Provide the query: "What is the primary film genre of 'The Importance of Being Icelandic'? Provide the genre name."
2. Use the `ask_search_agent` again to determine the genre of "The Five Obstructions" with the query: "What is the primary film genre of 'The Five Obstructions'? Provide the genre name."
3. Compare the two genre names obtained from the searches. If they are distinct, conclude that the films belong to different genres; otherwise, state they belong to the same genre.

---

**Refined plan**

1. Use `ask_search_agent` to determine the primary film genre of "The Importance of Being Icelandic" by querying a reliable source like IMDb or Wikipedia. Provide `output_content` as the genre name.
2. Use `ask_search_agent` to determine the primary film genre of "The Five Obstructions" using the same method. Provide `output_content` as the genre name.
3. Compare the two genre names obtained from the searches.
4. Use `final_answer` to state whether the films belong to different genres based on the comparison.

---

**Refinement patterns**

Improved tool and argument specification, provides (additional) trusted sources for search

---

**Example 2**

**Task**

According to Google Finance, when was the first year the Apple stock went above $50 (without adjusting for stock split)?

---

**Unrefined plan**

1. Use the `ask_search_agent` to query Google Finance for the first date when Apple's stock (AAPL) exceeded $50 without adjusting for stock splits. Specify the query as: "According to Google Finance, what was the first date Apple's stock (AAPL) went above $50 without adjusting for stock splits? Provide the date."
2. Extract the year from the identified date.
3. Verify the result by cross-checking Apple's stock split history to ensure the price threshold is unadjusted.
4. Use the `final_answer` tool to confirm the first year.

---

**Refined plan**

1. Use `ask_search_agent` to query Google Finance for the earliest date Apple's stock (AAPL) exceeded $50 **without adjusting for stock splits**. Query: "According to Google Finance, what was the first date Apple's stock (AAPL) closed above $50 without adjusting for stock splits? Provide the date."
2. Extract the year from the retrieved date.
3. Use `ask_search_agent` to confirm Apple's stock split history to ensure the identified date is **pre-split** (e.g., check split dates and ratios). Query: "When did Apple (AAPL) undergo stock splits? Provide split dates and ratios."
4. Cross-reference the split dates with the retrieved date to verify it is unadjusted.
5. Use `final_answer` to submit the confirmed year.

---

**Refinement patterns**

Improved tool and argument specification, intermediate cross-reference/consistency checks

---

**Example 3**

**Task**

How many pages of the 2023 IPCC report (85 pages version) mention nuclear energy?

---

**Unrefined plan**

1. Use `ask_search_agent` to search for the 2023 IPCC report's 85-page version text, specifying the need for the full content to analyze mentions of nuclear energy. Provide the query: "What is the full text of the 2023 IPCC report (85 pages version)? Provide the complete document content."
2. Use `inspect_file_as_text` to analyze the retrieved report text (assuming the search provides a file path or URL) and identify all instances of terms related to nuclear energy (e.g., "nuclear energy," "nuclear power," "nuclear fission").
3. Count the unique pages containing these terms from the inspected text.
4. Use `final_answer` to report the total number of pages mentioning nuclear energy.

---

**Refined plan**

1. Use `ask_search_agent` to locate the official 2023 IPCC report (85 pages version) and retrieve its full text or a downloadable PDF link from the IPCC website (https://www.ipcc.ch/). Provide the query: "Where can I access the full text of the 2023 IPCC report (85 pages version)? Provide a direct download link."
2. Use `inspect_file_as_text` to analyze the retrieved report text (via the provided link) and identify all instances of terms related to nuclear energy (e.g., "nuclear energy," "nuclear power," "nuclear fission").
3. Count the number of unique pages containing these terms, ensuring no duplicates.
4. Use `final_answer` to submit the total count of pages mentioning nuclear energy.

---

**Refinement patterns**

Provides (additional) trusted sources for search

---

**Example 4**

**Task**

Who was the director of Chinese Coffee?

---

**Unrefined plan**

Use `ask_search_agent` to search for "Who is the director of the movie Chinese Coffee? Provide the name." on IMDb (https://www.imdb.com).
Extract the director's name from the search result.
Use `final_answer` to provide the identified director's name.

---

**Refined plan**

1. Use `ask_search_agent` to search for "Who directed the movie Chinese Coffee?" and request the output to provide the director's name.
2. Analyze the detailed search results to confirm the director's identity from reliable sources (e.g., IMDb, Wikipedia).
3. Cross-check for title variations or potential ambiguities (e.g., "Chinese Coffee" vs. similar titles) in the search results.
4. Use `final_answer` to provide the confirmed director's name.

---

**Refinement patterns**

Provides (additional) trusted sources for search, intermediate cross-reference/consistency checks, clearer/more structured layout

---

### E.4.2. CODING EXAMPLES

---

**Example 5**

**Task**

Write a function to verify validity of a string of parentheses. Your code should pass the following test: `assert is_valid_parenthese("()[]")==True`

---

**Unrefined plan**

1. Implement a function using a stack to track opening brackets.
2. Create a mapping of closing brackets to their corresponding opening brackets.
3. Iterate through each character in the input string.
4. Push opening brackets onto the stack.
5. For closing brackets, check if the stack is empty or the top element does not match the expected opening bracket.
6. Return True if the stack is empty after processing all characters, otherwise return False.

---

**Refined plan**

1. Analyze the task: Create a function to validate balanced parentheses using a stack-based approach.
2. Define a mapping of closing brackets to their corresponding opening brackets.
3. Iterate through each character in the input string, pushing opening brackets onto the stack and checking for matches when encountering closing brackets.
4. Return `True` if the stack is empty after processing all characters (indicating balanced parentheses), otherwise `False`.
5. Validate the function with the test case `assert is_valid_parenthese("()[]") == True`.
6. Use the `final_answer` tool to submit the implemented code.

---

**Refinement patterns**

Explicit focus on returning the correct artifact, systematic testing and edge-case coverage

---

---

**Example 6**

**Task**

Write a function to calculate magic square. Your code should pass the following test: `assert magic_square_test([[7, 12, 1, 14], [2, 13, 8, 11], [16, 3, 10, 5], [9, 6, 15, 4]])==True`

---

**Unrefined plan**

1. Verify the input matrix is a square (all rows have the same length as the number of rows).
2. Check all elements are unique and within the range 1 to $n^2$.
3. Calculate the magic constant as the sum of the first row.
4. Validate all rows, columns, and both diagonals sum to the magic constant.
5. Return True if all checks pass, otherwise False.

---

**Refined plan**

1. Define the function `magic_square_test` that takes a matrix as input. 2. Verify the matrix is a square (all rows have the same length as the number of rows).
3. Check all elements are unique and within the range 1 to $n^2$ (where $n$ is the size of the matrix).
4. Calculate the magic constant using the formula `n * (n**2 + 1) // 2`.
5. Validate all rows, columns, and both diagonals sum to the magic constant.
6. Return `True` if all checks pass, otherwise `False`.
7. Use the `final_answer` tool to submit the implemented function as the solution.

---

**Refinement patterns**

Precise function and helper signatures, explicit focus on returning the correct artifact

---

**Example 7**

**Task**

Write a function to sort a list of elements using comb sort. Your code should pass the following tests: `assert comb_sort([5, 15, 37, 25, 79]) == [5, 15, 25, 37, 79]`

---

**Unrefined plan**

1. Create a copy of the input list to avoid modifying the original.
2. Initialize the gap as the length of the list divided by 1.3, then cast to an integer.
3. While the gap is greater than 1, iterate through the list with the current gap, comparing elements i and i + gap. If they are out of order, swap them.
4. After each pass, update the gap by dividing it by 1.3 and casting to an integer.
5. Once the gap is 1, perform a final bubble sort pass to ensure the list is fully sorted.
6. Return the sorted list.

---

**Refined plan**

1. Implement the `comb_sort` function to sort a list using the comb sort algorithm.
2. Initialize the gap as the length of the list divided by 1.3, then cast to an integer.
3. While the gap is greater than 1, iterate through the list and compare elements `i` and `i + gap`. Swap them if they are out of order.
4. After each pass, reduce the gap by dividing it by 1.3 and cast to an integer, ensuring it does not fall below 1.
5. Once the gap is 1, perform a final pass to ensure the list is fully sorted (similar to bubble sort).
6. Return the sorted list.
7. Use the `final_answer` tool to submit the implemented function as the solution.

---

**Refinement patterns**

Precise function and helper signatures, explicit focus on returning the correct artifact

---

**Example 8**

**Task**

Write a python function to find the sum of common divisors of two given numbers. Your code should pass the following test: `assert sum(10,15) == 6`

---

**Unrefined plan**

1. Compute the greatest common divisor (GCD) of the two input numbers.
2. Find all divisors of the GCD by iterating from 1 to the GCD and checking for divisibility.
3. Sum all the divisors found in step 2.
4. Return the computed sum as the result of the function.

---

**Refined plan**

1. **Understand the problem**: Create a Python function `sum(a, b)` that returns the sum of all common divisors of `a` and `b`.

2. **Compute GCD**: Use Euclidean algorithm to find the greatest common divisor (GCD) of the two numbers, as all common divisors of `a` and `b` are divisors of their GCD.

3. **Find divisors of GCD**: Efficiently find all divisors of the GCD by iterating up to its square root. For each divisor `i`, add both `i` and `gcd // i` to the sum (if they are distinct).

4. **Handle edge cases**: Ensure the function works for inputs where GCD is 0 (though inputs are positive integers as implied by the test case).

5. **Implement the function** in Python using the described logic.

6. **Verify correctness** with the provided test case `sum(10, 15) == 6`.

7. **Provide the final answer** using the `final_answer` tool.

---

**Refinement patterns**

Precise function and helper signatures, systematic testing and edge-case coverage, clearer/more structured layout

---

## E.5. Refinement and Auction Overhead

We quantify the overhead introduced by the auction phase, including refinement, along two axes: tokens consumed and wall-clock time.

**Token overhead.** On average, SALE generates 669 additional tokens per task for deep search and 1042 for coding, including strategy generation, refinement, and jury-vote tokens across all agents. By comparison, full agentic trajectories span tens of thousands to millions of tokens (Figure 2), so the auction phase represents under 1% of total token consumption.

**Wall-clock overhead.** We measure average wall time of the full auction phase, including refinement, on a stratified subset of 100 HST-BENCH tasks. Agents are deployed as vLLM servers and queried concurrently within each phase, so wall time is bounded by the slowest agent. Table 11 reports the per-task wall time under our setup. By comparison, executing the final agentic trace of the best-performing agent takes 5–10 minutes on average, so the auction phase represents approximately 1% of total wall time.

*Table 11.* Average wall-clock time of the auction phase per task, including strategy generation, jury scoring, memory-based refinement, and post-refinement jury scoring.

| GPU type | GPUs per node | Avg. wall time per task |
|----------|---------------|-------------------------|
| H100     | 8             | 3.59s                   |

# F. Significance Tests

Methods with memory are sensitive to test-set ordering, so we evaluate memory-based methods across five runs with different random permutations; baselines without memory use a single run. To assess significance, we employ two-tailed one-sample $t$-tests and bootstrap confidence intervals (CIs), which provide robustness against distributional assumptions.

We compare against five baselines: the best single agent, WTP, CARROT, TO-Router, and FrugalGPT (Tables 12 and 13). On the full test set, SALE achieves significantly higher pass@1 than all baselines across both task types, with large effect sizes (all $|t| > 5$, $p < .005$); bootstrap CIs confirm these results. Within individual bins, a few comparisons yield mixed results across the two tests, potentially reflecting reduced power in relatively smaller samples; where they diverge, at least one test typically remains significant: FrugalGPT on coding $\tau(t) \leq 0.1$ ($p = .208$ for pass@1, though $/Mt remains highly significant); best single agent and TO-Router on coding $\tau(t) \leq 60$ (borderline $p \approx .057$, with $/Mt strongly significant); and CARROT on coding $\tau(t) \leq 0.5$ ($p = .339$), where CARROT's marginal advantage over SALE is not statistically significant. For cost, SALE significantly reduces $/Mt versus the best single agent, CARROT, TO-Router, and FrugalGPT across all conditions. Compared to WTP, SALE dominates on deep search (both higher pass@1 and lower $/Mt); on coding, WTP achieves lower cost but at substantially degraded performance.

*Table 12.* One-sample $t$-test results comparing SALE against each baseline. We report $t$-statistics and $p$-values for pass@1 and $/Mt across task types and complexity bins. Dashes (—) indicate undefined tests due to zero variance.

| Task type | $\tau(t)$ | Best single agent Pass@1 $t$ | $p$ | $/Mt $t$ | $p$ | WTP Pass@1 $t$ | $p$ | $/Mt $t$ | $p$ | CARROT Pass@1 $t$ | $p$ | $/Mt $t$ | $p$ | TO-Router Pass@1 $t$ | $p$ | $/Mt $t$ | $p$ | FrugalGPT Pass@1 $t$ | $p$ | $/Mt $t$ | $p$ |
|---|---|---|---|---|---|---|---|---|---|---|---|---|---|---|---|---|---|---|---|---|---|
| *Deep search* | $\leq 0.1$ | — | — | $-37.95$ | $<.001$ | — | — | $-27.26$ | $<.001$ | — | — | $-13.90$ | $<.001$ | — | — | $-16.57$ | $<.001$ | — | — | $-67.35$ | $<.001$ |
| | $\leq 0.5$ | $4.00$ | $.016$ | $-36.88$ | $<.001$ | $8.80$ | $<.001$ | $-28.86$ | $<.001$ | $8.80$ | $<.001$ | $-15.50$ | $<.001$ | $8.80$ | $<.001$ | $-26.19$ | $<.001$ | $28.80$ | $<.001$ | $-68.95$ | $<.001$ |
| | $\leq 2.5$ | $7.76$ | $.002$ | $-35.28$ | $<.001$ | $9.80$ | $<.001$ | $-21.92$ | $<.001$ | $11.76$ | $<.001$ | $-16.57$ | $<.001$ | $9.80$ | $<.001$ | $-29.93$ | $<.001$ | $11.76$ | $<.001$ | $-80.71$ | $<.001$ |
| | $\leq 12.5$ | $4.54$ | $.011$ | $-51.31$ | $<.001$ | $3.15$ | $.035$ | $-40.62$ | $<.001$ | $8.52$ | $.001$ | $-40.62$ | $<.001$ | $4.57$ | $.010$ | $-51.31$ | $<.001$ | $7.20$ | $.002$ | $-88.73$ | $<.001$ |
| | $\leq 60$ | $6.00$ | $.004$ | $-11.09$ | $<.001$ | $10.96$ | $<.001$ | $-6.95$ | $.002$ | $10.96$ | $<.001$ | $-4.47$ | $.011$ | $6.00$ | $.004$ | $-11.09$ | $<.001$ | $6.00$ | $.004$ | $-30.95$ | $<.001$ |
| | All | $14.36$ | $<.001$ | $-47.43$ | $<.001$ | $20.12$ | $<.001$ | $-34.79$ | $<.001$ | $24.64$ | $<.001$ | $-22.14$ | $<.001$ | $17.65$ | $<.001$ | $-37.95$ | $<.001$ | $25.88$ | $<.001$ | $-94.87$ | $<.001$ |
| *Coding* | $\leq 0.1$ | $6.50$ | $.003$ | $-91.00$ | $<.001$ | $8.90$ | $<.001$ | $9.00$ | $<.001$ | $6.50$ | $.003$ | $-91.00$ | $<.001$ | $6.50$ | $.003$ | $-91.00$ | $<.001$ | $1.50$ | $.208$ | $-106.00$ | $<.001$ |
| | $\leq 0.5$ | $9.00$ | $<.001$ | $-23.52$ | $<.001$ | $23.80$ | $<.001$ | $32.61$ | $<.001$ | $-1.09$ | $.339$ | $5.88$ | $.004$ | $9.19$ | $<.001$ | $-23.52$ | $<.001$ | $49.08$ | $<.001$ | $-90.33$ | $<.001$ |
| | $\leq 2.5$ | $6.00$ | $.004$ | $-26.94$ | $<.001$ | $36.00$ | $<.001$ | $58.79$ | $<.001$ | $36.00$ | $<.001$ | $13.88$ | $<.001$ | $6.00$ | $.004$ | $-26.94$ | $<.001$ | $50.80$ | $<.001$ | $-129.01$ | $<.001$ |
| | $\leq 12.5$ | $2.98$ | $.041$ | $-6.42$ | $.003$ | $14.47$ | $<.001$ | $29.14$ | $<.001$ | $2.95$ | $.042$ | $-6.42$ | $.003$ | $2.95$ | $.042$ | $-6.42$ | $.003$ | $11.03$ | $<.001$ | $-35.10$ | $<.001$ |
| | $\leq 60$ | $2.65$ | $.057$ | $-14.70$ | $<.001$ | $15.95$ | $<.001$ | $48.58$ | $<.001$ | $3.69$ | $.021$ | $-12.66$ | $<.001$ | $2.64$ | $.058$ | $-14.70$ | $<.001$ | $12.88$ | $<.001$ | $-65.73$ | $<.001$ |
| | All | $5.90$ | $.004$ | $-38.38$ | $<.001$ | $24.02$ | $<.001$ | $63.69$ | $<.001$ | $8.74$ | $<.001$ | $-17.96$ | $<.001$ | $5.90$ | $.004$ | $-38.38$ | $<.001$ | $23.37$ | $<.001$ | $-124.11$ | $<.001$ |

*Table 13.* Bootstrap 95% confidence intervals (10,000 resamples) for the difference between SALE and each baseline. Positive pass@1 and negative $/Mt indicate SALE outperforms the baseline. CIs are constructed by resampling SALE's five runs against each baseline's single-run reference value. †Not significant (CI includes 0).

| Task type | $\tau(t)$ | Best single agent Pass@1 | $/Mt | WTP Pass@1 | $/Mt | CARROT Pass@1 | $/Mt | TO-Router Pass@1 | $/Mt | FrugalGPT Pass@1 | $/Mt |
|---|---|---|---|---|---|---|---|---|---|---|---|
| *Deep search* | $\leq 0.1$ | $[3.75, 3.75]$ | $[-0.15, -0.14]$ | $[7.45, 7.45]$ | $[-0.11, -0.10]$ | $[6.25, 6.25]$ | $[-0.06, -0.05]$ | $[4.95, 4.95]$ | $[-0.07, -0.06]$ | $[4.95, 4.95]$ | $[-0.26, -0.25]$ |
| | $\leq 0.5$ | $[0.50, 1.25]$ | $[-0.14, -0.13]$ | $[1.70, 2.45]$ | $[-0.11, -0.10]$ | $[1.70, 2.45]$ | $[-0.06, -0.05]$ | $[1.70, 2.45]$ | $[-0.10, -0.09]$ | $[6.70, 7.45]$ | $[-0.26, -0.25]$ |
| | $\leq 2.5$ | $[3.75, 5.75]$ | $[-0.14, -0.13]$ | $[5.00, 7.00]$ | $[-0.09, -0.08]$ | $[6.20, 8.20]$ | $[-0.07, -0.06]$ | $[5.00, 7.00]$ | $[-0.12, -0.11]$ | $[6.20, 8.20]$ | $[-0.31, -0.30]$ |
| | $\leq 12.5$ | $[2.68, 5.61]$ | $[-0.20, -0.19]$ | $[1.41, 4.34]$ | $[-0.16, -0.15]$ | $[6.31, 9.24]$ | $[-0.16, -0.15]$ | $[2.71, 5.64]$ | $[-0.20, -0.19]$ | $[5.11, 8.04]$ | $[-0.34, -0.33]$ |
| | $\leq 60$ | $[3.13, 5.00]$ | $[-0.16, -0.12]$ | $[6.23, 8.10]$ | $[-0.11, -0.07]$ | $[6.23, 8.10]$ | $[-0.08, -0.04]$ | $[3.13, 5.00]$ | $[-0.16, -0.12]$ | $[3.13, 5.00]$ | $[-0.40, -0.36]$ |
| | All | $[3.04, 3.83]$ | $[-0.16, -0.14]$ | $[4.44, 5.23]$ | $[-0.12, -0.10]$ | $[5.54, 6.33]$ | $[-0.08, -0.06]$ | $[3.84, 4.63]$ | $[-0.13, -0.11]$ | $[5.84, 6.63]$ | $[-0.31, -0.29]$ |
| *Coding* | $\leq 0.1$ | $[2.25, 3.75]$ | $[-0.19, -0.18]$ | $[3.45, 4.95]$ | $[0.01, 0.02]$ | $[2.25, 3.75]$ | $[-0.19, -0.18]$ | $[2.25, 3.75]$ | $[-0.19, -0.18]$ | $[-0.25, 1.25]^{\dagger}$ | $[-0.22, -0.21]$ |
| | $\leq 0.5$ | $[1.77, 2.53]$ | $[-0.09, -0.08]$ | $[5.52, 6.28]$ | $[0.12, 0.13]$ | $[-0.78, -0.02]$ | $[0.02, 0.03]$ | $[1.82, 2.58]$ | $[-0.09, -0.08]$ | $[11.92, 12.68]$ | $[-0.34, -0.33]$ |
| | $\leq 2.5$ | $[1.25, 2.00]$ | $[-0.07, -0.06]$ | $[8.75, 9.50]$ | $[0.14, 0.15]$ | $[8.75, 9.50]$ | $[0.03, 0.04]$ | $[1.25, 2.00]$ | $[-0.07, -0.06]$ | $[12.45, 13.20]$ | $[-0.32, -0.31]$ |
| | $\leq 12.5$ | $[1.48, 4.94]$ | $[-0.07, -0.04]$ | $[13.84, 17.30]$ | $[0.24, 0.27]$ | $[1.44, 4.90]$ | $[-0.07, -0.04]$ | $[1.44, 4.90]$ | $[-0.07, -0.04]$ | $[10.14, 13.60]$ | $[-0.32, -0.29]$ |
| | $\leq 60$ | $[1.27, 5.32]$ | $[-0.08, -0.06]$ | $[17.75, 21.80]$ | $[0.23, 0.25]$ | $[2.55, 6.60]$ | $[-0.07, -0.05]$ | $[1.25, 5.30]$ | $[-0.08, -0.06]$ | $[13.95, 18.00]$ | $[-0.33, -0.31]$ |
| | All | $[1.95, 3.55]$ | $[-0.10, -0.09]$ | $[10.25, 11.85]$ | $[0.15, 0.16]$ | $[3.25, 4.85]$ | $[-0.05, -0.04]$ | $[1.95, 3.55]$ | $[-0.10, -0.09]$ | $[9.95, 11.55]$ | $[-0.31, -0.30]$ |

# G. Baselines

## G.1. Baseline Implementation Details

We provide implementation details and hyperparameter configurations for the baseline routing methods evaluated in Section 6.

**WTP router.** The WTP router predicts model performance using a $K$-Nearest Neighbors classifier, then selects models based on a performance–cost trade-off controlled by the willingness-to-pay parameter. Following Hu et al. (2024), we use $k = 50$ neighbors with cosine distance over sentence embeddings. We tune the willingness-to-pay parameter on the deep search and coding training sets. Table 14 presents the hyperparameter configuration.

*Table 14.* WTP router hyperparameters.

| Hyperparameter | Value |
| --- | --- |
| $k$ (neighbors) | 50 |
| Embedding model | `all-MiniLM-L6-v2` |
| Distance metric | Cosine |
| Willingness-to-pay (deep search) | 5.0 |
| Willingness-to-pay (coding) | 5.0 |

**CARROT router.** CARROT (Somerstep et al., 2025) fine-tunes a RoBERTa encoder as a multi-label classifier to predict the probability of success for each candidate model given an input query. At inference time, routing decisions are made by selecting the model that maximizes a cost-performance trade-off score: $(1 - \mu) \cdot p_i - \mu \cdot c_i$, where $p_i$ is the predicted success probability and $c_i$ is the normalized cost of model $i$. The hyperparameter $\mu \in [0, 1]$ controls the trade-off between performance ($\mu = 0$) and cost ($\mu = 1$). Due to the limited size of our training set, we freeze the RoBERTa backbone and only train the classification head, which we found to yield more stable predictions. We set $\mu = 0.0$ based on development set performance; higher values of $\mu$ caused the router to route almost exclusively to the smallest agent, degrading task performance without meaningful cost savings. Table 15 presents the hyperparameter configuration.

*Table 15.* CARROT hyperparameters.

| Hyperparameter | Value |
| --- | --- |
| *Performance Predictor* | |
| Base model | `roberta-base` |
| Backbone | Frozen |
| Optimizer | AdamW |
| Learning rate | $1 \times 10^{-3}$ |
| Weight decay | $1 \times 10^{-2}$ |
| Batch size | 16 |
| Epochs | 10 |
| *Routing* | |
| Cost-performance trade-off ($\mu$) | 0.0 |

**TO-Router.** The TO-Router baseline fine-tunes a BERT encoder to predict model performance scores, using soft labels and inverse class frequency weighting to handle imbalanced performance distributions. We follow the implementation of Stripelis et al. (2024). Table 16 presents the hyperparameter configuration.

*Table 16.* TO-Router hyperparameters.

| Hyperparameter | Value |
|---|---|
| Base model | `bert-base-uncased` |
| Optimizer | AdamW |
| Learning rate | $5 \times 10^{-5}$ |
| Weight decay | $1 \times 10^{-4}$ |
| Batch size | 8 |
| Epochs | 5 |
| Soft label temperature | 10 |
| Sample weighting | Inverse class frequency |
| Loss function | Cross-entropy |

**FrugalGPT.** For the FrugalGPT baseline, we follow the implementation of Chen et al. (2024), which consists of two components: (i) a scoring function that predicts answer correctness, and (ii) a cascade optimizer that determines the optimal model ordering and acceptance thresholds. Table 17 presents the hyperparameter configuration.

*Table 17.* FrugalGPT hyperparameters.

| Hyperparameter | Value |
|---|---|
| *Scoring Function* | |
| Base model | `distilbert-base-uncased` |
| Optimizer | AdamW |
| Learning rate | $2 \times 10^{-5}$ |
| Weight decay | $1 \times 10^{-2}$ |
| Warmup steps | 500 |
| Batch size | 8 |
| Epochs | 8 |
| Gradient clipping | 1.0 |
| *Cascade Optimization* | |
| Max cascade length | 3 |
| Quantile grid steps | 20 |
| Optimizer | `scipy.brute` + Nelder-Mead |

### G.2. Pareto Frontier vs. Baseline Routers

Figure 7 visualizes the accuracy–cost trade-offs reported in Section 6 (Table 1). On deep search tasks, SALE shifts the Pareto frontier outward for every value of $\tau$. On coding, SALE improves the frontier in all bins except $\tau \leq 0.5$, where CARROT attains a comparable trade-off. We also note that WTP can achieve lower cost on coding tasks, but only with substantial accuracy degradation.

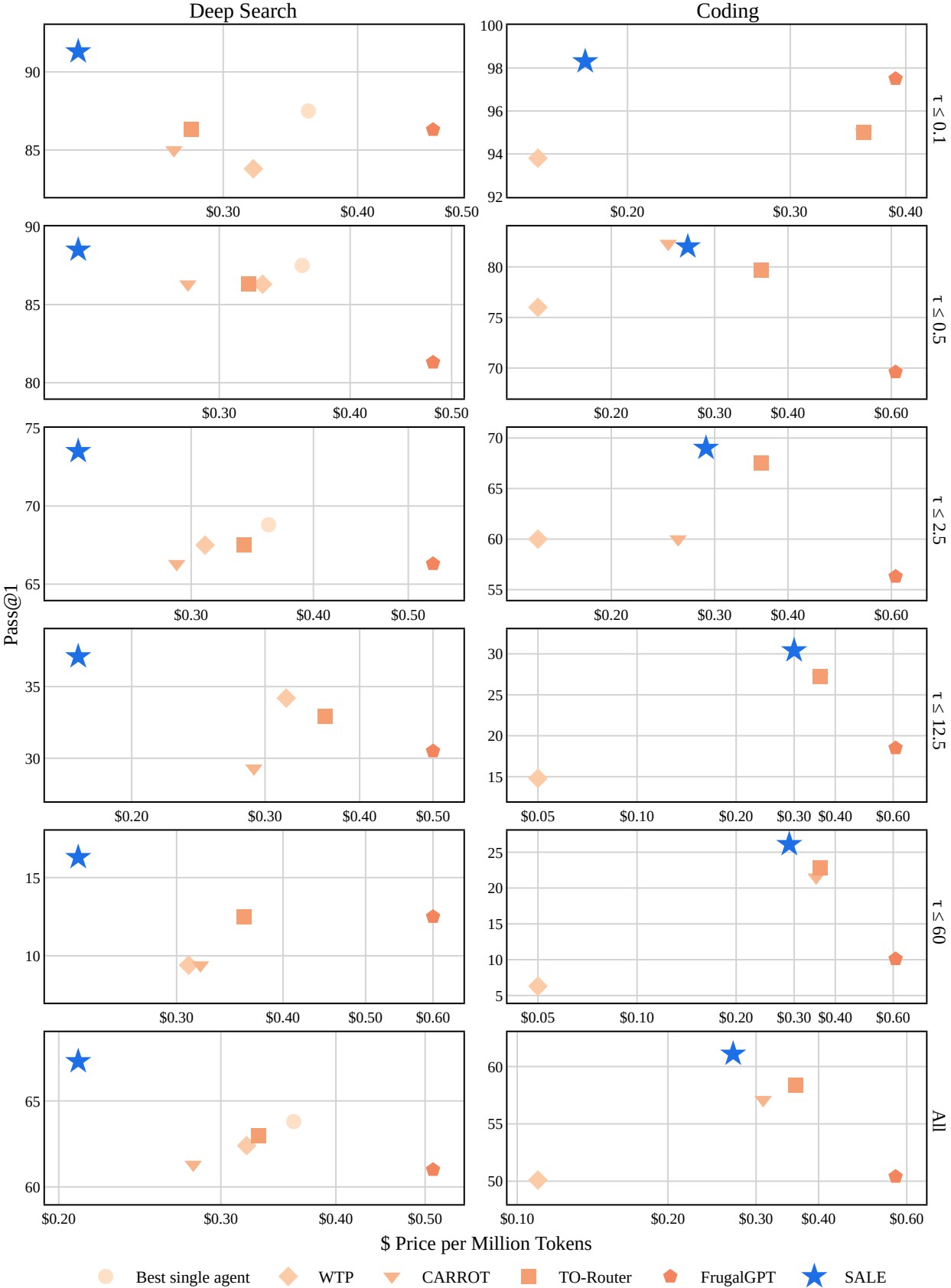

*Figure 7.* SALE vs. baselines on the accuracy–cost trade-off. The upper-left envelope is Pareto-optimal.

# H. Oracle Results and Agent Selection Analysis

## H.1. Oracle Router

Table 18 reports the oracle router results on HST-BENCH, as an upper bound on what is achievable from per-task model selection alone. For each task $t$ we run all candidate agents and select the smallest agent that produces a correct final answer. If all answers are incorrect, we select the smallest agent in the pool. This yields the minimum possible spend subject to achieving correctness whenever any agent in the pool can solve the task. We note, however, that in complexity bins where no agent solves a large fraction of instances, the oracle's effective cost can be artificially low: on those unsolved tasks the choice of agent does not affect accuracy, and our definition defaults to the cheapest model, thereby minimizing expenditure in a regime where additional compute would not change outcomes. As a result, oracle cost in low-accuracy bins should be interpreted cautiously, primarily as a lower bound on spend rather than a representative operating point.

Relative to this hindsight upper bound, SALE captures a substantial fraction of the available routing gains on easier tasks, but the gap to oracle performance grows with task complexity. For deep search, at $\tau(t) \leq 0.1$ the oracle attains 97.5 pass@1 versus 91.3 for SALE (Table 1), while for coding the gap is only 0.5 points (98.8 vs. 98.3). At higher complexity, the oracle–SALE gap widens (deep search $\tau(t) \leq 60$: 25.0 vs. 16.3; coding $\tau(t) \leq 60$: 34.2 vs. 26.1). We also observe a growing gap between the oracle and the best single agent as task complexity increases, peaking in the medium–high complexity bins. This implies increased opportunity for routing in these instances.

While a gap between SALE and the oracle remains, SALE consistently exhibits the smallest accuracy gap to the oracle among all methods. On deep search, SALE's gap to the oracle is 10.6 points (67.3 vs. 77.9), compared to 14.1–16.9 points for the baselines (WTP: 15.5; CARROT: 16.6; TO-Router: 14.9; FrugalGPT: 16.9; best single agent: 14.1). On coding, SALE's accuracy gap is 7.3 points (61.1 vs. 68.4), whereas baselines range from 10.0 to 18.3 points behind the oracle (WTP: 18.3; FrugalGPT: 18.0; CARROT: 11.3; TO-Router and best single agent: 10.0). In cost, SALE also closes more of the gap on deep search (0.21 vs. oracle 0.07) than any baseline (0.28–0.51); on coding, WTP achieves lower cost ($0.11) but at the expense of a substantially larger accuracy gap (18.3 points vs. SALE's 7.3 points). In short, the oracle–SALE gap should be interpreted relative to the considerably wider oracle–baseline gaps: SALE captures more of the available routing gains than any existing method, and the residual headroom reflects the inherent difficulty of task-complexity prediction rather than a limitation unique to our approach.

*Table 18.* Oracle-router performance (pass@1) and price per million tokens ($/Mt) across task complexity bins for deep search and coding. The oracle selects, for each task, the smallest agent that returns a correct final answer; if no agent is correct, it defaults to the smallest agent.

| Task type | $\tau(t)$ | Oracle router | |
|---|---|---|---|
| | | Pass@1($\uparrow$) | $/Mt($\downarrow$) |
| *Deep search* | $\leq 0.1$ | 97.5 | 0.06 |
| | $\leq 0.5$ | 93.8 | 0.06 |
| | $\leq 2.5$ | 86.3 | 0.08 |
| | $\leq 12.5$ | 56.1 | 0.08 |
| | $\leq 60$ | 25.0 | 0.09 |
| | All | 77.9 | 0.07 |
| *Coding* | $\leq 0.1$ | 98.8 | 0.05 |
| | $\leq 0.5$ | 89.9 | 0.08 |
| | $\leq 2.5$ | 76.3 | 0.08 |
| | $\leq 12.5$ | 43.2 | 0.08 |
| | $\leq 60$ | 34.2 | 0.09 |
| | All | 68.4 | 0.08 |

## H.2. Routing Diagnostics

To better understand SALE's routing behavior and identify opportunities for improvement, we conduct a systematic error analysis comparing SALE's agent selections over a single run against the oracle router described in Appendix H.1. We categorize each routing decision into one of four outcomes:

1. *Correct:* SALE selects the same agent as the oracle.

2. *Over-escalation:* SALE selects a larger (more expensive) agent than necessary.

3. *Under-escalation*: SALE selects a smaller agent that fails when a larger one would succeed.

4. *Unavoidable:* No agent in the pool produces a correct answer.

**Deep search diagnostics.** Figure 8 presents row-normalized confusion matrices comparing SALE's agent selections against the oracle router described in Appendix H.1, both overall and stratified by complexity bin. Each cell shows the percentage of tasks where SALE selected a given agent (row) and the oracle selected another (column), annotated as: correct ($\checkmark$, diagonal), over-escalation ($\uparrow$, SALE selects larger than necessary), under-escalation ($\downarrow$, SALE selects an agent that fails when a larger one succeeds), or unavoidable (—, no agent succeeds).

In the aggregate matrix ($n = 354$), the dominant off-diagonal mass lies in the leftmost column, indicating frequent over-escalation: when SALE selects the 14B or 32B agents, 65.1% and 61.0% of those cases could have been handled by the 4B agent. Under-escalation ($\downarrow$) appears infrequently across all rows, confirming that SALE rarely sacrifices accuracy by selecting an insufficiently capable agent. Notably, when SALE does route to the 4B agent, it achieves the highest diagonal accuracy—54.0% overall, and even higher within individual bins (100% at $\tau \leq 0.1$, 75.0% at $\tau \leq 2.5$)—suggesting the system correctly identifies 4B-suitable tasks but triggers such routing too conservatively.

Across complexity bins, over-escalation decreases systematically: at $\tau \leq 0.1$, over-escalation from 32B selections to the 4B oracle accounts for 86.2%, dropping to 16.7% at $\tau \leq 12.5$ and near-zero at $\tau \leq 60$. This shift reflects a structural change in the error landscape: as complexity grows, the "none" column increasingly dominates (reaching 73–100% at $\tau \leq 60$), indicating that most failures become unavoidable regardless of routing.

**Coding diagnostics.** Figure 9 shows row-normalized confusion matrices for coding-agent selections. Mirroring the deep-search pattern, over-escalation dominates the aggregate matrix ($n = 399$): when SALE selects 14B or 32B, the oracle often prefers smaller agents: 55.2% of 14B selections could have been handled by 4B, and 31.5% of 32B selections map to 4B (with an additional 25.0% mapping to 8B). Under-escalation remains similarly limited: SALE's 4B selections are predominantly correct (69.6% diagonal), with most remaining errors unavoidable (21.7%). This asymmetry where SALE prioritizes accuracy via conservative routing at the cost of compute closely matches the deep-search behavior.

Stratifying by complexity reveals a parallel transition. At $\tau \leq 0.1$, small-agent routing is perfect (4B achieves 100%), but over-escalation is extreme: all 14B and 32B selections could have been routed to 4B (100% in both rows). As complexity increases ($\tau \leq 0.5$, $\tau \leq 2.5$), over-escalation persists while the "none" column grows. However, the hardest bin ($\tau \leq 60$) diverges from deep search: rather than collapsing to near-total unavoidable outcomes, coding shows more heterogeneous behavior, i.e., the "none" column reaches 50–78% across rows, but under-escalation also emerges (e.g., 4B shows 25% under-escalation to 8B). This indicates that high-complexity coding tasks retain meaningful variation in required agent capability, whereas deep-search tasks become predominantly unsolvable.

Overall, while SALE already achieves the smallest oracle gap among all evaluated routers (Section 6), the error analysis reveals that further improvements require further reducing over-escalation on low-complexity tasks where the oracle favors 4B yet SALE triggers larger agents. Importantly, this failure mode is conservative: SALE errs toward preserving accuracy rather than sacrificing correctness, and under-escalation remains rare across both domains. Moreover, when SALE does route to 4B, diagonal accuracy is high (54.0% for deep search, 69.6% for coding), suggesting the system can correctly identify easy tasks but currently triggers such routing too infrequently.

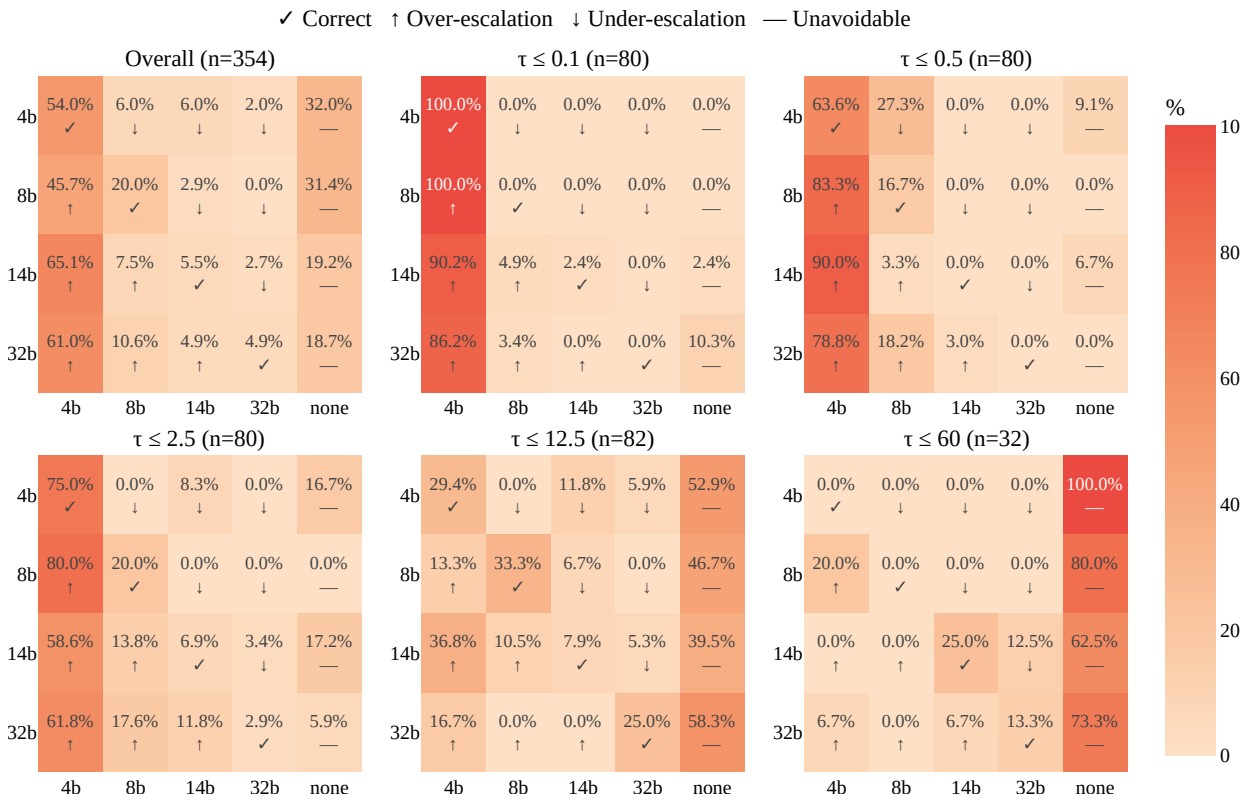

*Figure 8.* Deep search routing confusion matrices (SALE vs. oracle). Cells show row-normalized percentages.

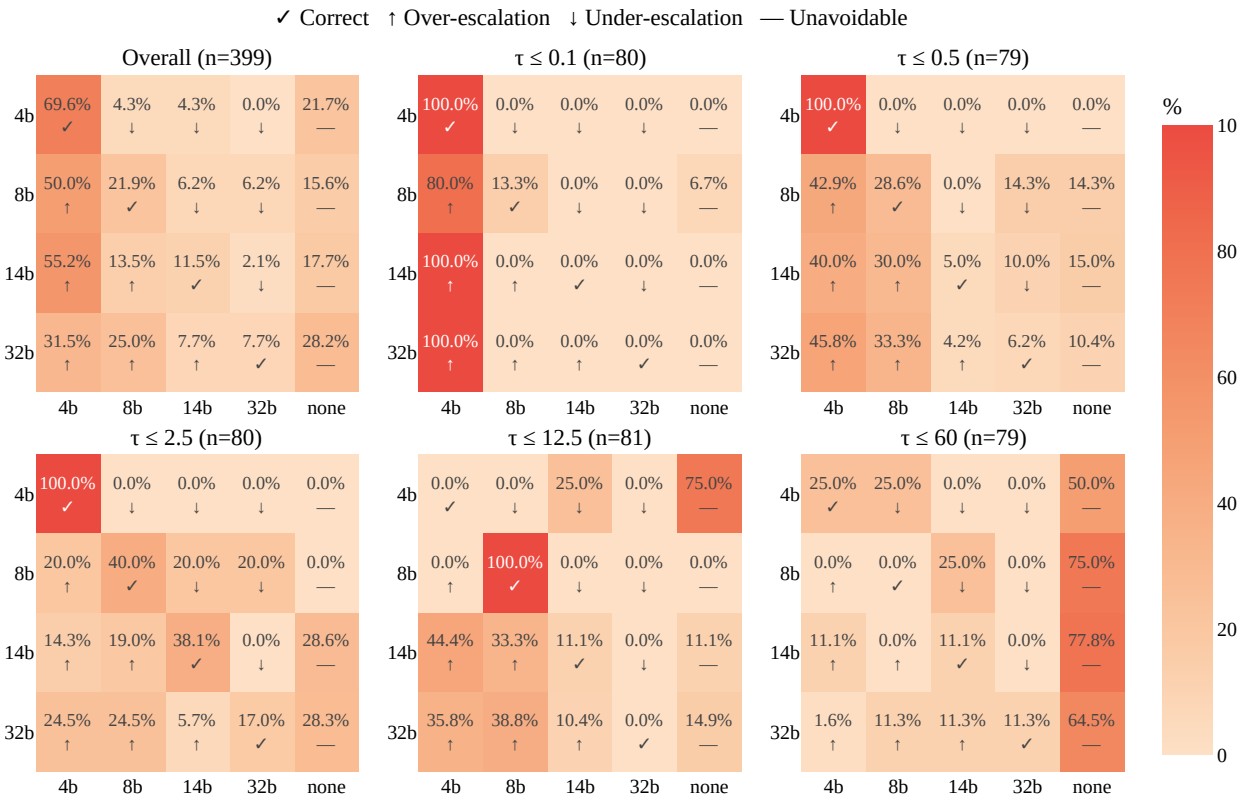

*Figure 9.* Coding routing confusion matrices (SALE vs. oracle). Cells show row-normalized percentages.

# I. Further Ablations

## I.1. Cost–Value Function Ablations

Table 19 reports the results of ablating individual terms in the cost–value function. We separately remove the price per million tokens $\pi(a_i)$, the strategy length $|s_{t,i}|$, the normalized entropy $H(s_{t,i})$, and the entire jury term $\sum w_j \cdot \gamma(s_{t,i})$. For each ablated configuration, we re-optimize the remaining weights to best fit that specific setup.

Across both deep search and coding tasks, ablating any of the four terms leads to lower pass@1 scores, indicating that each component contributes meaningfully to SALE's overall performance. For deep search, all ablations also result in higher average cost per million tokens, suggesting that the full cost–value function more effectively balances accuracy and efficiency. In contrast, for coding tasks, ablating $\pi(a_i)$ and $H(s_{t,i})$ yields slightly lower costs, reflecting an increased selection of smaller models, though this comes at the expense of accuracy. This pattern is consistent with the findings in Section 7.1, where we show that SALE already favors larger models for coding tasks even in its full form, whereas model selection is more varied for deep search.

The impact of each ablation also differs by task type and complexity. Removing the jury assessment causes the largest performance drop for deep search ($-3.6$ pass@1 on average), with the gap widening substantially on more complex tasks ($-7.6$ at $\tau \leq 12.5$; $-6.3$ at $\tau \leq 60$), underscoring the importance of jury-based scoring in challenging settings. For coding, SALE is more robust to jury removal; instead, ablating any of the cost terms (price and strategy length) results in the steepest decline, particularly at mid-complexity levels ($-4.2$ and $-5.5$, respectively, at $\tau \leq 2.5$), reflecting their role in routing beyond cost alone. Entropy ablation presents a nuanced trade-off: at $\tau \leq 0.1$ for deep search, it achieves the lowest cost but sacrifices accuracy, suggesting entropy helps prevent under-spending on deceptively simple tasks. We note that a few ablated configurations marginally outperform SALE in isolated bins, though these gains are offset by larger losses in other bins.

*Table 19.* Deep search and coding pass@1 and price per million tokens ($\$/Mt$) across task-complexity bins for SALE in its full form and under four ablation conditions. We remove the price term $\pi(a_i)$ (w/o price), the strategy length $|s_{t,i}|$ (w/o length), the entropy term $H(s_{t,i})$ (w/o entropy), and the jury scores $\sum w_j \cdot \gamma(s_{t,i})$ (w/o jury).

| Task type | $\tau(t)$ | w/o price | | w/o length | | w/o entropy | | w/o jury | |
|---|---|---|---|---|---|---|---|---|---|
| | | Pass@1($\uparrow$) | $\$/Mt$($\downarrow$) | Pass@1($\uparrow$) | $\$/Mt$($\downarrow$) | Pass@1($\uparrow$) | $\$/Mt$($\downarrow$) | Pass@1($\uparrow$) | $\$/Mt$($\downarrow$) |
| *Deep search* | $\leq 0.1$ | $89.8_{0.5}$ | $0.22_{0.01}$ | $91.0_{0.5}$ | $0.28_{0.00}$ | $89.8_{1.5}$ | $0.19_{0.00}$ | $86.3_{0.0}$ | $0.28_{0.00}$ |
| | $\leq 0.5$ | $88.5_{0.9}$ | $0.26_{0.01}$ | $88.8_{1.0}$ | $0.31_{0.01}$ | $85.5_{1.7}$ | $0.25_{0.00}$ | $89.0_{1.7}$ | $0.23_{0.01}$ |
| | $\leq 2.5$ | $69.7_{1.5}$ | $0.25_{0.01}$ | $71.8_{1.7}$ | $0.30_{0.01}$ | $73.0_{1.9}$ | $0.26_{0.01}$ | $72.2_{0.9}$ | $0.21_{0.00}$ |
| | $\leq 12.5$ | $33.7_{2.3}$ | $0.19_{0.01}$ | $33.4_{1.8}$ | $0.26_{0.01}$ | $35.1_{0.9}$ | $0.22_{0.01}$ | $29.5_{0.5}$ | $0.19_{0.00}$ |
| | $\leq 60$ | $16.9_{1.5}$ | $0.25_{0.01}$ | $15.6_{2.0}$ | $0.29_{0.01}$ | $14.4_{1.5}$ | $0.25_{0.01}$ | $10.0_{2.3}$ | $0.20_{0.01}$ |
| | All | $65.4_{0.7}$ | $0.23_{0.00}$ | $66.0_{0.6}$ | $0.29_{0.00}$ | $65.5_{0.7}$ | $0.23_{0.00}$ | $63.7_{0.5}$ | $0.23_{0.00}$ |
| *Coding* | $\leq 0.1$ | $98.3_{1.0}$ | $0.18_{0.00}$ | $98.8_{0.0}$ | $0.28_{0.00}$ | $97.2_{0.5}$ | $0.17_{0.00}$ | $96.8_{0.6}$ | $0.17_{0.00}$ |
| | $\leq 0.5$ | $79.2_{1.5}$ | $0.28_{0.00}$ | $78.7_{2.0}$ | $0.29_{0.00}$ | $78.0_{2.2}$ | $0.28_{0.01}$ | $79.2_{0.6}$ | $0.30_{0.00}$ |
| | $\leq 2.5$ | $64.8_{2.2}$ | $0.28_{0.01}$ | $63.5_{1.2}$ | $0.28_{0.00}$ | $64.8_{2.5}$ | $0.25_{0.00}$ | $67.5_{1.9}$ | $0.28_{0.00}$ |
| | $\leq 12.5$ | $30.1_{2.4}$ | $0.27_{0.00}$ | $28.1_{2.4}$ | $0.29_{0.01}$ | $29.9_{2.0}$ | $0.29_{0.00}$ | $29.1_{1.0}$ | $0.29_{0.00}$ |
| | $\leq 60$ | $23.8_{0.5}$ | $0.26_{0.01}$ | $24.8_{1.6}$ | $0.28_{0.01}$ | $25.6_{3.0}$ | $0.28_{0.00}$ | $27.3_{0.6}$ | $0.29_{0.00}$ |
| | All | $59.2_{0.8}$ | $0.26_{0.00}$ | $58.7_{0.7}$ | $0.28_{0.00}$ | $59.1_{1.0}$ | $0.26_{0.00}$ | $59.9_{0.5}$ | $0.27_{0.00}$ |

## I.2. Jury Ablations

Table 20 reports results when the full jury (comprising the 4B, 8B, 14B, and 32B agents) is replaced by a single judge. Across both task types, all single-judge configurations underperform the full jury: for deep search, overall Pass@1 drops from 67.3 to 65.0–66.4, while for coding it falls from 61.1 to 59.0–60.5. Notably, no single judge is consistently superior—the 8B achieves the highest deep search accuracy (66.4), while the 14B leads on coding (60.5)—and larger judges do not reliably outperform smaller ones. Single judges also incur substantially higher costs for deep search (0.25–0.32 vs. 0.21 $/Mt), suggesting the jury enables more efficient model selection. These results indicate that the jury's strength lies not in any individual member but in the diversity of perspectives: combining judges of varying capacity produces a regularizing effect that yields more robust and cost-effective decisions than any single judge alone. Crucially, this ensemble incurs negligible overhead: each judge produces only a single token (a 1–5 discrete score) and we reuse the same agents already loaded in memory as part of the pool $\mathcal{A}$. This makes the jury's benefits effectively free.

Given that individual judges underperform the full jury, we further ask whether each member is necessary by removing one judge at a time (Table 21). Indeed, every ablation leads to a decline in pass@1, confirming that each judge contributes unique signal. For coding, removing the 4B judge causes the largest drop ($-2.9$ pass@1), despite the 4B not being the strongest individual judge, highlighting its complementary role within the ensemble. For deep search, removing the 14B has the smallest impact (66.9 vs. 67.3), though performance still decreases. Interestingly, removing the 4B also substantially increases deep search cost (0.28 vs. 0.21 $/Mt), suggesting the smaller judge helps steer the system toward more economical selections. Overall, these results reinforce that jury diversity is not redundant: even the smallest judge provides information that improves both accuracy and efficiency.

Finally, we ablate agent self-judgment (retaining only peer-judgment) and peer-judgment (retaining only self-judgment) from SALE (Table 22). Without self-judgment, the system approaches and occasionally slightly exceeds the best single agent baseline (64.1 vs. 63.8 for deep search; 58.8 vs. 58.4 for coding), though it remains below the full SALE (67.3 and 61.1, respectively). In contrast, removing peer-judgment causes a larger drop, especially for coding, where overall pass@1 falls to 48.7. Notably, this configuration yields markedly lower costs (0.07 $/Mt for deep search vs. 0.21), suggesting that smaller agents tend to be more confident self-scorers, leading the system to favor them when peer signals are unavailable. At low task complexity, relying solely on self-judgment still matches or even slightly exceeds the best single agent (e.g., 97.5 vs. 95.0 at $\tau \leq 0.1$ for coding); yet as complexity increases, performance degrades sharply (5.6 vs. 12.5 at $\tau \leq 60$ for deep search), indicating that self-judgment alone does not scale to harder tasks. These results underscore the complementary nature of both feedback types: peer-judgment provides the external calibration necessary for difficult problems, while self-judgment contributes efficient, low-cost signal on simpler ones.

*Table 20.* Deep search and coding pass@1 and price per million tokens ($/Mt) across task-complexity bins for SALE with the agent jury replaced by one single agent judge.

| Task type | $\tau(t)$ | 4B judge only | | 8B judge only | | 14B judge only | | 32B judge only | |
|---|---|---|---|---|---|---|---|---|---|
| | | Pass@1($\uparrow$) | $/Mt($\downarrow$) | Pass@1($\uparrow$) | $/Mt($\downarrow$) | Pass@1($\uparrow$) | $/Mt($\downarrow$) | Pass@1($\uparrow$) | $/Mt($\downarrow$) |
| *Deep search* | $\leq 0.1$ | $91.3_{0.0}$ | $0.25_{0.01}$ | $90.0_{0.0}$ | $0.30_{0.00}$ | $89.0_{0.5}$ | $0.29_{0.00}$ | $90.0_{0.0}$ | $0.28_{0.00}$ |
| | $\leq 0.5$ | $88.0_{0.6}$ | $0.28_{0.01}$ | $88.8_{0.0}$ | $0.30_{0.01}$ | $86.3_{0.6}$ | $0.33_{0.01}$ | $88.0_{0.6}$ | $0.32_{0.01}$ |
| | $\leq 2.5$ | $71.0_{1.5}$ | $0.26_{0.01}$ | $72.5_{0.8}$ | $0.34_{0.01}$ | $69.2_{0.6}$ | $0.35_{0.01}$ | $69.2_{1.0}$ | $0.32_{0.01}$ |
| | $\leq 12.5$ | $33.4_{0.6}$ | $0.22_{0.01}$ | $35.4_{0.0}$ | $0.33_{0.00}$ | $36.1_{1.0}$ | $0.31_{0.01}$ | $34.1_{2.2}$ | $0.31_{0.01}$ |
| | $\leq 60$ | $15.6_{0.0}$ | $0.25_{0.01}$ | $15.6_{0.0}$ | $0.31_{0.01}$ | $15.6_{0.0}$ | $0.31_{0.01}$ | $14.4_{1.5}$ | $0.29_{0.01}$ |
| | All | $65.7_{0.4}$ | $0.25_{0.00}$ | $66.4_{0.2}$ | $0.32_{0.00}$ | $65.0_{0.3}$ | $0.32_{0.00}$ | $65.1_{0.6}$ | $0.31_{0.00}$ |
| *Coding* | $\leq 0.1$ | $96.8_{0.6}$ | $0.18_{0.00}$ | $97.5_{0.0}$ | $0.18_{0.00}$ | $98.5_{0.5}$ | $0.20_{0.01}$ | $97.8_{0.5}$ | $0.19_{0.00}$ |
| | $\leq 0.5$ | $80.3_{0.6}$ | $0.29_{0.00}$ | $80.3_{1.0}$ | $0.28_{0.00}$ | $80.5_{0.6}$ | $0.30_{0.00}$ | $81.5_{0.6}$ | $0.29_{0.00}$ |
| | $\leq 2.5$ | $66.8_{1.0}$ | $0.29_{0.00}$ | $67.0_{2.8}$ | $0.28_{0.00}$ | $68.2_{1.0}$ | $0.29_{0.00}$ | $66.5_{1.5}$ | $0.29_{0.00}$ |
| | $\leq 12.5$ | $27.2_{1.4}$ | $0.27_{0.00}$ | $29.1_{1.7}$ | $0.30_{0.01}$ | $29.4_{1.2}$ | $0.31_{0.00}$ | $28.4_{1.6}$ | $0.30_{0.02}$ |
| | $\leq 60$ | $24.3_{0.9}$ | $0.28_{0.00}$ | $25.3_{1.6}$ | $0.28_{0.01}$ | $26.1_{1.7}$ | $0.29_{0.01}$ | $25.8_{2.2}$ | $0.29_{0.00}$ |
| | All | $59.0_{0.4}$ | $0.26_{0.00}$ | $59.8_{0.8}$ | $0.26_{0.00}$ | $60.5_{0.5}$ | $0.28_{0.00}$ | $60.0_{0.6}$ | $0.27_{0.00}$ |

*Table 21.* Deep search and coding pass@1 and price per million tokens ($/Mt) across task-complexity bins for SALE with one judge removed from the jury in turn.

| Task type | $\tau(t)$ | w/o 4B judge | | w/o 8B judge | | w/o 14B judge | | w/o 32B judge | |
|---|---|---|---|---|---|---|---|---|---|
| | | Pass@1($\uparrow$) | $/Mt($\downarrow$) | Pass@1($\uparrow$) | $/Mt($\downarrow$) | Pass@1($\uparrow$) | $/Mt($\downarrow$) | Pass@1($\uparrow$) | $/Mt($\downarrow$) |
| *Deep search* | $\leq 0.1$ | $89.2_{0.6}$ | $0.20_{0.00}$ | $92.3_{0.5}$ | $0.23_{0.00}$ | $91.8_{1.0}$ | $0.22_{0.01}$ | $90.0_{0.0}$ | $0.22_{0.01}$ |
| | $\leq 0.5$ | $86.0_{0.9}$ | $0.28_{0.00}$ | $86.0_{0.9}$ | $0.25_{0.01}$ | $88.8_{1.1}$ | $0.22_{0.01}$ | $87.0_{1.0}$ | $0.24_{0.01}$ |
| | $\leq 2.5$ | $71.2_{0.8}$ | $0.35_{0.00}$ | $68.2_{1.5}$ | $0.24_{0.01}$ | $73.0_{2.0}$ | $0.23_{0.01}$ | $72.5_{1.1}$ | $0.23_{0.01}$ |
| | $\leq 12.5$ | $36.1_{1.0}$ | $0.31_{0.01}$ | $34.9_{1.8}$ | $0.18_{0.01}$ | $36.3_{1.8}$ | $0.18_{0.00}$ | $33.9_{2.4}$ | $0.17_{0.00}$ |
| | $\leq 60$ | $14.4_{1.5}$ | $0.28_{0.01}$ | $16.9_{1.5}$ | $0.23_{0.01}$ | $13.8_{1.7}$ | $0.23_{0.02}$ | $16.3_{1.3}$ | $0.23_{0.01}$ |
| | All | $65.3_{0.4}$ | $0.28_{0.00}$ | $65.3_{0.6}$ | $0.23_{0.00}$ | $66.9_{0.7}$ | $0.21_{0.00}$ | $65.7_{0.7}$ | $0.22_{0.00}$ |
| *Coding* | $\leq 0.1$ | $97.0_{0.6}$ | $0.16_{0.00}$ | $97.8_{1.2}$ | $0.18_{0.00}$ | $97.0_{0.6}$ | $0.18_{0.00}$ | $97.8_{1.2}$ | $0.18_{0.00}$ |
| | $\leq 0.5$ | $75.7_{2.6}$ | $0.25_{0.01}$ | $79.0_{1.3}$ | $0.27_{0.00}$ | $79.5_{1.5}$ | $0.27_{0.01}$ | $77.7_{1.5}$ | $0.27_{0.01}$ |
| | $\leq 2.5$ | $62.8_{1.8}$ | $0.22_{0.01}$ | $65.8_{1.3}$ | $0.28_{0.00}$ | $65.0_{2.2}$ | $0.26_{0.00}$ | $66.0_{2.2}$ | $0.27_{0.00}$ |
| | $\leq 12.5$ | $29.9_{0.9}$ | $0.30_{0.00}$ | $29.4_{1.4}$ | $0.31_{0.01}$ | $29.6_{1.6}$ | $0.30_{0.01}$ | $28.9_{1.7}$ | $0.30_{0.00}$ |
| | $\leq 60$ | $25.8_{1.5}$ | $0.28_{0.00}$ | $25.6_{1.2}$ | $0.28_{0.00}$ | $25.1_{1.2}$ | $0.28_{0.00}$ | $24.6_{1.0}$ | $0.27_{0.00}$ |
| | All | $58.2_{0.7}$ | $0.24_{0.00}$ | $59.5_{0.6}$ | $0.26_{0.00}$ | $59.2_{0.7}$ | $0.26_{0.00}$ | $59.0_{0.7}$ | $0.26_{0.00}$ |

*Table 22.* Deep search and coding pass@1 and price per million tokens ($/Mt) across task-complexity bins for SALE with ablated self-feedback and peer-feedback.

| Task type | $\tau(t)$ | w/o self-feedback | | w/o peer-feedback | |
|---|---|---|---|---|---|
| | | Pass@1($\uparrow$) | $/Mt($\downarrow$) | Pass@1($\uparrow$) | $/Mt($\downarrow$) |
| *Deep search* | $\leq 0.1$ | $92.3_{0.5}$ | $0.26_{0.01}$ | $91.0_{0.9}$ | $0.06_{0.01}$ |
| | $\leq 0.5$ | $86.0_{0.9}$ | $0.21_{0.01}$ | $87.3_{1.8}$ | $0.07_{0.01}$ |
| | $\leq 2.5$ | $68.2_{0.6}$ | $0.22_{0.01}$ | $65.8_{2.0}$ | $0.07_{0.00}$ |
| | $\leq 12.5$ | $31.5_{2.9}$ | $0.17_{0.01}$ | $32.7_{1.6}$ | $0.09_{0.01}$ |
| | $\leq 60$ | $12.5_{1.9}$ | $0.21_{0.02}$ | $5.6_{1.3}$ | $0.06_{0.01}$ |
| | All | $64.1_{0.7}$ | $0.21_{0.00}$ | $63.2_{0.7}$ | $0.07_{0.00}$ |
| *Coding* | $\leq 0.1$ | $97.5_{0.8}$ | $0.17_{0.00}$ | $97.5_{0.0}$ | $0.05_{0.01}$ |
| | $\leq 0.5$ | $81.0_{1.1}$ | $0.27_{0.01}$ | $71.9_{1.9}$ | $0.05_{0.00}$ |
| | $\leq 2.5$ | $65.3_{2.9}$ | $0.27_{0.01}$ | $49.0_{1.7}$ | $0.06_{0.00}$ |
| | $\leq 12.5$ | $26.8_{0.6}$ | $0.29_{0.01}$ | $15.1_{1.5}$ | $0.08_{0.00}$ |
| | $\leq 60$ | $23.8_{0.5}$ | $0.29_{0.01}$ | $10.4_{1.8}$ | $0.12_{0.01}$ |
| | All | $58.8_{0.7}$ | $0.26_{0.00}$ | $48.7_{0.7}$ | $0.07_{0.00}$ |

## I.3. Memory Ablations

To assess the robustness of the memory-based refinement stage to aggressive pruning, we conduct two complementary experiments on a fixed pool of 100 tasks using a leave-one-out protocol. First, we vary the memory bank sample size $M \in \{0, 25, 50, 75, 99\}$, with 5 random seeds for sampling at each size. Second, we apply TTL pruning, retaining only the most recent $W$ records for $W \in \{10, 25, 50, 99\}$. Results are shown in Tables 23 and 24. For each configuration we report the average jury score of the initial bid (*Jury score – Original*), the average jury score of the refined bid where applicable (*Jury score – Refined*), the fraction of tasks where refinement flips the provisional winner (*Allocation – Flip rate*), and the share of tasks ultimately executed by the 32B agent (*Allocation – 32B share*).

The critical transition is from $M = 0$ to $M > 0$: enabling memory lowers 32B reliance from 35% to roughly 17% and raises refined-plan jury scores by approximately 0.3 points on the 0–5 scale. Beyond this point, increasing memory size yields only small further improvements, within seed-level noise ($\sigma = 1.7$–$2.6$ percentage points). TTL pruning shows the same pattern: even an aggressive 10-task window produces comparable jury scores and flip rates, with only a modestly higher 32B share (21% vs. 18% at $W = 99$). Together, these results indicate that memory-based refinement is robust to substantial pruning, and that a sliding window of ∼10–25 recent tasks is sufficient for effective refinement in practice.

*Table 23.* Effect of memory bank size $M$ on refinement quality and agent allocation. $M = 0$ corresponds to disabling memory-based refinement.

| Memory | Jury score | | Allocation | |
|---|---|---|---|---|
| | Original | Refined | Flip rate | 32B share |
| $M = 0$ | 3.36 | — | 0% | 35% |
| $M = 25$ | 3.38 | 3.68 | 19.1% | 17% |
| $M = 50$ | 3.36 | 3.67 | 17.5% | 17% |
| $M = 75$ | 3.42 | 3.72 | 17.3% | 16% |
| $M = 99$ | 3.41 | 3.74 | 17.2% | 17% |

*Table 24.* Effect of TTL window size $W$ on refinement quality and agent allocation. Only the most recent $W$ records are retained for retrieval.

| TTL window | Jury score | | Allocation | |
|---|---|---|---|---|
| | Original | Refined | Flip rate | 32B share |
| $W = 10$ | 3.47 | 3.68 | 16.1% | 21% |
| $W = 25$ | 3.45 | 3.68 | 16.1% | 21% |
| $W = 50$ | 3.45 | 3.65 | 16.7% | 19% |
| $W = 99$ | 3.46 | 3.74 | 16.7% | 18% |

# J. Jury Validation

To validate that LLM-jury scores produce a meaningful signal, we run two complementary analyses: (i) we measure alignment between jury scores and blind expert human annotations of plan quality, and (ii) we measure the predictiveness of jury scores with respect to downstream task success.

## J.1. Alignment with Human Judgments

We collect blind expert annotations on a stratified subset of 100 plans from HST-BENCH, evenly distributed across task types, complexity bins, and agent sizes. Annotators score each plan on the same 0–5 Likert scale used by the jury, without access to the jury's scores. We then compute the Spearman rank correlation between the average jury score and the human score within each complexity bin. Results are shown in Table 25.

Correlations are strong across all complexity bins ($\rho \in [0.74, 0.82]$) with no meaningful degradation as task complexity increases, supporting the use of jury scores as a reliable signal in the auction.

*Table 25.* Spearman rank correlation between SALE jury scores and blind expert human annotations on 100 stratified HST-BENCH plans.

| Complexity | Spearman $\rho$ |
|---|---|
| $\tau(t) \leq 0.1$ | 0.76 |
| $\tau(t) \leq 0.5$ | 0.74 |
| $\tau(t) \leq 2.5$ | 0.82 |
| $\tau(t) \leq 12.5$ | 0.75 |
| $\tau(t) \leq 60$ | 0.74 |

## J.2. Predictiveness of Task Success

To assess whether jury scores predict downstream success, we compute the point-biserial correlation between continuous jury scores and binary task accuracy on the full development set. We obtain a correlation of $0.36$ ($p < 0.01$), which falls within the range conventionally interpreted as good item discrimination (Ebel & Frisbie, 1991). Combined with the human-alignment results above, this provides converging evidence that the jury scoring signal is both aligned with human judgments of plan quality and predictive of downstream task success.

# K. Agent Selection Over Time

In Figure 10, we show the cumulative selection rate over time, defined as the running fraction of tasks ultimately delegated to each agent, for all agents that can refine their strategies during SALE's auction mechanism (4B, 8B, and 14B). Recall that an agent only enters the memory-based self-refinement stage when a more expensive agent is the provisional winner, so cheaper agents have more opportunities to refine their strategies than more expensive ones. As a result, the 4B agent generates a refined strategy on 92% of samples for deep search and 93% for coding, the 8B agent on 91% for deep search and 92% for coding, and the 14B agent on 62% for deep search and 77% for coding, leading to different patterns in how their cumulative selection rates evolve as auction memory accumulates.

Across both domains, the 4B agent's cumulative selection share (also reported in Figure 6) shows a clear upward trend, indicating that it is increasingly chosen as auction memory grows. The 8B agent's share is also broadly increasing for deep search, albeit with higher variance, but remains relatively flat for coding. In contrast, the 14B agent's cumulative share exhibits a mild downward drift in both domains, consistent with workload being gradually reallocated toward the cheaper agents. Because each curve at time step $t$ reflects the average selection rate computed over all tasks up to $t$, the trajectories are naturally more volatile at the beginning, when this average is based on few samples, and become smoother as more tasks accumulate.

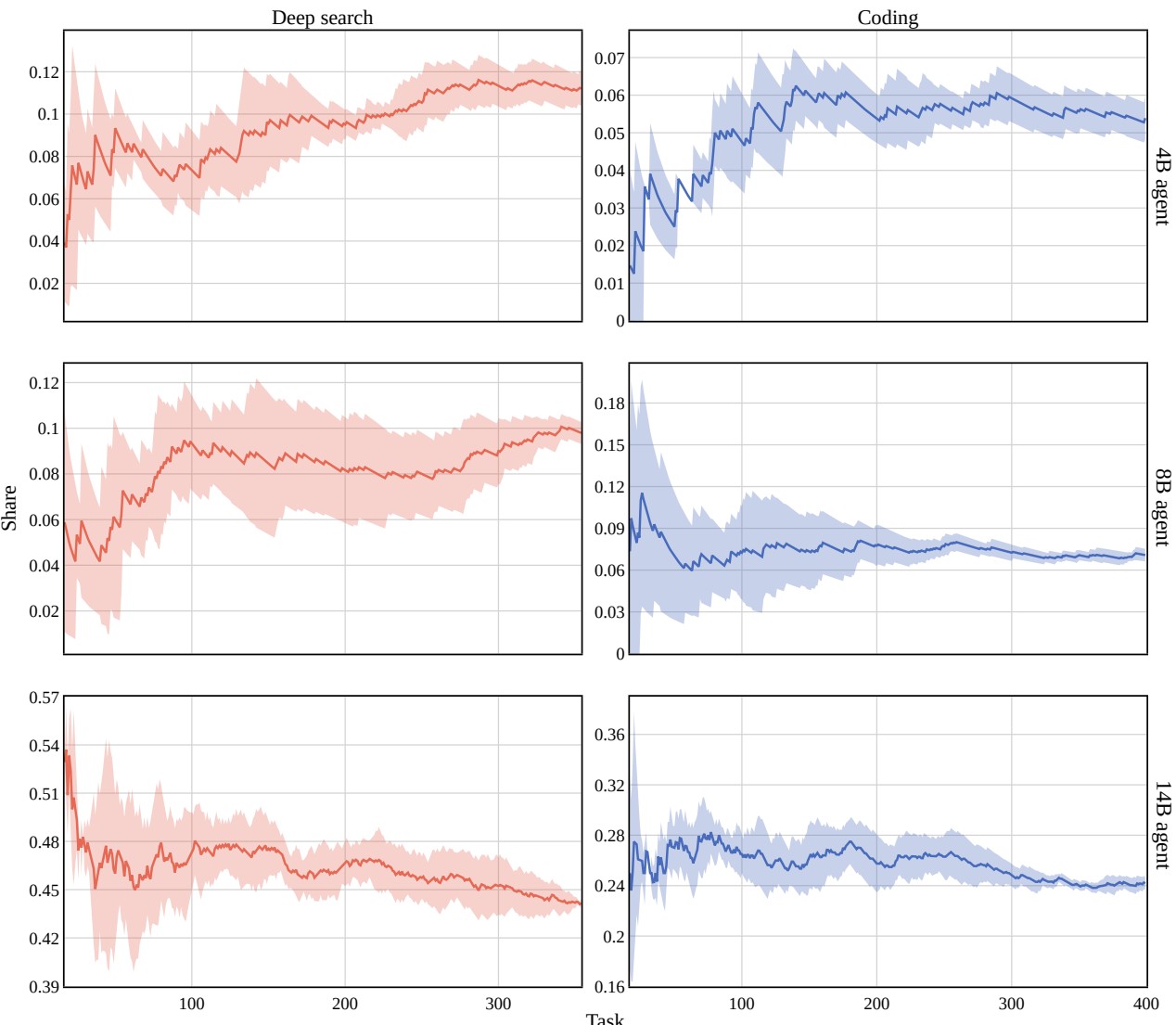

*Figure 10.* Cumulative selection rates over time for the 4B, 8B, and 14B agents on deep search and coding tasks, showing increasing delegation to cheaper agents while the 14B share gradually declines as auction memory accumulates.

# L. Qualitative Performance Comparison

This section presents examples, spanning task complexity levels, where the largest agent (32B) fails and a smaller agent succeeds. As these examples originate from the test set, we omit the agents' final answers. Given that full traces span tens of thousands to millions of tokens, we present only the agent's solution strategy and indicate whether the final answer is correct. In many cases, the strategy alone exhibits telltale signs of failure: misunderstanding of the task, excessive complexity, or hallucinated details that ultimately lead to an incorrect answer.

## L.1. Performance Comparison on Deep Search Tasks

For deep search tasks, we identify three recurring failure modes in the 32B agent. First, the agent occasionally bypasses tool use entirely and provides an answer from parametric memory, often incorrect. This tendency is sometimes visible in the initial strategy itself, where the eventual answer is already present despite a search plan having been outlined (Example 1). Second, the agent may over-engineer its approach to straightforward questions, pursuing tangential information, such as a musician's date of death (Example 2), or issuing excessive sequential queries that lead it further from the answer (Example 5). Third, the agent retrieves partial or related information from parametric knowledge that interferes with the search process: in Example 3, a legal case is conflated with an unrelated one, while in Example 4, parametric recall of an intermediate fact introduces ambiguity between a television series and its sequel.

---

**Deep Search Example 1**

---

**Task**

A patient suffers a stroke just in the tip of the paracentral artery on the left side where the artery rounds over to the lateral surface. This would likely result in
Answer Choices:
A. More sensory loss in the right arm than the foot
B. More sensory loss in the right foot than the arm
C. More sensory loss in the left arm than the foot
D. More sensory loss in the left foot than the arm
E. More weakness of the right foot than the arm
Answer with the letter of the single correct option.

**Incorrect 32B response**

**Plan:**
`ask_search_agent`: What is the anatomical territory of the paracentral artery and its role in sensory/motor functions? Provide a summary of the brain regions it supplies and their functional roles.
`ask_search_agent`: How does a stroke in the left paracentral artery affect sensory/motor functions contralaterally? Provide a summary of the expected deficits on the right side of the body.
`ask_search_agent`: In the sensory homunculus, is the foot or arm more represented in the medial portion of the paracentral lobule? Provide a concise answer based on neuroanatomy resources.
`final_answer`: B

**Final answer:** ❌

**Correct 14B response**

**Plan:**
`ask_search_agent`: What is the somatotopic representation of the arm and foot in the paracentral lobule of the somatosensory cortex? Provide a concise summary of the anatomical distribution. `ask_search_agent`: Does a stroke in the left paracentral lobule cause contralateral sensory loss in the right arm or foot? Provide a concise summary of the clinical implications.

**Final answer:** ✅

---

**Deep Search Example 2**

**Task**

How many studio albums were published by Mercedes Sosa between 2000 and 2009 (included)? You can use the latest version of english wikipedia.

**Incorrect 32B response**

**Plan:**

1. Use `ask_search_agent` to retrieve Mercedes Sosa's discography (studio albums with release years) from the English Wikipedia page. Specify the query: "List all studio albums by Mercedes Sosa with their release years. Provide a table of album titles and years." Website: https://en.wikipedia.org/wiki/Mercedes_Sosa.
2. Use `ask_search_agent` to confirm Mercedes Sosa's death date and check for posthumous album releases in 2009. Query: "When did Mercedes Sosa die? Were any studio albums released posthumously in 2009? Provide dates and album names."
3. Filter the discography data to count studio albums released between 2000–2009 (inclusive), excluding posthumous 2009 releases.
4. Use `final_answer` to submit the verified count.

**Final answer:** 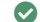

**Correct 8B response**

**Plan:**

1. Use `ask_search_agent` to retrieve the list of studio albums by Mercedes Sosa along with their release years from the English Wikipedia page (https://en.wikipedia.org/wiki/Mercedes_Sosa). Provide `output_content` as a structured list of albums with release years.
2. Extract the release years of all studio albums from the search result.
3. Filter the albums to include only those released between 2000 and 2009 (inclusive).
4. Count the number of filtered albums.
5. Use `final_answer` to provide the derived count.

**Final answer:** 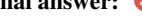

---

**Deep Search Example 3**

**Task**

What amount was the settlement that the character from the Son of al Qaeda got in 2017?

**Incorrect 32B response**

**Plan:**

1. Use `ask_search_agent` to verify if "Son of al Qaeda" is a real entity or a misstatement (e.g., "Son of Sam"). Query: "Is 'Son of al Qaeda' a real person or legal case? If not, what is the correct reference (e.g., 'Son of Sam')? Provide clarification."
2. If the search confirms a misstatement (e.g., "Son of Sam"), use `ask_search_agent` to check for 2017 settlements: "What legal settlement amount was associated with the 'Son of Sam' case in 2017? Provide the numerical value."
3. If no 2017 data is found, cross-reference historical records for the original "Son of Sam" case (1979) and determine if updates or subsequent settlements occurred in 2017.
4. Use `final_answer` to report the verified settlement amount.

**Final answer:** 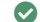

**Correct 8B response**

**Plan:**

1. Use `ask_search_agent` to identify what "Son of al Qaeda" refers to (e.g., a real person, fictional character, or legal case). Query: "What is 'Son of al Qaeda'? Provide `output_content`".
2. Use `ask_search_agent` to determine the specific settlement amount related to the identified entity in 2017. Query: "What was the 2017 settlement amount related to [entity from Step 1]? Provide `output_content`".
3. Verify the settlement amount via credible sources (e.g., court records, news articles) using `ask_search_agent` if needed.
4. Use `final_answer` to state the confirmed settlement amount.

**Final answer:** 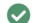

**Deep Search Example 4**

**Task**

What season was the character introduced that becomes the main antagonist in the following season, from the animated television series created by Bryan Konietzko and Michael Dante DiMartino as a sequel to "...", which aired from 2005 to 2008?

**Incorrect 32B response**

**Plan:**
1. Use `ask_search_agent` to confirm the name of the sequel series to "Avatar: The Last Airbender" created by Bryan Konietzko and Michael Dante DiMartino. Provide the name of the sequel.
2. Use `ask_search_agent` to identify characters introduced in each season of the sequel series who later become main antagonists in subsequent seasons. Provide a list of characters with their introduction season and antagonist season.
3. Use `ask_search_agent` to verify the specific season number where the character was introduced that becomes the main antagonist in the following season. Provide the season number.

**Final answer:** 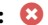

**Correct 8B response**

**Plan:**
1. Use `ask_search_agent` to identify the original series (2005-2008) created by Bryan Konietzko and Michael Dante DiMartino, and confirm the sequel series name. Query: "What is the name of the animated series created by Bryan Konietzko and Michael Dante DiMartino that aired from 2005 to 2008? Provide the name."
2. Use `ask_search_agent` to determine the main antagonists for each season of the sequel series. Query: "What are the main antagonists for each season of [sequel series name]? Provide a list of seasons and corresponding antagonists."
3. Use `ask_search_agent` to identify which antagonist was introduced in a season and became the main antagonist in the following season. Query: "In which season was a character introduced in [sequel series name] that became the main antagonist in the next season? Provide the season number."
4. Use `final_answer` to compile the result from step 3.

**Final answer:** 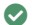

**Deep Search Example 5**

**Task**

D1NZ is a series based on what oversteering technique?

**Incorrect 32B response**

**Plan:**
`ask_search_agent`: "What is the D1NZ series and which oversteering technique is it based on? Provide a concise explanation of the series and the specific oversteering technique it employs."
`ask_search_agent`: "What oversteering techniques are commonly used in drifting competitions like D1NZ? Provide a list of techniques and their association with specific drifting series or events."
`ask_search_agent`: "What vehicle specifications or rules define the D1NZ drifting series? Provide details on car configurations that indicate the required oversteering technique."
`final_answer`: <answer derived from cross-referencing D1NZ's vehicle specs, competition style, and associated oversteering techniques>

**Final answer:** 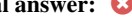

**Correct 4B response**

**Plan:**
1. Use `ask_search_agent` to identify the oversteering technique associated with the series D1NZ. Query: "What oversteering technique is the series D1NZ based on? Provide the name of the technique."
2. Use `final_answer` to submit the identified oversteering technique as the answer.

**Final answer:** 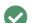

## L.2. Performance Comparison on Coding Tasks

For coding tasks, we observe analogous failure modes in the 32B agent. First, the agent over-engineers straightforward problems: in Example 6, it plans a recursive solution for what is explicitly fixed two-level nesting, whereas the 4B correctly identifies the specific structure. Such unnecessary complexity increases the surface area for errors. Second, the agent draws on parametric knowledge at the expense of task constraints: Example 7 shows the 32B planning to import a library function not in the available import list, while the 14B opts for a simpler, more robust approach. This overconfidence extends to skipping the planning phase entirely: in Examples 8 and 9, the 32B produces code directly in place of a high-level strategy, without using the Python shell for verification and pursuing over-optimized solutions that introduce subtle errors, e.g., ignoring intermediate computation steps or applying mathematically incorrect shortcuts, where simpler approaches would have fewer failure points. Third, the agent neglects edge cases: in Example 10, the 32B fails to handle the "no duplicate found" case, resulting in an incorrect return type that breaks hidden tests.

**Coding Example 6**

**Task**

You are an expert Python programmer, and here is your task: Write a function to convert a tuple of string values to a tuple of integer values. Your code should pass the following test: `assert tuple_int_str((('333', '33'), ('1416', '55')))==((333, 33), (1416, 55))`

**Incorrect 32B response**

**Plan:**
1. Analyze the task: Create a function that converts nested tuples of strings to integers.
2. Implement a solution using nested tuple comprehensions to recursively convert each string element to an integer.
3. Validate the solution against the provided test case to ensure correctness.

**Final answer:** ✖

**Correct 4B response**

**Plan:**
The plan is to create a function that converts each string in the input tuple of tuples to an integer, preserving the nested structure. This is achieved by using a generator expression within a `tuple()` to iterate over each inner tuple, convert each string to an integer, and collect the results into a new tuple of tuples.

**Final answer:** ✔

**Coding Example 7**

**Task**

You are an expert Python programmer, and here is your task: Write a function to merge multiple sorted inputs into a single sorted iterator using heap queue algorithm. Your code should pass the following tests: `assert merge_sorted_list([25, 24, 15, 4, 5, 29, 110],[19, 20, 11, 56, 25, 233, 154],[24, 26, 54, 48])==[4, 5, 11, 15, 19, 20, 24, 24, 25, 25, 26, 29, 48, 54, 56, 110, 154, 233]`

**Incorrect 32B response**

**Plan:**
1. Import the `heapq` module to utilize the merge function for efficiently combining sorted iterables.
2. Define a function `merge_sorted_list` that accepts multiple lists as input.
3. For each input list, sort it individually to ensure they are in ascending order.
4. Use `heapq.merge` to merge the sorted lists into a single sorted iterator.
5. Convert the merged iterator to a list and return it as the final result.

**Final answer:** 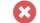

**Correct 14B response**

**Plan:**
1. Define the function `merge_sorted_list` to accept multiple lists as input.
2. Sort each input list individually to ensure they are in ascending order.
3. Initialize a min-heap to track the smallest elements from each sorted list, storing tuples of (value, list index, element index).
4. While the heap is not empty, extract the smallest element, append it to the result, and push the next element from the same list into the heap if available.
5. Return the merged sorted list as the final result.

**Final answer:** 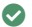

**Coding Example 8**

**Task**

You are an expert Python programmer. You will be given a question (problem specification) and will generate a correct Python program that matches the specification and passes all tests.

### Question:
You are given an array of integers nums of size `n` and a positive integer threshold.
There is a graph consisting of n nodes with the `ith` node having a value of `nums[i]`. Two nodes `i` and `j` in the graph are connected via an undirected edge if `lcm(nums[i], nums[j]) <= threshold`.
Return the number of connected components in this graph.
A connected component is a subgraph of a graph in which there exists a path between any two vertices, and no vertex of the subgraph shares an edge with a vertex outside of the subgraph.
The term `lcm(a, b)` denotes the least common multiple of a and b.

Example 1:

Input: `nums = [2,4,8,3,9]`, `threshold = 5`
Output: `4`
Explanation:
The four connected components are `(2, 4)`, `(3)`, `(8)`, `(9)`.

Example 2:

Input: `nums = [2,4,8,3,9,12]`, `threshold = 10`
Output: `2`
Explanation:
The two connected components are `(2, 3, 4, 8, 9)`, and `(12)`.

Constraints:

```
1 <= nums.length <= 105
1 <= nums[i] <= 109
```
All elements of nums are unique.
```
1 <= threshold <= 2 * 105
```

### Format: You will use the following starter code to write the solution to the problem and enclose your code within delimiters.

````
```python final_answer("""
class Solution:
    def countComponents(self, nums:  List[int], threshold:  int) -> int:

""")```
````

### Answer: (use the provided format with backticks)

**Incorrect 32B response**

**Plan:**
To solve this problem, we need to determine the number of connected components in a graph where nodes are connected if the least common multiple (LCM) of their corresponding values is less than or equal to a given threshold. The solution involves efficiently using Union-Find (Disjoint Set Union, DSU) to group connected nodes.

### Approach

1.  **Filter and Sort**: First, filter out elements from the input array that are greater than the threshold and sort the remaining elements.
2. **Union-Find Data Structure**: Use a Union-Find (DSU) structure to manage the connected components.
3. **Process Multiples**: For each number in the filtered list, connect it with all its multiples that are also in the list and within the threshold.
4. **Process Pairs with LCM Check**: For each number, check pairs where the LCM is within the threshold and connect them if they meet the condition.
5. **Count Components**: After processing all connections, count the number of unique connected components using the DSU structure and add components from elements that were filtered out due to exceeding the threshold.

### Solution Code

```python
import bisect
from math import gcd
from typing import List

class Solution:
    def countComponents(self, nums:  List[int], threshold:  int) -> int:
        # Filter and sort the numbers that are <= threshold
        filtered = [x for x in nums if x <= threshold]
        filtered.sort()
        n = len(filtered)
        if n == 0:
            return len(nums)

        # Map each value to its index in the filtered list
        value_to_index = {v:  i for i, v in enumerate(filtered)}

        # Define DSU (Disjoint Set Union) structure
        class DSU:
            def __init__(self, size):
                self.parent = list(range(size))
                self.rank = [0] * size

            def find(self, x):
                if self.parent[x] != x:
```

```
                self.parent[x] = self.find(self.parent[x])
            return self.parent[x]

        def union(self, x, y):
            x_root = self.find(x)
            y_root = self.find(y)
            if x_root == y_root:
                return
            if self.rank[x_root] < self.rank[y_root]:
                self.parent[x_root] = y_root
            else:
                self.parent[y_root] = x_root
                if self.rank[x_root] == self.rank[y_root]:
                    self.rank[x_root] += 1

    dsu = DSU(n)

    # Process each element in the filtered list
    for i in range(n):
        x = filtered[i]
        # Connect with multiples of x
        m = 2 * x
        while m <= threshold:
            if m in value_to_index:
                idx = value_to_index[m]
                dsu.union(i, idx)
            m += x
        # Connect with other numbers where LCM(x, y) <= threshold
        upper = threshold // x
        if upper >= x:
            left = i + 1
            right_idx = bisect.bisect_right(filtered, upper) - 1
            if left <= right_idx:
                for j in range(left, right_idx + 1):
                    y = filtered[j]
                    if (x * y) // gcd(x, y) <= threshold:
                        dsu.union(i, j)

    # Count the number of unique roots in DSU
    roots = set()
    for i in range(n):
        roots.add(dsu.find(i))

    # Total components is the number of roots in DSU plus elements > threshold
    return len(roots) + (len(nums) - n)
```

### Explanation

1. **Filtering and Sorting**: The input array is filtered to include only values that are less than or equal to the threshold. This helps in reducing unnecessary computations.
2. **Union-Find Structure**: The DSU structure is used to efficiently manage and merge connected components.
3. **Multiples Processing**: For each number, we connect it with all its multiples within the threshold. This ensures that all direct connections based on multiples are considered.
4. **LCM Check for Pairs**: For each number, we check pairs where the LCM is within the threshold. This ensures that even non-multiples that meet the LCM condition are connected.
5. **Component Counting**: After processing all connections, the number of unique roots in the DSU structure gives the number of connected components for the filtered values. Components from values exceeding the threshold are added separately.

This approach efficiently handles the constraints and ensures that all valid connections are considered, leading to the correct count of connected components.

**Final answer:** ✗

**Correct 14B response**

**Plan:**
1. **Sort the input array NUMS** to facilitate efficient processing of multiples.
2. **Create a mapping from values to their indices** to quickly check if a number exists in the array.
3. **Use a Disjoint Set Union (DSU) structure** to manage connected components.
4. **For each number in NUMS**, iterate through its multiples up to the threshold and connect the number with its multiples if they exist in the array.
5. **Count the number of unique parents** in the DSU to determine the number of connected components.
6. **Return the count of connected components** as the result.

**Final answer:** 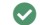

---

**Coding Example 9**

---

**Task**

You are an expert Python programmer. You will be given a question (problem specification) and will generate a correct Python program that matches the specification and passes all tests.

### Question:
You are given a string `initialCurrency`, and you start with 1.0 of `initialCurrency`.
You are also given four arrays with currency pairs (strings) and rates (real numbers):

`pairs1[i] = [startCurrencyi, targetCurrencyi]` denotes that you can convert from `startCurrencyi` to `targetCurrencyi` at a rate of `rates1[i]` on day 1.
`pairs2[i] = [startCurrencyi, targetCurrencyi]` denotes that you can convert from `startCurrencyi` to `targetCurrencyi` at a rate of `rates2[i]` on day 2.
Also, each `targetCurrency` can be converted back to its corresponding `startCurrency` at a rate of `1 / rate`.

You can perform any number of conversions, including zero, using `rates1` on day 1, followed by any number of additional conversions, including zero, using `rates2` on day 2.
Return the maximum amount of `initialCurrency` you can have after performing any number of conversions on both days in order.
Note: Conversion rates are valid, and there will be no contradictions in the rates for either day. The rates for the days are independent of each other.

Example 1:

Input:  initialCurrency = "EUR", pairs1 = [["EUR","USD"],["USD","JPY"]], rates1 = [2.0,3.0], pairs2 = [["JPY","USD"],["USD","CHF"],["CHF","EUR"]], rates2 = [4.0,5.0,6.0]
Output: 720.00000
Explanation:
To get the maximum amount of EUR, starting with 1.0 EUR:

On Day 1:

Convert EUR to USD to get 2.0 USD.
Convert USD to JPY to get 6.0 JPY.

On Day 2:

Convert JPY to USD to get 24.0 USD.
Convert USD to CHF to get 120.0 CHF.
Finally, convert CHF to EUR to get 720.0 EUR.

Example 2:

Input:        initialCurrency = "NGN", pairs1 = [["NGN","EUR"]], rates1 = [9.0], pairs2 = [["NGN","EUR"]], rates2 = [6.0]
Output: 1.50000
Explanation:
Converting NGN to EUR on day 1 and EUR to NGN using the inverse rate on day 2 gives the maximum amount.

Example 3:

Input:                 initialCurrency = "USD", pairs1 = [["USD","EUR"]], rates1 = [1.0], pairs2 =
[["EUR","JPY"]], rates2 = [10.0]
Output: 1.00000
Explanation:
In this example, there is no need to make any conversions on either day.

Constraints:

```
1 <= initialCurrency.length <= 3
```
initialCurrency consists only of uppercase English letters.
```
1 <= n == pairs1.length <= 10
1 <= m == pairs2.length <= 10
pairs1[i] == [startCurrencyi, targetCurrencyi]
pairs2[i] == [startCurrencyi, targetCurrencyi]
1 <= startCurrencyi.length, targetCurrencyi.length <= 3
```
startCurrencyi and targetCurrencyi consist only of uppercase English letters.
```
rates1.length == n
rates2.length == m
1.0 <= rates1[i], rates2[i] <= 10.0
```
The input is generated such that there are no contradictions or cycles in the conversion graphs for either day.
The input is generated such that the output is at most 5 * 1010.

### Format: You will use the following starter code to write the solution to the problem and enclose your code within delimiters.
```python final_answer("""
class Solution:
     def maxAmount(self, initialCurrency:  str, pairs1:  List[List[str]], rates1:
List[float], pairs2:  List[List[str]], rates2:  List[float]) -> float:

""")```

### Answer: (use the provided format with backticks)

**Incorrect 32B response**

**Plan:**
```python
class Solution:
    def maxAmount(self, initialCurrency:  str, pairs1:  List[List[str]], rates1:
List[float], pairs2:  List[List[str]], rates2:  List[float]) -> float:
        def bellman_ford(start, edges, all_currencies):
            max_amounts = currency:  -float('inf') for currency in all_currencies
            max_amounts[start] = 1.0
            V = len(all_currencies)
            for _ in range(V - 1):
                updated = False
                for (u, v, rate) in edges:
                    if max_amounts[u] != -float('inf'):
                        if max_amounts[u] * rate > max_amounts[v]:
                            max_amounts[v] = max_amounts[u] * rate
                            updated = True
                if not updated:
                    break
            return max_amounts

        # Process day1
        edges1 = []
        for i in range(len(pairs1)):
```

```
            a, b = pairs1[i]
            rate = rates1[i]
            edges1.append((a, b, rate))
            edges1.append((b, a, 1.0 / rate))
        currencies_day1 = set()
        for a, b in pairs1:
            currencies_day1.add(a)
            currencies_day1.add(b)
        currencies_day1.add(initialCurrency)
        day1_max = bellman_ford(initialCurrency, edges1, currencies_day1)

        # Process day2
        edges2 = []
        for i in range(len(pairs2)):
            a, b = pairs2[i]
            rate = rates2[i]
            edges2.append((a, b, rate))
            edges2.append((b, a, 1.0 / rate))
        currencies_day2 = set()
        for a, b in pairs2:
            currencies_day2.add(a)
            currencies_day2.add(b)
        max_result = 1.0 # initial amount

        for c in day1_max:
            amount_c = day1_max[c]
            if amount_c == -float('inf'):
                continue
            all_currencies_day2_run = currencies_day2.copy()
            all_currencies_day2_run.add(c)
            day2_max = bellman_ford(c, edges2, all_currencies_day2_run)
            current_value = day2_max.get(initialCurrency, -float('inf'))
            if current_value > max_result:
                max_result = current_value

        return max_result```
```

**Final answer:** 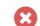

**Correct 8B response**

**Plan:**
To solve the problem of maximizing the amount of the initial currency after performing conversions on two days, we need to model the conversion process as a graph traversal problem. The key steps involve:

1. **Building the Conversion Graphs**: For each day, construct a graph where nodes represent currencies and edges represent conversion rates (both forward and reverse).

2. **Topological Sorting**: Since the input guarantees no cycles, we can perform a topological sort on the graph to process nodes in an order that ensures all dependencies are resolved.

3. **Maximizing Currency Amounts**: Using dynamic programming on the topologically sorted nodes, compute the maximum amount of each currency achievable after each day's conversions.

4. **Combining Results**: After processing both days, compute the maximum amount of the initial currency by considering all possible paths from the currencies obtained after day 2 back to the initial currency.

—

### Step-by-Step Plan

1. **Build the Graph for Day 1**:
- Construct a graph where each currency pair from `pairs1` and `rates1` is represented as bidirectional edges (forward and reverse conversions).

2. **Compute Day 1 Amounts**:
- Starting with 1.0 of the initial currency, use topological sorting to compute the maximum amount of each currency achievable after day 1 conversions.
3. **Build the Graph for Day 2**:
- Construct a similar graph using `pairs2` and `rates2`.
4. **Compute Day 2 Amounts**:
- Use the results from day 1 as the starting point for day 2 conversions. Again, use topological sorting to compute the maximum amount of each currency after day 2.
5. **Convert Back to Initial Currency**:
- For each currency obtained after day 2, compute the maximum amount of the initial currency that can be obtained by converting through day 2's graph.
6. **Return the Maximum Result**:
- The final answer is the maximum value among all possible conversions back to the initial currency.

—

### Final Plan

1. **Build the day1 graph** using `pairs1` and `rates1`, including both forward and reverse conversions.
2. **Compute day1_amounts** by traversing the graph in topological order, starting from the initial currency.
3. **Build the day2 graph** using `pairs2` and `rates2`, including both forward and reverse conversions.
4. **Compute day2_amounts** by traversing the day2 graph in topological order, starting from the day1_amounts.
5. **For each currency in day2_amounts**, compute the maximum amount of the initial currency by traversing the day2 graph again, starting from that currency.
6. **Return the maximum value** among all computed results.

**Final answer:** 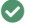

---

**Coding Example 10**

**Task**

You are an expert Python programmer, and here is your task: Write a function to find the first duplicate element in a given array of integers. Your code should pass the following test: `assert find_first_duplicate(([1, 2, 3, 4, 4, 5]))==4`

**Incorrect 32B response**

**Plan:**
1. Create a function `find_first_duplicate` that takes an array as input.
2. Initialize an empty set to track seen elements.
3. Iterate through each element in the array:
a. If the element is already in the set, return it immediately.
b. Otherwise, add the element to the set.
4. Use the `final_answer` tool to return the implemented function code.

**Final answer:** 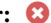

**Correct 14B response**

**Plan:**
The task is to write a Python function that identifies the first duplicate element in an array of integers. The approach involves iterating through the array while tracking seen elements using a set. When a duplicate is encountered, it is immediately returned as the result.

1. Initialize an empty set to keep track of elements that have been encountered.
2. Iterate through each element in the input array.
3. For each element, check if it is already in the set:
- If it is, return the element as it is the first duplicate.
- If it is not, add the element to the set.
4. If no duplicates are found during the iteration, return a default value (e.g., -1) to indicate the absence of duplicates.

This method ensures that the first duplicate is identified efficiently with a time complexity of $O(n)$ and a space complexity of $O(n)$, where n is the number of elements in the array.

**Final answer:** 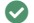

---

# M. Cross-Family Generalization

To assess whether the auction mechanism and its learned weights transfer beyond the Qwen3 family, we run two complementary experiments: (i) we re-run the auction on a different open-weight family (Gemma 3) and check whether the same mechanism produces complexity-sensitive allocations and comparable auction signals, and (ii) we introduce cross-family agents (Llama 3.1, Ministral 3) into the Qwen3 pool without retraining the MILP weights and verify that they receive sensible allocations.

## M.1. Auction Transfer to Gemma 3

We run SALE's auction stage on the Gemma 3 family (4B, 12B, 27B), which differs from Qwen3 in both architectural details and training recipe. We generate strategic plans for 100 HST-BENCH tasks (evenly stratified by type and complexity), compute the cost–value proxies (with a Gemma 3 jury), and run the MILP independently on this Gemma-only pool to learn auction weights from scratch.

The auction mechanism transfers cleanly: the MILP converges and produces complexity-sensitive allocations matching those observed for Qwen3. On the hardest coding tasks ($\tau(t) \in (12.5, 60]$), the auction routes 80% of work to Gemma 27B, mirroring the pattern observed for Qwen3 32B. On easier tasks, work is distributed more evenly across agent sizes, consistent with our main finding that smaller agents are sufficient at low complexity.

Auction signals also follow a similar trend across families. Cross-family comparisons between size-matched Gemma 3 and Qwen3 agents (4B–4B, 12B–14B, 27B–32B) show convergence across all three signals: strategy-length distributions have Wasserstein distance as low as 9 tokens on average for the largest models, jury scores correlate strongly (Spearman $\rho \in [0.45, 0.76]$, all $p < 0.001$), and entropy correlates significantly (Spearman $\rho \in [0.46, 0.60]$, all $p < 0.001$) despite different absolute scales, which the MILP compensates for via its learned weight $w_h$. The two families' juries thus converge on the same task-level quality signal.

## M.2. Heterogeneous Agent Pool

To test SALE under heterogeneous conditions, we expand the Qwen3 pool (4B, 8B, 14B, 32B) with two agents from different families: Llama 3.1 8B and Ministral 3 8B. The two cross-family agents act as bidders only; the jury remains the original Qwen3 four-agent panel, so the per-agent judge weights $w_j$ are unchanged. This isolates the question of whether the global cost and entropy weights ($w_c$, $w_h$) transfer to a heterogeneous bidder pool; assigning new judges would require either re-running the MILP or transferring weights from size-matched agents, which we leave for future work. Crucially, we freeze the MILP weights learned on Qwen3 and perform no retraining: the cross-family agents are dropped into the existing auction unchanged. On the same 100-task subset, both new agents receive non-trivial workload shares that decrease with task complexity, consistent with their 8B-class size (Table 26).

We find that the cross-family agents behave as expected under the auction: their entropy and jury scores are close to Qwen3 8B (e.g., Llama jury mean 3.48 vs Qwen3 8B 3.56), and per-task cost-minus-value scores correlate with Qwen3 8B rankings (Spearman $\rho = 0.40$, $p < 0.01$). Taken together with the Gemma 3 results in Section M.1, these findings show that SALE's auction signals (i.e., the strategy text, entropy, and peer jury scores) are not family-specific, and that the mechanism is able to coordinate agents across different architectures and training recipes.

*Table 26.* Workload allocation to Llama 3.1 8B and Ministral 3 8B when added to the Qwen3 pool, with MILP weights frozen on Qwen3.

| Complexity | Llama 3.1 8B | Ministral 3 8B |
|---|---|---|
| $\tau(t) \leq 0.1$ | 25% | 20% |
| $\tau(t) \leq 0.5$ | 20% | 15% |
| $\tau(t) \leq 2.5$ | 15% | 10% |
| $\tau(t) \leq 12.5$ | 15% | 10% |
| $\tau(t) \leq 60$ | 5% | 5% |
| Overall | 16% | 12% |

