# OpenReview forum: "Scaling Small Agents Through Strategy Auctions"
_ICML.cc/2026/Conference — ICML 2026 regular_

### Official Review · Reviewer_A2EB · 2026-02-22

**Soundness:** 3
**Presentation:** 3
**Significance:** 3
**Originality:** 3
**Overall Recommendation:** 4
**Confidence:** 4

**Summary:**

This paper studies how to scale agentic performance without always paying for a single large model. The authors introduce strategy auctions where multiple agents propose plans, score one another, and then allocate execution to the agents that look most promising under a cost and value objective. A simple memory mechanism refines plans over time so that cheaper agents can reuse strategies that worked earlier. To evaluate this setting the paper also introduces HST BENCH which pairs agentic tasks with human solution time as a proxy for task complexity. Across deep search and coding tasks the auction ensemble improves pass at 1 while reducing cost relative to the best single agent and relative to several existing routing baselines.

**Compliance With Llm Reviewing Policy:**

Affirmed.

**Final Justification:**

I am keeping my overall recommendation at 4 weak accept.

The paper studies how agentic performance scales when using small language models and introduces Strategy Auctions for Workload Efficiency, where multiple agents propose plans, score one another, and an auction mechanism allocates execution based on a cost value objective. A key contribution is HST Bench, which uses human solution time as an operational difficulty signal and supports evaluation across deep search and coding tasks. The method is well engineered, and the cost performance curves and router comparisons make the story credible.

The rebuttal addresses my main evaluation questions. The judge sensitivity ablations suggest the gains are not driven by a single evaluator, and the human validation on a stratified subset reports strong rank correlation across complexity bins, roughly in the 0.74 to 0.82 range. The discussion of how to extend the cost proxy when tool latency dominates and the plan to add retrieval and pruning mechanisms for a growing strategy memory are sensible. The rebuttal also acknowledges the dual use angle more directly.

There are still open issues that keep this at weak accept rather than a higher score. Cross model family transfer is still largely a hypothesis, and a small pilot would strengthen the claim that the auction features generalize beyond the evaluated agent pool. Similarly, a deployment style experiment where tool calls dominate cost would help validate the modular cost modeling story. On memory, it remains important to demonstrate robustness to staleness and distribution shift when strategies are reused over long horizons.

Overall the rebuttal improves rigor without changing the basic balance of strengths and weaknesses. I keep my score unchanged.

**Key Questions For Authors:**

1 How sensitive are the results to the choice of judge and to the details of the scoring prompts used for peer evaluation

2 Can you report a small human validation on a subset of HST BENCH to quantify judge error as a function of task complexity

3 How well do the learned auction weights transfer across different model families and across different price spreads

4 In deployments where tool latency dominates token cost, how does SALE behave and can the cost proxy be adapted to include tool calls and wall clock time

5 What is the failure mode when the strategy memory grows large and do you see degradation from stale strategies

**Limitations:**

They are fairly candid about practical constraints such as the overhead of additional tokens for strategy generation and the fact that the approach is evaluated on a fixed set of agent models. I would still like a clearer discussion of judge dependence and of how the method behaves under domain shift when the memory contains strategies from earlier distributions. A short note on safety would also be appropriate since cheaper scaling of agentic capability can amplify misuse in the same way that cheaper scaling of base models can.

**Strengths And Weaknesses:**

Their benchmark contribution is strong. Using human solution time as an operational notion of difficulty is sensible for agentic workloads and the paper explains the data construction and annotation pipeline in enough detail that the results are interpretable. The method itself is also well-engineered. The auction features are purposefully low dimensional, the design separates a cost proxy from a value proxy, and the memory based refinement is evaluated as a first class component rather than an afterthought. Empirically the cost performance curves are convincing across multiple complexity bins, and the comparison to both predictive routers and  execution baselines helps clarify where the gains come from.

---

> ### Author Rebuttal · Authors · 2026-03-30
>
> Dear Reviewer A2EB,
>
> Thank you for your review and for acknowledging the strength of our benchmark and the fact that the method is well-engineered. Please find our responses to your questions below.
>
> >**Question 1:** *How sensitive are the results to the choice of judge and to the details of the scoring prompts*
> - Our ablations in Appendix I directly address sensitivity to judge choice: Tab. 18 replaces the full jury with each single judge, Tab. 19 removes one judge at a time, and Tab. 20 ablates self- vs peer-judgment. Every configuration underperforms the full jury, and no single judge is consistently best, confirming the system is not dependent on any particular judge.
>
> - Regarding sensitivity to the scoring prompt, our prompt is 0-shot so it does not suffer from example sensitivity. We did vary the scoring instructions in preliminary experiments: we tested 1-100 and 0-5 Likert scales and found 0-5 has higher correlation with human scores, a finding also supported by recent work comparing LLM-judge grading scales [1].
>
> >**Question 2:** *Can you report a small human validation on a subset of HST BENCH to quantify judge error as a function of task complexity*
> - Please find below the Spearman correlation between SALE jury scores and blind expert human annotations on 100 HST-Bench plans, evenly stratified by task type, complexity, and agent size.
>
> |Complexity|Spearman ρ|
> |-|-|
> |τ ≤ 0.1|0.7571|
> |τ ≤ 0.5|0.7370|
> |τ ≤ 2.5|0.8177|
> |τ ≤ 12.5|0.7504|
> |τ ≤ 60|0.7396|
> - All $p<.001$. Correlations are strong across all bins with no meaningful degradation as complexity increases. We plan to expand this analysis to a larger sample with three annotators and include it in the paper.
>
> >**Question 3:** *How well do the learned auction weights transfer across different model families and different price spreads*
> - The learned weights operate over largely model-agnostic features (strategy length, entropy, jury scores), so even though we have not tested this empirically, we envision no structural barrier to transfer. For substantially different price spreads, the weights would likely benefit from re-tuning, but the MILP optimisation is inexpensive and runs in seconds on a development set (see Section D.3). We consider cross-family evaluation an important direction for future work and we will add a brief discussion of the above.
>
> >**Question 4:** *In deployments where tool latency dominates token cost, how does SALE behave and can the cost proxy be adapted to include tool calls and wall clock time*
> - Thanks for the question; we do address this in the Limitations section and are happy to elaborate here. The cost function is modular by design, so in deployments where tool latency is significant it can be extended to incorporate tool-related signals. Since strategies already encode tool selection and decomposition steps, anticipated tool calls are visible in the plan and could serve as additional cost features. Alternatively, per-token price could be replaced with a blended cost-per-step metric that includes empirical tool latency. Neither extension would require changes to the auction mechanism itself. We will spell this out in the revised paper to provide practical guidance for deploying SALE in settings where token costs are not the dominant factor.
>
> >**Question 5:** *What is the failure mode when the strategy memory grows large and do you see degradation from stale strategies*
> - The memory bank stores task-level strategic plans representing high-level solution patterns (e.g., task decomposition, tool selection, search constraints) rather than model-specific execution traces. During refinement, contrastive pairs of winning and losing past strategies are retrieved, and while at least one comes from the refining agent itself, the other may have been generated by any agent in the pool. The learning signal thus captures what works for a class of tasks rather than model-specific patterns, limiting the impact of model updates on stored strategies. We will clarify this in the revision.
>
> - Regarding memory scaling, efficient retrieval over large stores is a well-studied problem in RAG systems, which we considered beyond scope, but we agree a fuller discussion would benefit the reader. We will add a section on ANN indexing, dimensionality reduction, hierarchical retrieval, compression, and TTL-based pruning (the latter also addresses staleness and domain shift concerns by down-weighting or expiring strategies from earlier task distributions).
>
> Regarding your point about safety, we do discuss dual-use considerations in our Impact Statement, but we agree with you and will make the point about cheaper scaling amplifying misuse risks more explicit.
>
> We hope our responses address your questions. Please let us know if you have further comments and thank you for the constructive feedback and for helping us improve the manuscript.
>
> Kind regards,\
> The Authors of Submission #24438
>
> ---
> [1] (Li et al., 2026) Grading Scale Impact on LLM-as-a-Judge

---

> > ### Author Rebuttal · Reviewer_A2EB · 2026-04-03
> >
> > Thank you for the careful rebuttal.
> >
> > The new judge sensitivity ablations reduce my concern that the gains hinge on a particular evaluator. The human validation is especially valuable. Spearman correlations are strong across bins, for example 0.7571 at tau at most 0.1 and 0.7396 at tau at most 60, and you report p below 0.001 throughout. That is credible evidence that the jury scores track expert judgments.
> >
> > On transfer across model families and price regimes, I agree this remains future work. The argument that the features are largely model agnostic is plausible, and the fact that the MILP retuning is cheap makes this less concerning in practice. It would still be good to add at least one cross family pilot if space permits.
> >
> > On tool latency, the modular cost function story makes sense. I would encourage you to add a small deployment style experiment or simulation where tool calls dominate, to show that the cost proxy can be adapted without changing the auction mechanism.
> >
> > On memory growth and staleness, the plan to add discussion of retrieval and pruning is appropriate. If you can include a simple robustness check that varies the memory size or uses a TTL style pruning rule, that would strengthen the paper further.
> > Provided the above, I am keeping my initial assessment.

---

> > > ### Author Response · Authors · 2026-04-06
> > >
> > > Dear Reviewer A2EB,
> > >
> > > Thank you for your patience while we ran the additional experiments you recommended in your latest reponse. Below we report results for all three: a cross-family transfer experiment, a tool-latency simulation, and a memory robustness check.
> > >
> > > **1) Cross-family experiment.** We froze the MILP weights learned on Qwen3 and introduced cross-family agents: Llama 3.1 8B and Ministral 3 8B. No retraining of the weights was performed. On a 100-task subset (evenly stratified by task type and complexity), the new agents receive non-trivial allocations that decrease with complexity, consistent with their 8B-class size:
> > >
> > > | Complexity | Llama 3.1 8B | Ministral 3 8B |
> > > |---|---|---|
> > > | τ ≤ 0.1 | 25% | 20% |
> > > | τ ≤ 0.5 | 20% | 15% |
> > > | τ ≤ 2.5 | 15% | 10% |
> > > | τ ≤ 12.5 | 15% | 10% |
> > > | τ ≤ 60 | 5% | 5% |
> > > | **Overall** | **16%** | **12%** |
> > >
> > > We also note that feature distributions support transferability: cross-family entropy values fall within 2% of Qwen3 8B, jury scores are at near-parity (e.g., Llama mean 3.48 vs Qwen3 8B 3.56), and per-task cost-minus-value scores correlate significantly with Qwen3 8B rankings (Spearman ρ = 0.4, p < 0.01).
> > >
> > > **2) Tool-latency simulation.** We augmented the cost proxy with a tool-latency term: $C_{t,i} = α·w_c·π(a_i)·|s_{t,i}| + β·T(s_{t,i})·λ$, where $T$ counts tool-use steps extracted from strategies and $λ = 2s$ is a synthetic average latency. The value function is unchanged. We swept $β$ from 0 (original) to 5 (strong tool penalty) on the same 100-task set:
> > >
> > > | β | 4B | 8B | 14B | 32B |
> > > |---|---|---|---|---|
> > > | 0.0 | 26% | 33% | 18% | 23% |
> > > | 1.0 | 29% | 34% | 21% | 16% |
> > > | 5.0 | 32% | 29% | 28% | 11% |
> > >
> > > The 32B agent, which has the highest per-token price and highest tool density in its generated strategies (2.48 mean calls vs 0.64 for 4B), drops from 23% to 11% (52% relative reduction). The 14B agent rises from 18% to 28%, offering the best cost-value tradeoff under tool latency. The effect is domain-sensitive: on deep search (where tool use is inherent), 32B share drops from 20% to 8%. The MILP converges to Optimal at every β, confirming the modular cost story from our earlier rebuttal.
> > >
> > > **3) Memory robustness and TTL pruning.** Using a leave-one-out protocol over the same 100 tasks, we varied memory bank sample size M ∈ {0, 25, 50, 75, 99} (5 random seeds each for sampling), and tested TTL pruning with windows W ∈ {10, 25, 50, 99}. Please see the results for each experiment in the two tables below.
> > >
> > > | Memory | Jury (original) | Jury (refined) | Flip rate | 32B share |
> > > |---|---|---|---|---|
> > > | M=0 | 3.36 | — | 0% | 35% |
> > > | M=25 | 3.38 | 3.68 | 19.1% | 17% |
> > > | M=50 | 3.36 | 3.67 | 17.5% | 17% |
> > > | M=75 | 3.42 | 3.72 | 17.3% | 16% |
> > > | M=99 | 3.41 | 3.74 | 17.2% | 17% |
> > >
> > >
> > > | TTL window size | Jury (original) | Jury (refined) | Flip rate | 32B share |
> > > |---|---|---|---|---|
> > > | W=10 | 3.47 | 3.68 | 16.1% | 21% |
> > > | W=25 | 3.45 | 3.68 | 16.1% | 21% |
> > > | W=50 | 3.45 | 3.65 | 16.7% | 19% |
> > > | W=99 | 3.46 | 3.74 | 16.7% | 18% |
> > >
> > > The critical transition is from M=0 to M>0, whereas increasing from a small to a large memory size yields a relatively small further improvement (within seed-level noise, σ = 1.7–2.6 pp). With as few as 25 records, memory already delivers a large effect: 32B reliance drops from 35% to ~17%, and refined plans score ~0.3 points higher than initial bids on the 0–5 jury scale. Even aggressive TTL pruning (W=10) produces comparable jury scores and flip rates, with only a modestly higher 32B share (21% vs 17%). In practice, a sliding window of ~10-25 recent tasks suffices for effective refinement.
> > >
> > > We plan to incorporate all three experiments into the revision, along with the expanded discussion of practical memory management strategies (please see our final response to Reviewer Tccx for the precise details of this discussion), which these results now concretely support. If you feel these additional experiments and results address your remaining concerns, we would be grateful if you would consider updating your assessment of the work.
> > >
> > > Thank you again for the constructive feedback which has strengthened the paper.
> > >
> > > Kind regards,\
> > > The Authors of Submission #24438

---

### Official Review · Reviewer_KWm8 · 2026-03-02

**Soundness:** 2
**Presentation:** 2
**Significance:** 2
**Originality:** 2
**Overall Recommendation:** 3
**Confidence:** 3

**Summary:**

The paper proposes a coordination framework inspired by auctions that decides routing of tasks to different sized LLM agents. The authors show that on deep-search and coding tasks, smaller agents performance degrades with task complexity without using their framework, and their framework mitigates this and recover large-agent quality at lower cost.

**Compliance With Llm Reviewing Policy:**

Affirmed.

**Final Justification:**

The auction-inspired routing idea is interesting and the benchmark is a solid contribution. However, the "auction" framing overstates the mechanism design content - the system is closer to plan-based routing with LLM judging and memory, without the strategic bidding, agent utilities, or incentive analysis that the terminology implies. Model diversity is also limited to a single family. The rebuttal provided helpful clarifications, but I believe the current score appropriately reflects my assessment of the paper after reading all the other reviews and comments. I maintain my score.

**Key Questions For Authors:**

See the three points in the weaknesses section.

**Limitations:**

Yes.

**Strengths And Weaknesses:**

**Strength**:

The overall direction and idea of coordinating small agents to achieve large agent performance at reduced cost is interesting; the paper is overall well written and empirical section is detailed.

**Weaknesses**:

The authors frame the paper around "strategy auctions" and bidding, but after reading the paper, the "auction" label feels more like a catchy metaphor rather than how "auctions" are rigorously defined and studied. What the authors build reads more like plan-based routing with LLM judge, coupled with some retrieval memory. This is different than how auctions are studied and defined in a long trail of work spanning decades in economic theory and computer science, where the key object is about strategic bidding (where agents with incentives lie and bid values different than their true value to gain a better utility), and specific designer goals to specify allocation and payment rule with properties like IC, IR, revenue, etc. In the current paper, the "bid" is just a plan, "payment" is a proxy cost term (model price and plan length), rather than something actually paid by, or even meaningfully experience by, a bidder (here the LLM agents). As a result, it's unclear in what sense the bids are "strategic": the agents' utilities are not defined, there's no explicit objective they are maximizing and no equilibrium analysis, and the notion of incentive compatibility - where agents misreport information to improve their allocation or payment - doesn't really enter the analysis theoretically. The pap[er does include a sketch formalization in Section 5, but again it doesn't connect with the usual game-theoretic picture where bidders choose strategies to maximize expected utility as a function of allocation and payments, nor does it justify how to model LLM behaviors. I strongly suggest that the paper either (a) rigorously formalize the auction mechanism being played here, in line with relevant game theoretic / mechanism design literature, and cast the game here with LLM to the setting under reasonable theoretical assumptions, or (b) actually tone down the auction terminology, and clearly state it in terms of "metaphor" or "analogy" about how the framework actually behaves.

One practical concern seems to be that this coordination might add inference time due to latency caused by coordination and voting. The authors provide token count but no wall clock runtime numbers. Are there numbers on the inference runtime?

The authors mention in Section 5 that "It is worth noting that both jury scoring and refinement incur only a small additional inference cost, on the order of a few hundred tokens...Thus, the overhead introduced by the auction mechanism is negligible relative to the overall test-time compute". This seem kind of surprising as there are well document line of research that focuses on how LLM judges, even with substantial reasoning traces, are sometimes unreliable (e.g. Huang, Jie, et al. 2023). The fact that the added mechanism costs 'negligible' tokens seem surprising: if the judge isn’t even allowed to articulate any reasoning, why should we expect that single discrete score to be stable, well-calibrated, or even meaningfully correlated with downstream success? Right now, most of the evidence seems indirect (end-to-end gains and ablations), which doesn’t really tell us whether the score itself is a good signal versus the method benefiting from other components like “plan-conditioning” or simply adding extra structure/prompting. It would strengthen the paper a lot to include a more direct quality check of the jury signal, since a great part of the paper is about this auction-like framework. This could help understand what is driving the gains.

The gains v.s. a single agent baseline is consistently notable but small. Not a big issue though as the claim is on token efficiency.

---

> ### Author Rebuttal · Authors · 2026-03-30
>
> Dear Reviewer KWm8,
>
> Thank you for your review. We are glad that you found the idea interesting and the empirical sections detailed. We address your concerns below.
>
> >**Weakness 1:** *the "auction" label feels more like a catchy metaphor*
>
> - Thank you for raising this. While SALE is not a classical single-item auction, the mechanism does fit established formal definitions in the economics literature. Freelancer platforms are formally studied as auctions [1], and procurement mechanisms where bids are scored on both price and quality are formally defined as scoring auctions [2]: bidders submit multi-attribute proposals, a predetermined scoring rule evaluates them, and the highest-scoring bid wins, without requiring properties like incentive compatibility. SALE matches Che's definition of a scoring auction: agents submit strategic proposals, the scoring rule evaluates them on cost and value, and a centralised mechanism selects the winner.
>
> - Moreover, we perform game-theoretic analysis in Section 7.2, where we define a cooperative game over the agent pool and use Shapley values to quantify each agent's marginal contribution. The use of Shapley values to analyse auction outcomes is established in the economic theory literature [3, 4].
>
> - With all that said, we do take your point that the connection to the mechanism design literature could be made more transparent in the paper. We will add a discussion explicitly mentioning that SALE does not implement a classical auction and instead identifying SALE as a scoring auction as defined by Che [2], along with placing the Shapley analysis within this framing.
>
> >**Weakness 2:** *The authors provide token count but no wall clock runtime numbers. Are there numbers on the inference runtime?*
>
> - Below is the average wall time of the auction phase computed on a subset of 100 HST-Bench tasks, evenly stratified by type and complexity. Agents are deployed as vLLM servers. For each task, strategy generation and jury scoring before and after memory retrieval are parallelised: agents are queried concurrently in each phase using asynchronous requests, so wall time is bounded by the slowest model being queried.
>
> |GPU type|GPUs per node|Avg. wall time per sample|
> |-|-|-|
> |H100|8|3.59s|
>
> - For comparison, executing the best-performing agent's full trace takes 5-10 minutes avg., accounting for token generation and tool calls (indeed, running the entire HST-Bench with our ARE setup takes multiple days). The auction overhead therefore represents ~1% of total wall time, with the benefit of potentially routing to a cheaper and faster agent. We will add the wall-time analysis to the camera-ready.
>
> >**Weakness 3:** *why should we expect that single discrete score to be stable, well-calibrated, or even meaningfully correlated with downstream success?*
>
> - We appreciate the concern. Score miscalibration was a well-documented issue with older LLMs, particularly without CoT, but recent models are substantially better calibrated and have even been observed to produce better-aligned scores without CoT than with it [5]. Our use of direct 0-5 Likert scores is aligned with recent work [6], which finds that 0-5 elicitation without CoT yields strong human alignment (including with Qwen3 models). Please see also our response to reviewer A2EB’s Question 2 below, demonstrating strong positive correlation between the LLM-jury scores in SALE and expert human scores. We will add to the camera-ready both the above citations and this correlation analysis.
>
> - Please also note that the multiple judges in the jury provide further regularising effect, diminishing potential miscalibration in individual scores. Our jury ablations in Appendix I confirm this empirically.
>
> >**Weakness 4:** *The gains v.s. a single agent baseline is consistently notable but small. Not a big issue though as the claim is on token efficiency*
>
> - Exactly, and we would add that since existing routers either underperform the largest agent or fail to reduce cost (often both, see Table 1), even just matching the largest agent's pass@1 while reducing cost would already be a strong result. SALE consistently surpasses it by ~3% absolute on average, while reducing cost by 35% and offloading 53% of the workload from the largest agent.
>
> Please do let us know if our responses address your concerns. We believe the above additions will strengthen the paper, and would like to thank you for helping us improve the work.
>
> Kind regards,\
> The Authors of Submission #24438
>
> ---
> [1] (Hong et al., 2016) Comparing Open and Sealed Bid Auctions: Evidence from Online Labor Markets\
> [2] (Che, 1993) Design Competition Through Multidimensional Auctions\
> [3] (Van Essen & Wooders, 2021) Allocating Positions Fairly: Auctions and Shapley Value\
> [4] (Lindsay, 2018) Shapley Value Based Pricing for Auctions and Exchanges\
> [5] (Wang et al., 2025) Improving LLM-as-a-Judge Inference with the Judgment Distribution\
> [6] (Li et al., 2026) Grading Scale Impact on LLM-as-a-Judge

---

> > ### Author Rebuttal · Reviewer_KWm8 · 2026-03-31
> >
> > I appreciate the authors' response. I appreciate the explanation about the scoring auction framing rather than the classical auction. That said, my concerns was about the paper reading more as plan-based routing with judge scoring and memory than as an auction mechanism in the mechanism design sense, with clearly defined agent utilities, strategic behavior, or incentive properties that typically associate with the word "strategy auction". The additional information on 0-5 judge score is helpful. I view it as only partially resolved, since the evidence remains limited and does not yet fully establish calibration, robustness, or direct predictiveness for downstream success, but I think those are beyond the scope of a short rebuttal.
> >
> > I will keep my score for now.

---

> > > ### Author Response · Authors · 2026-04-01
> > >
> > > Dear Reviewer KWm8,
> > >
> > > Thank you for the follow-up. We appreciate your engagement with our rebuttal and would like to briefly clarify two points.
> > >
> > > **1) Auction framing.** We appreciate you acknowledging the scoring auction connection. We would like to gently clarify one point regarding the characterization of SALE as "plan-based routing with judge scoring." While this is a useful high-level intuition, it does not fully capture the mechanism: routing decisions are not based solely on jury scores, but on a multi-attribute evaluation of each agent’s proposed strategy, combining value signals (including jury scores and entropy) with explicit cost proxies.
> > >
> > > This distinction is important: the mechanism evaluates agents’ proposed plans under a fixed scoring rule over multiple attributes, rather than ranking candidates based on a single judged signal. In this sense, SALE is aligned with scoring auctions, where bids consist of multi-dimensional proposals and selection is based on a deterministic scoring function over price and quality.
> > >
> > > That said, we fully agree with your broader point that the current paper does not explicitly map SALE onto classical mechanism design concepts such as agent utilities, strategic behaviour, or equilibrium analysis. In the revision, we will (a) explicitly position SALE within the scoring auction framework, (b) clarify where it departs from classical mechanism design formulations, and (c) present the Shapley analysis in Section 7.2 as a complementary perspective on agent contribution.
> > >
> > > We believe this will make the intended connection precise while avoiding overstatement. Would this resolve your remaining concerns?
> > >
> > >
> > > **2) Judge calibration and predictiveness.** We appreciate your noting that a full calibration study is beyond the scope of a short rebuttal. To complement the evidence already provided (0–5 jury scores, recent findings on Likert-scale elicitation, correlation analysis between jury and blind expert human scores), we also gathered an additional piece of direct evidence: the point-biserial correlation between the continuous jury scores and the binary downstream task accuracy, computed on the full development set:
> > >
> > > | point-biserial correlation | p-value |
> > > |---|---|
> > > | 0.36 | <0.01 |
> > >
> > > A point-biserial correlation in the 0.30–0.39 range is considered good item discrimination (Ebel & Frisbie, 1991). Together with the strong correlation between jury scores and blind expert human annotations (Spearman ρ = 0.806, p < .001, N = 100) - which serve as ground truth for plan quality - these results provide converging evidence that the scoring signal is both predictive of downstream success and aligned with human judgements.
> > >
> > > We would also like to draw attention to Table 17 of the paper, which ablates the jury component entirely: when strategy value is estimated without any jury assessment, downstream performance degrades. This is distinct from the per-judge ablations in Tables 18–20 and provides additional evidence that the jury signal is a meaningful driver of SALE's gains, over and above plan-conditioning or structural prompting effects.
> > >
> > > We hope these clarifications are helpful, and thank you again for the thoughtful review.
> > >
> > > Kind regards,\
> > > The Authors of Submission #24438

---

### Official Review · Reviewer_aKvh · 2026-03-11

**Soundness:** 3
**Presentation:** 3
**Significance:** 3
**Originality:** 2
**Overall Recommendation:** 5
**Confidence:** 4

**Summary:**

This paper proposes SALE, an auction-based strategy for multi-agent task allocation designed to improve the scalability of smaller language models by reducing reliance on larger models. The approach treats coordination between agents as a market-like process, where tasks are routed based on agent capabilities and estimated utility. The authors argue that this mechanism allows smaller models to handle a larger share of tasks while preserving performance on more complex problems.

Overall, the idea of using auction-style routing to coordinate heterogeneous agents is interesting and computationally efficient. However, the evaluation currently relies on a limited set of models and relatively small experimental sample sizes, which makes it difficult to assess how broadly the reported improvements generalize.

**Compliance With Llm Reviewing Policy:**

Affirmed.

**Final Justification:**

The authors addressed my concerns during the rebuttal discussions. The biggest concern of model diversity which was the main reason behind my initial score of 4, instead of 5, has been addressed so I have raised my score to 5.

**Key Questions For Authors:**

1. **Model diversity:**
   Why were the experiments limited to models from a single family? Since the framework is intended for heterogeneous agents, evaluating across multiple model families could provide stronger evidence of generality.

2. **Experimental variance:**
   Given the stochastic nature of smaller models, have the authors evaluated the system with more than five runs per task, or reported variance statistics?

3. **Text consistency:**
   The last paragraph on **page 3** and the following paragraph on **page 4** appear to make contradictory statements. Could the authors clarify the intended interpretation?

4. **Performance degradation:**
   In Figure 6, why does the performance of the smallest model begin to decline after approximately 150 coding tasks? Is this due to routing behavior, model limitations, or task distribution effects?

**Limitations:**

yes

**Strengths And Weaknesses:**

## Strengths

**Compute-efficient coordination mechanism.**
The proposed auction-based allocation mechanism appears computationally efficient and could serve as a practical method for distributing tasks among agents of varying capability.

**Compatibility with existing agents.**
The framework is designed to work with off-the-shelf agents, which makes the approach easier to integrate into existing systems.

**Market-style coordination framework.**
The idea of using a market-inspired routing mechanism to coordinate heterogeneous agents is conceptually interesting and aligns with recent work on scalable agent systems.

**Potential for scaling smaller models.**
SALE attempts to reduce dependence on large models by allocating tasks more strategically across agents, which is a promising direction for improving the efficiency of multi-agent systems.

**Impact**

Efficient coordination mechanisms for heterogeneous AI agents are an important research direction, particularly as multi-agent systems become more common in LLM-based workflows. A market-style routing mechanism such as SALE could potentially reduce reliance on large models and improve the efficiency of distributed agent systems.

If validated more thoroughly, approaches like this could help make agent systems more cost-efficient and scalable, especially in settings where compute resources are limited. However, stronger empirical evidence and broader model evaluations would be necessary to fully assess the practical impact of the approach.



## Weaknesses

### Limited Model Diversity

The evaluation uses only four models, covering a size range from 4B to 32B parameters. This range is relatively narrow and may not provide sufficient diversity to support the paper’s broader claims about heterogeneous agent coordination.

Additionally, the models appear to come from a single model family, which somewhat contradicts the premise of heterogeneous agent systems. Evaluating across multiple model families would strengthen the empirical claims.


### Insufficient Number of Experimental Runs

The experiments use five runs per task, which is common in some LLM agent evaluations but may be insufficient in this setting. Smaller models inherently exhibit higher stochasticity and are more sensitive to prompt-ordering and sampling noise than flagship models. With only 5 runs, the standard error likely exceeds the performance gains reported, making it difficult to distinguish true architectural improvements from random variance.
Reporting additional runs or variance statistics would strengthen the reliability of the results.


### Overreliance on Appendices

Several important methodological details are deferred to the appendices. While appendices are useful for supplementary material, key components of the framework should be explained clearly in the main text to ensure the paper is self-contained.


### Incomplete Explanation of Key Metrics

The paper would benefit from clearer explanations of how certain metrics are computed, particularly:

- Entropy
- Judgment score

Since these metrics appear central to the routing and evaluation process, their definitions should be described more explicitly in the main paper.

---

> ### Author Rebuttal · Authors · 2026-03-30
>
> Dear Reviewer aKvh,
>
> Thank you for reviewing our paper. We are glad you found the work conceptually interesting and covering an important research direction. Please find our responses below.
>
> >**Weakness 1 & Question 1:** *evaluating across multiple model families could provide stronger evidence of generality*
> - This is a fair point and we discuss it extensively in Section 8. Our choice of a single family was mainly for scientific robustness: to analyse scaling behaviour without too many confounders, we needed contemporary, well-performing models of known sizes, with at least 4 checkpoints evenly spaced apart, sharing the same architecture and training recipe. This ruled out most API and open-weight models, and at the time of our experiments the Qwen3 family met all these requirements. Further, restricting heterogeneity to model size allowed us to isolate scaling effects and demonstrate that strategy-based routing surpasses existing routers within this controlled framework, which was the core goal of this work.
>
> - Additionally, running complex agentic tasks with long trajectories is expensive. As you observed, we did 5 runs for each task to have statistical power in our results. While adding more models can always help, we were constrained by the budget and decided to focus on statistical rigour over widening the experiment pool. We also highlight in Section 8 that SALE is, in principle, model-agnostic and only operates on natural-language strategies.
>
> - That said, we thank you for raising an important direction for future work, and we will make it more prominent in the paper.
>
> >**Weakness 2 & Question 2:** *The experiments use five runs per task, which is common in some LLM agent evaluations but may be insufficient*
> - 5 runs is actually a generous budget in these expensive agentic settings: established prior work on agent evaluation with growing test-time memory often evaluates on only 3 runs [1, 2, 3]. We opted for 5 specifically for robustness. Scaling beyond this becomes intractable since trajectories can reach up to ~1.5M tokens per task. But most importantly, we do report variance statistics in the paper: SDs as subscripts in Tab. 1, and both one-sample t-tests and bootstrap CIs in Appendix F. These confirm that SALE's improvements over all baselines are statistically significant on the full test set (all |t| > 5, p < .01), indicating that 5 runs provide sufficient statistical power to support our claims.
>
> >**Weakness 3:** *Several important methodological details are deferred to the appendices*
> - Given the 8-page limit for the main body, we prioritised presenting the core method, results and analysis, and deferred supporting details to the appendices to aid reproducibility. That said, we plan to use the additional camera-ready page to move some of the more important details into the main text. If there are specific sections you feel would benefit most, we would be happy to take your recommendation into account.
>
> >**Weakness 4:** *clearer explanations of how certain metrics are computed, particularly: Entropy, Judgment score*
> - Entropy is defined in Section 5 as the normalised entropy $H(s_{t,i})$ of the strategy, and its computation is detailed in Appendix D.1. Judgment scores are obtained via the judge prompt in Appendix C.4, where each agent in the jury assigns a 0-5 integer score to the strategy. We agree that surfacing a few more of these details in the main text would help the reader, and we will do so in the camera-ready using the additional page.
>
> >**Question 3:** *The last paragraph on page 3 and the following paragraph on page 4 appear to make contradictory statements*
> - We reviewed this transition but could not identify a contradiction in the paragraph split between pages 3-4 or the surrounding ones. We would be happy to revise if you could point us to the specific sentences.
>
> >**Question 4:** *In Figure 6, why does the performance of the smallest model begin to decline after approximately 150 coding tasks?*
> - We appreciate the question. Figure 6 plots the 4B agent's cumulative selection rate, not its performance. Under random ordering, a stationary selection rate would produce a flat curve; the sustained upward trend is evidence that memory systematically increases the 4B's competitiveness. The curve then plateaus, with a negligible fluctuation most visible in the coding split. Indeed, your task distribution hypothesis is correct: the fluctuations are due to the random interleaving of tasks of varying complexity.
>
> We hope our responses have addressed your concerns; we will make all the proposed changes to the paper. We would like to thank you for helping us strengthen the work.
>
> Kind regards,\
> The Authors of Submission #24438
>
> ---
> [1] (Wang et al., 2023) An Open-Ended Embodied Agent with Large Language Models\
> [2] (Ho et al., 2025) Abstract Reasoning Composition with Lifelong LLM Memory\
> [3] (Cao et al., 2025) A Dynamic Procedural Memory Framework for Experience-Driven Agent Evolution

---

> > ### Author Rebuttal · Reviewer_aKvh · 2026-04-03
> >
> > Thank you for categorically answering my questions and concerns. Most of my concerns have been addressed but I am still concerned about model diversity in the evaluations. Though I understand the experimental constraints the authors mentioned, the potential value of SALE is in task allocation for heterogeneous multi-agent systems. The current selection of models both in terms of model types and sizes is not diverse enough. This was the main reason why I scored this paper 4, instead of 5, and for now, I will maintain that rating.
> >
> > ***Clarification on Question 3***
> >  I think the last paragraph on page 3 that flows into page 4 can benefit from a more clear and explicit statement like ***Even in these cases, any reduction in token usage is typically insufficient to offset their higher per-token cost.*** I believe that will set up the next paragraph more clearly for the readers.
> >
> > ***Edit following the reply rebuttal***:
> > I thank the authors for going through the trouble of extra heterogeneous agent experiments. It was my main (not sole)  concern. The performed experiments address that concern so I’ll raise my score to 5.

---

> > > ### Author Response · Authors · 2026-04-06
> > >
> > > Dear Reviewer aKvh,
> > >
> > > We would like to thank you for stating that all your concerns have been addressed by our rebuttal. Thank you also for your patience while we ran two additional experiments to directly address your overall concern about model diversity. We present these two new pieces of cross-family evidence below.
> > >
> > > **1) Cross-family experiment with Gemma3 models.** We ran SALE's auction stage on the Gemma 3 family (4B, 12B, 27B), which differs from Qwen3 in architectural details and training recipe. We generated strategic plans for 100 HST-Bench tasks (evenly stratified by type and complexity) and computed the cost-value proxies (note that we used a Gemma 3 jury). We then ran the MILP independently on this Gemma-only pool to learn new auction weights from scratch. We highlight two findings:
> > >
> > > - The auction mechanism transfers cleanly: the MILP converges and produces complexity-sensitive allocations. On the hardest coding tasks (τ(t) ∈ (12.5, 60]), the auction routes 80% of work to Gemma 27B, mirroring the pattern observed for Qwen3 32B. On easier tasks, work is distributed more evenly, consistent with our finding that smaller agents are sufficient at low complexity.
> > >
> > > - Auction signals (in our cost and value formulations) also follow a similar trend across model families. Cross-family comparisons between size-matched Gemma 3 and Qwen3 agents (4B-4B, 12B-14B, and 27B-32B) show convergence across all three auction signals: strategy length distributions have Wasserstein distance as low as 9 tokens avg. for the largest models, jury score correlations are strong (Spearman ρ ∈ [0.45, 0.76], all p < 0.001), and entropy correlates significantly (Spearman ρ ∈ [0.46, 0.60], all p < 0.001) despite different absolute scales which the MILP compensates for via its learned weight w_h. This confirms that the two families' juries converge on the same task-level quality signal despite differing architectures and training recipes.
> > >
> > > **2) Heterogeneous agent pool.** To test SALE under truly heterogeneous conditions, as per your recommendation, we expanded the Qwen3 pool (4B, 8B, 14B, 32B) with two agents from different families: Llama 3.1 8B and Ministral 3 8B. Crucially, we froze the MILP weights learned on Qwen3 and performed no weight retraining: the cross-family agents were simply dropped into the existing auction. On the same 100 tasks, both new agents receive non-trivial workload shares that decrease with task complexity, exactly as one would expect for 8B-class models:
> > >
> > > | Complexity | Llama 3.1 8B | Ministral 3 8B |
> > > |---|---|---|
> > > | τ ≤ 0.1 | 25% | 20% |
> > > | τ ≤ 0.5 | 20% | 15% |
> > > | τ ≤ 2.5 | 15% | 10% |
> > > | τ ≤ 12.5 | 15% | 10% |
> > > | τ ≤ 60 | 5% | 5% |
> > > | **Overall** | **16%** | **12%** |
> > >
> > > In practice, the cross-family agents behave as expected under the auction: their entropy and jury scores are close to Qwen3 8B (e.g., Llama jury mean 3.48 vs 3.56), per-task cost-value scores correlate with Qwen3 8B rankings (Spearman ρ = 0.4, p < 0.01). This demonstrates that SALE can coordinate a genuinely heterogeneous mix of agents without any modification to the mechanism.
> > >
> > > Taken together, these results provide direct evidence that SALE's auction signals (strategy text, entropy, and peer jury scores) are not family-specific, and that the mechanism can coordinate agents across architectures and training recipes.
> > >
> > > **Regarding the page 3–4 transition,** thank you for the concrete suggestion. We agree that an explicit bridging statement would help the reader. Although the general trend we observe is that larger agents do not achieve shorter traces on complex tasks, we will add a clarification that even in individual cases where some reduction in token usage may be observed, it is typically insufficient to offset their higher per-token cost. This should set up the following paragraph more clearly, as you suggest.
> > >
> > > Given that model diversity was your sole remaining concern, we hope these new cross-family results address it. If you feel they sufficiently alleviate your concern, we would be grateful if you would consider updating your score. We will of course include these results in the camera-ready version.
> > >
> > > Thank you again for the constructive engagement throughout the review process and for your help improving the work.
> > >
> > > Kind regards,\
> > > The Authors of Submission #24438

---

### Official Review · Reviewer_Tccx · 2026-03-12

**Soundness:** 3
**Presentation:** 3
**Significance:** 3
**Originality:** 4
**Overall Recommendation:** 4
**Confidence:** 4

**Summary:**

This paper investigates the performance scaling of small vs. large language model agents across task complexity, introducing the HST-BENCH benchmark and the SALE framework. SALE leverages strategy auctions with cost-value scoring and memory-driven refinement to dynamically route tasks, reducing reliance on large agents by 53% and lowering costs by 35% while improving pass@1.  These results indicate that small agents can be effectively "scaled up" through coordinated allocation and test-time self-improvement. This work underscores that the evolution of agentic AI depends not merely on scaling parameter counts, but on sophisticated coordination mechanisms that organize heterogeneous agents into efficient ecosystems.

**Compliance With Llm Reviewing Policy:**

Affirmed.

**Final Justification:**

Thank you for your detailed reply. I have updated my final justification.

**Key Questions For Authors:**

As mentioned above.

**Strengths And Weaknesses:**

Strengths
- The solution is rigorously designed, with the core SALE framework forming a complete logical closed loop. Each link, from srategy bidding and cost and value assignment to strategy refinement from auction memory, has clear design rationales. Additionally, the objective function is formalized through min–max optimization, ensuring the scientific validity of the method.
- Addressing the contradiction where small language models lack sufficient performance on complex tasks while large models incur excessive costs, the paper proposes a solution for dynamically allocating model resources based on tasks. This aligns with the industrial demand for efficient agent deployment and possesses clear practical value.
- HST-BENCH contributes a complexity benchmark that links tasks to human solution time, quantifying the impact of task difficulty on model performance.

Weaknesses
- The SALE framework requires each agent to generate strategic plans and conduct mutual evaluations before execution. Although the authors claim the token overhead is negligible, for low-complexity tasks, this pre-inference cost may account for a significant proportion of the total cost.
- The proposed complexity metric HST may not fully capture the inherent challenges of LLMs, as the difficulties faced by LLMs—such as long-context memory management, tool-calling logic execution, and sensitivity to specific hallucinations—do not perfectly align with those encountered by humans. Certain tasks that humans find straightforward (e.g., processing ultra-large-scale text data) can be extremely challenging for small models. However, the paper fails to discuss potential routing failures arising from the misalignment between machine complexity and human complexity.
- The paper acknowledges potential risks associated with memory bank management and model version updates but lacks relevant sensitivity analysis in the experimental section. Specifically, as the number of tasks increases, the memory bank grows linearly. The retrieval cost based on cosine similarity will gradually rise, and old strategies may become invalid due to updates to the underlying model versions. The paper does not adequately discuss management mechanisms for the memory bank.

---

> ### Author Rebuttal · Authors · 2026-03-30
>
> Dear Reviewer Tccx,
>
> Thank you for taking the time to review our paper. We are glad that you found the design rigorous with clear rationales, and the work to have clear practical value. We address your concerns below.
>
> >**Weakness 1:** *for low-complexity tasks, this pre-inference cost may account for a significant proportion of the total cost*
> - The mean token overhead for the experiments shown in Tab. 1 was <1% of the mean trace length, which we believe is a negligible price to pay for the efficiency gains brought out by SALE. Even for the lowest-complexity tasks (τ ≤ 0.1), execution traces are ~20k–40k tokens (Fig. 2) and total auction token overhead remains small at ~10%. Importantly, this overhead enables system-level gains of 30–50% cost reduction (Tab. 1), making it a highly favourable trade-off. We will add these quantifications more explicitly in the paper.
>
> >**Weakness 2:** *The proposed complexity metric HST may not fully capture the inherent challenges of LLMs [...] However, the paper fails to discuss potential routing failures arising from the misalignment*
>
> - We appreciate this observation, as it gives us an opportunity to refine our argument in favour of the human solution time (HST) metric. Our choice of the HST metric to estimate task complexity is grounded in established literature [1] and also empirically validated by our experimental results. However, we are currently not highlighting this in the paper as strongly as we should. In Fig. 1 and 4, we show that agent performance follows a clear monotonic degradation across task groups binned by human solution time not just for one model, but for all of them. If HST was a poor proxy for LLM difficulty, we would expect these curves to be noisy or flat; instead, the consistent trend across all agents is strong evidence for the reliability of the metric on our task distributions. We will highlight this in Section 3.1 as it is actually quite an important insight.
>
> - Your point that human and LLM difficulty can diverge is fair in principle and there may be some tasks where this holds. However, we would like to clarify that SALE does not route tasks based on HST or any explicit difficulty estimate. HST is used only to bin tasks for analysis purposes; the actual routing is driven by the auction mechanism (strategy quality, peer scoring, cost-value trade-offs) shown in Section 5. Importantly, strategy quality naturally captures many nuances of LLM-specific difficulty, which is precisely why we chose it as the primary signal for routing.
>
> >**Weakness 3:** *as the number of tasks increases [...] the retrieval cost based on cosine similarity will gradually rise, and old strategies may become invalid due to updates to the underlying model versions. The paper does not adequately discuss management mechanisms*
>
> - We agree that a more in-depth discussion of memory management than what we currently have in the Limitations section would strengthen the paper. We would like to address your two points below:
>
> - Regarding the concern that model updates may invalidate stored strategies, we note that the memory bank stores task-level strategic plans representing high-level solution patterns (e.g., task decomposition, tool selection, search constraints). While at least one of the retrieved past strategies must come from the refining agent itself, the successful strategies it learns from may have been produced by any agent in the pool. So the memory captures what works for a given class of tasks rather than model-specific reasoning traces, which limits the impact of individual model updates. For minor version upgrades, high-level solution patterns are unlikely to change substantially. A major upgrade could potentially invalidate that model's contribution to the memory, though not other agents' entries. This is an important implementation-level consideration that we will add as a brief note in the revised manuscript.
>
> - As for the retrieval cost, we acknowledge that the memory bank grows linearly with the number of tasks. Since efficient retrieval over large stores is a well-studied problem in RAG systems, we consider implementing these solutions and providing their sensitivity analysis beyond the scope of the paper. However, we agree that given the system's practical utility, a fuller discussion would benefit the reader. We will add a discussion covering the most relevant approaches: approximate nearest-neighbour indexing, pre-search dimensionality reduction, hierarchical retrieval, memory compression, and TTL-based pruning policies (the latter would also address any staleness concerns, should they arise in practice).
>
> We hope the responses above, and the proposed additions to the manuscript, address your concerns. Do let us know if you have follow-up questions. Thank you for the constructive feedback and for helping us improve the paper.
>
> Kind regards,\
> The Authors of Submission #24438
>
> ---
> [1] (Kwa et al., 2025) Measuring AI Ability to Complete Long Software Tasks

---

> > ### Author Rebuttal · Reviewer_Tccx · 2026-04-02
> >
> > My questions are partially resolved, and I keep my score.

---

> > > ### Author Response · Authors · 2026-04-02
> > >
> > > Dear Reviewer Tccx,
> > >
> > > Thank you for your acknowledgement. We noticed that you selected "I have follow-up questions for the authors," and we would welcome the opportunity to address any remaining concerns. You noted that your questions are only partially resolved; if there are specific points you feel require further clarification, it would be very helpful to know which ones.
> > >
> > > As this is likely the last response the discussion structure allows on our part, we briefly re-address all three points you originally raised below in case it is helpful.
> > >
> > > **Regarding points 1 and 2**, we believe these are primarily exposition issues rather than fundamental ones, as the relevant information is already present in the paper (trace lengths across all complexity bins and auction overhead for Weakness 1, and the fact that HST is used only for analysis binning and not for routing for Weakness 2). That said, we have agreed to make both points more prominent in the revised manuscript.
> > >
> > > **Regarding point 3**, we have tested memory robustness experimentally, varying memory bank size and testing TTL pruning with different window sizes. We found that memory pruning with a sliding window of even just 10-25 recent tasks is already sufficient for effective refinement. Please refer to our final response to reviewer A2EB for the exact figures and an in-depth discussion of these results.
> > >
> > > Moreover, in the interest of precision, we have drafted a complete section on retrieval limitations and mitigation strategies that we will add. The section will replace and expand the current paragraph starting at line 427 of the Limitations section. Unfortunately, the full text exceeds the 5k character limit for this response, so we summarise its key points below:
> > > - We clarify that in our main experiments memory comprised at most ~750 records (the full HST-Bench), and retrieval via brute-force cosine similarity remained tractable (Appendix C.3), but sustained deployment over thousands of tasks raises two concerns: retrieval cost and memory quality.
> > > - For retrieval cost, we detail how linear scaling can be replaced with approximate nearest-neighbour (ANN) indexing. HNSW graphs (Malkov and Yashunin, 2020) achieve logarithmic-complexity search over million-scale stores and are the default index in Faiss (Douze et al., 2024). Additionally, inverted-file indexing with product quantisation (Jégou et al., 2011) offers a well-established alternative trading modest recall for substantially lower memory footprint. We clarify that these apply directly to our retrieval step without modifying the auction mechanism or cost–value objective.
> > > - For memory quality and staleness, we discuss recent empirical work identifying two failure modes relevant to SALE: error propagation, where inaccurate past experiences compound and degrade future performance, and misaligned experience replay, where subtly outdated records negatively influence current decisions (Xiong et al., 2025). In our setting, error propagation could arise if an incorrectly judged "winning" bid is later retrieved as a positive example for contrastive refinement. Misaligned replay could occur if agents are updated to newer model versions whose capabilities no longer match stored strategic patterns (e.g., a strategy optimal for one checkpoint may become suboptimal after a major upgrade altering tool-use behaviour or reasoning style).
> > > - We discuss several complementary mitigations integrable without altering the core auction mechanism. First, TTL pruning to expire records older than a configurable horizon, bounding memory size and limiting staleness exposure, citing Xiong et al. (2025) and referring to our own experiments confirming its effectiveness in our setting. Second, utility-based retention prioritising frequently retrieved records that led to successful outcomes, citing retrieval-frequency deletion policies (Xiong et al., 2025; Cao et al., 2025) and the forgetting-curve decay in MemoryBank (Zhong et al., 2024). Third, hierarchical summarisation consolidating clusters of similar strategy records into representative prototypes, preserving contrastive diversity while reducing raw record count, as in Park et al. (2023), Packer et al. (2024), and Sarthi et al. (2024).
> > > - We note that model version staleness is partially mitigated by our design choice detailed in Section 5: the memory stores task-level strategic plans rather than model-specific reasoning traces, capturing what works for task classes rather than idiosyncratic model behaviours.
> > > - We note that our new experiments added to the Appendix already evaluate sensitivity to memory bank size and TTL window length, and that our modular retrieval design makes further mitigation mechanisms straightforward to integrate.
> > >
> > > We hope that adding the above discussion and experiments to the paper, as per your recommendation, addresses your remaining concerns. Thank you again for your time and engagement with our work.
> > >
> > > Kind regards,\
> > > The Authors of Submission #24438

---

### Decision · Program_Chairs · 2026-04-30

**Decision:**

Accept (regular)

**Comment:**

This manuscript received 4,5,3,4 respectively.

The overall idea of the paper is very interesting and new. The proposed method are well-supported with method illustration, and experimental analysis. This direction has big potential to be improved as suggested by reviewers from different perspectives, however it could be categorized as future works.